# NEVER SADDLE FOR REPARAMETERIZED STEEPEST DESCENT AS MIRROR FLOW

**Tom Jacobs**   **Chao Zhou**   **Rebekka Burkholz**
CISPA Helmholtz Center for Information Security, Saarbrücken, Germany
{tom.jacobs, burkholz}@cispa.de

## ABSTRACT

How does the choice of optimization algorithm shape a model's ability to learn features? To address this question for steepest descent methods —including sign descent, which is closely related to Adam —we introduce steepest mirror flows as a unifying theoretical framework. This framework reveals how optimization geometry governs learning dynamics, implicit bias, and sparsity and it provides two explanations for why Adam and AdamW often outperform SGD in fine-tuning. Focusing on diagonal linear networks and deep diagonal linear reparameterizations (a simplified proxy for attention), we show that steeper descent facilitates both saddle-point escape and feature learning. In contrast, gradient descent requires unrealistically large learning rates to escape saddles, an uncommon regime in fine-tuning. Empirically, we confirm that saddle-point escape is a central challenge in fine-tuning. Furthermore, we demonstrate that decoupled weight decay, as in AdamW, stabilizes feature learning by enforcing novel balance equations. Together, these results highlight two mechanisms how steepest descent can aid modern optimization.

## 1 INTRODUCTION

Optimization is a central driver of modern machine learning. First-order methods are particularly common in deep learning, where models are heavily overparameterized and trained on highly non-convex objectives populated with many saddle points and multiple global minima. In this regime, the choice of optimizer is not merely about convergence speed (Pascanu et al., 2025): different algorithms can converge to different solutions with markedly different properties like generalization, sparsity, and robustness (Woodworth et al., 2020; Arora et al., 2019; Jacobs & Burkholz, 2025; Tsilivis et al., 2024).

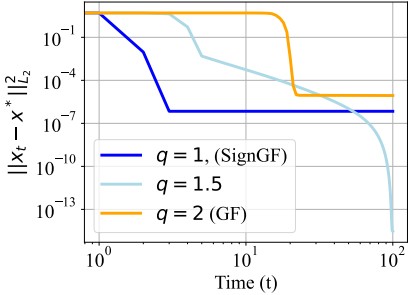

Figure 1: For a deep diagonal linear network initialized close to a saddle point, sign gradient flow (SignGF) converges faster than gradient flow (GF).

To understand the solutions that are preferred due to an interplay between overparameterization and the optimization algorithm, a geometric lens has proven especially useful. It is well known that overparameterization under gradient flow (GF) can induce mirror flows, changing the effective geometry in which optimization proceeds (Li et al., 2022). This perspective clarifies how symmetries and balance constraints are preserved, how implicit regularization emerges, and how specific design choices – like large learning rates, stochasticity, momentum, and explicit regularization – can shape learned solutions (Marcotte et al., 2023; Kunin et al., 2024; Gunasekar et al., 2017; Woodworth et al., 2020; Pesme et al., 2021; Even et al., 2023; Jacobs & Burkholz, 2025; Jacobs et al., 2025b; Papazov et al., 2024; Wang & Klabjan, 2024; Tarzanagh et al., 2023). Yet, most theories still center on gradient descent/flow, while modern practice in fine-tuning often operates in a setting where plain (Stochastic) Gradient Descent (SGD) with small learning rates underperforms. In contrast, Adam (Kingma & Ba, 2017) or AdamW (Loshchilov & Hutter, 2017) variants routinely deliver more robust and stronger results.

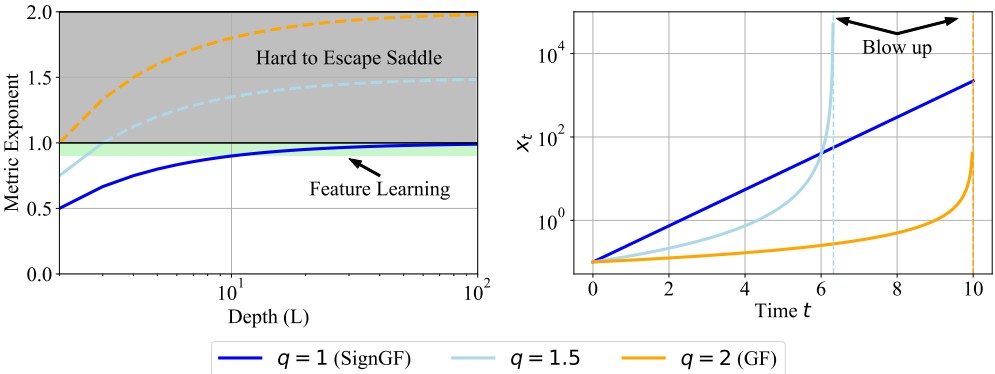

Figure 2: Illustration of different steepest mirror flows (with varied $q$). On the left side, the metric exponent is shown dependent on the associated depth. A high metric exponent increases the difficulty to escape zero and the instability of the flow. The right side illustrates saddle escape by plotting the solutions of the ODE's corresponding to the metric exponents, $dx_t = x_t^q dt$, with $x_0 = 0.1$ (from the origin). Concluding, SignGF does not get stuck near saddles and still allows feature learning by entering the green strip in the plot on the left, effectively inducing sparsity.

*Why do modern adaptive methods work so well in fine-tuning, and what solutions do they favor?* We approach this question by analyzing overparameterization and steepest descent methods via their resulting steepest mirror flows. Concretely, we study an optimizer family indexed by $q \in [1, 2]$ that interpolates between GF ($q = 2$) and SignGF ($q = 1$), where the latter closely related to sign-based methods exhibiting Adam-like behavior. Working in this broader geometric setting is technically more challenging than for gradient flow, as we lose the inner product structure, making the optimization process operate in a Banach instead of a Hilbert space.

Following Nam et al. (2025)'s call for simple, analytically tractable models that nevertheless reflect common phenomena, we focus on deep diagonal reparameterizations (a simplified diagonal proxy forthe dynamics of $KQ$ between the $K$ key and $Q$ query matrices in attention at depth ($L = 2$)) and diagonal linear networks. Within this setting, we derive new balance equations that characterize the induced mirror flows and the metric exponent governing dynamics as a function of depth (see Figure 2). This reveals a significant qualitative difference for varying $q$: steeper descent (smaller $q$, approaching SignGF) facilitates saddle-point escape and feature learning, while GF (larger $q$) typically requires unrealistically large learning rates to escape saddles (Pesme & Flammarion, 2023; Du et al., 2017), which is uncommon in fine-tuning. Here feature learning refers to induce sparsity in the learned representation. Moreover, we show that decoupled weight decay (AdamW) controls a different balance equation from GF, which stabilizes feature learning without driving the dynamics into high-exponent ($m > 1$) regimes that impede saddle escape. The high exponent regime corresponds to initial (exponential) slow down of convergence and finite time blow up corresponding to global instability. These findings are in line with empirical observations.

A scenario for which the implicit bias is known is classification on separable data. Recently, in this setting, max-margin characterizations have been derived for steepest descent (Tsilivis et al., 2025) and Adam (Zhang et al., 2024), establishing an $L_\infty$-margin for both SignGF and Adam. However, this margin does not see the full geometry induced by overparameterization, as our analysis shows. For diagonal deep networks, the $L_\infty$-margin would be independent of depth, whereas our findings reveal that the margin actually depends critically on depth through the geometry that controls feature learning by the metric exponent (see Figure 2) where a larger metric exponent leads to a sparser representation.

We validate these predictions for linear regression and separable binary classification, demonstrating ground-truth recovery and the predicted saddle-escape behavior. Fine-tuning experiments on standard vision tasks and LLM adaptation further corroborate the generality of our insights. Empirically, we find that Adam-like steepest flows escape saddles faster and achieve stable feature learning at small learning rates. Decoupled vs. coupled weight decay exhibits the anticipated sparsity and stability trade-offs, aligning with our balance-equation analysis.

**Contributions.**

- **Steepest mirror flows for a family of reparameterized steepest flow dynamics.** We develop a framework connecting reparameterizations to steepest mirror flows for a family of steepest descent methods in separable settings, combining steepest descent and mirror geometry.

- **Qualitative gap between GF and SignGF.** For deep diagonal reparameterizations, we prove that steeper descent (lower $q$) simultaneously escapes saddles faster and supports feature learning for deeper networks, whereas GF requires time rescaling / large learning rates to achieve comparable escape.

- **Decoupled weight decay for stability and sparsity.** We show that AdamW-style decoupled weight decay enforces distinct balance equations from GF, yielding more stable feature learning and needs higher depth for sparsity.

- **Empirical validation in fine-tuning.** We corroborate our theory for diagonal linear models by fine-tuning vision models and LLMs, highlighting (i) faster saddle escape with Adam-like flows and (ii) the predicted differences between coupled vs. decoupled weight decay for sparse, reparameterized training.

## 2 RELATED WORK

**Mirror flow and reparameterizations**   Specific reparameterizations trained with gradient flow induce a mirror flow Li et al. (2022). This finding has been used to describe the implicit regularization induced by overparameterization (Azulay et al., 2021; Vaškevičius et al., 2019; Zhao et al., 2022; Li et al., 2021; Gunasekar et al., 2017; Woodworth et al., 2020), explaining, why highly overparameterized neural networks can generalize well despite the risk of overfitting. Even the effect of large learning rates, stochastic noise, explicit regularization, and momentum can be covered by the theory (Pesme et al., 2021; Even et al., 2023; Jacobs & Burkholz, 2025; Jacobs et al., 2025b; Papazov et al., 2024). Generalizing these results that apply to gradient flows, we extend the mirror flow analysis to steepest flows. This includes sign gradient descent, which has a similar implicit bias as Adam (see Appendix A). As a highlight, we characterize the mirror flow stability with respect to the depth and type of descent algorithm. From a technical point of view, our derivations overcome the challenge that, unlike gradient flows that operate in Hilbert spaces, steepest descent algorithms live in Banach spaces. (Banach spaces have less mathematically convenient structure, as norms but not necessarily scalar products are defined.)

**Application of reparameterization to sparsity**   Recent work has used the implicit bias of reparameterizations to induce sparsity. (Jacobs & Burkholz, 2025; Gadhikar et al., 2025; Jacobs et al., 2025a) employ the mirror flow framework for gradient flows to guide the (re-parameterized) training dynamics, which are controlled by explicit regularization (Jacobs et al., 2025b). The analysis is centered around vision benchmarks where stochastic gradient descent with momentum is usually preferred over Adam. Kolb et al. (2025); Ziyin & Wang (2022) also exploit that reparameterized loss functions with $L_2$-regularization are equivalent to a differently regularized optimization problem in the original parameters. Combining deep pointwise reparameterizations with weight decay, Kolb et al. (2025) observe that higher depth leads to extreme sparsity and performance degradation. For sign gradient descent we show that decoupled weight decay, in contrast, actually needs higher depth to induce sparsity. This reveals a key difference between coupled and decoupled weight decay.

**Steepest descent and saddles**   Recent studies have revisited steepest descent as a unifying lens for understanding optimization in modern machine learning. Fan et al. (2025) and Tsilivis et al. (2025) analyze the implicit regularization induced by different steepest descent algorithms in classification settings with separable data, showing that the iterates approach a particular max margin solution. Building on this line of work, Large et al. (2024) and Bernstein & Newhouse (2025) highlight how modular duality provides a basis for steepest descent based algorithm design. A similar max margin implicit bias characterization has been provided for adaptive algorithms, including Adam (Zhang et al., 2024). For AdamW, the effect of decoupled weight decay on implicit bias can be expressed as a bound on the $L_\infty$ norm for general objective functions (Li et al., 2025). The convergence of sign gradient descent, an optimizer with implicit bias similar to Adam, has also been studied, connecting

its behavior to Lipschitz smoothness and yielding looser convergence bounds than gradient descent (Balles et al., 2020), with comparable rates in settings with unbounded smoothness (Crawshaw et al., 2022). As we show, overparameterization can lead to faster convergence for sign gradient flow than for standard gradient flow, which we attribute to better saddle point escape.

In finetuning, a small learning rate is preferred to not alter the representation to much too prevent catastrophic forgetting (Zhou et al., 2025). This clashes with the fact that saddle point escape needs time rescaling in gradient flow dynamics (Pesme & Flammarion, 2023). Note that different mechanisms that have been shown and studied allowing for saddle point escape are large learning rate and noise perturbation (Jin et al., 2017; Fang et al., 2020; Roy et al., 2020). In contrast, our analysis reveals a different mechanism which only relies on the geometry of the dynamics. As we show in experiments (Figure 5a), SGD with a small learning rate can not escape saddle points while Adam can.

**Conservation and algebraic invariance**   The reason why reparameterizations can induce a mirror flow is that gradient flow satisfies symmetries that do not change during training (Marcotte et al., 2025; 2024; 2023), i.e. so called balance equations. The scale and the relative scales of these invariances are important. Note that the relative scale is also referred to as $\lambda$-balance (see Definition 3.3). A slight initial imbalance can support feature learning, according to (Kunin et al., 2024). The gradient flow of deeper networks has also been studied under balanced invariance as a dynamical system Arora et al. (2019); Gadhikar & Burkholz (2024); Gadhikar et al. (2025); Boursier et al. (2022). Even exact solutions have been derived for two layer networks using a Ricatti equation (Dominé et al., 2024; Saxe et al., 2014; Xu & Ziyin, 2024). Less is known about steepest descent algorithms. We show that the relative scale for steepest descent optimizers can differ significantly, explaining, why sign gradient descent can train relatively faster than gradient descent.

## 3   BACKGROUND: REPARAMETERIZATION AND MIRROR FLOW

Consider minimizing a continuously differentiable objective $f \in C^1(\mathbb{R}^n, \mathbb{R})$. This can be accomplished with gradient descent: $x_{k+1} = x_k - \eta \nabla_x f(x_k)$, $x_0 = x_{\text{init}}$, where $\eta > 0$ is the learning rate. We study the resulting flow by taking the learning rate $\eta \to 0$, resulting in the differential equation: $dx_t = -\nabla_x f(x_t)dt$, $x_0 = x_{\text{init}}$.

**Reparameterizations and mirror flow**   Training reparameterizations of $x$ with gradient flow have been connected to mirror flows (Li et al., 2022; Jacobs et al., 2025b). (See Appendix C for a summary). Concretely, consider the reparameterization $g \in C^1(M, \mathbb{R}^n)$, assuming that $M$ is a smooth manifold. This corresponds to the gradient flow: $dw_t = -\nabla_w f(g(w_t))dt$, $w_0 = w_{\text{init}}$. Under suitable conditions, this can be described by a mirror flow:

$$d\nabla_x R(x_t) = -\nabla_x f(x_t)dt, \qquad x_0 = x_{\text{init}}, \tag{1}$$

where $R : \mathbb{R}^n \to \mathbb{R}$ is a Legendre function (see Definition 3.1). A mirror flow can control the implicit bias (Sun et al., 2022; Pesme et al., 2024; Gunasekar et al., 2018), i.e. the type of solution we converge to.

**Definition 3.1.** (Legendre Function, Definition 3.8 ((Li et al., 2022))) Let $R : \mathbb{R}^d \to \mathbb{R} \cup \{\infty\}$ be a differentiable convex function. We say $R$ is a Legendre function when the following holds: 1) $R$ is strictly convex on the interior of its domain $\text{int}(\text{dom}R)$. 2) For any sequence $\{x_i\}_{i=1}^{\infty}$ going to the boundary of $\text{dom}R$, the gradient diverges, i.e. $\lim_{i \to \infty} ||\nabla_x R(x_i)||_{L_2}^2 = \infty$.

*Example* 3.2. Let the reparameterization $g : \mathbb{R}^n \times \mathbb{R}^n \to \mathbb{R}^n$ be a deep diagonal linear network $g(m, w) = m \odot w$ or equivalently $g(m, w) = \text{diag}(m)\,\text{diag}(w)$. Assuming $|w_{i,\text{init}}| < m_{i,\text{init}}$, the corresponding Legendre function is:

$$R(x) = \frac{1}{2} \sum_{i \in [n]} x_i \operatorname{arcsinh}\left(\frac{x_i}{\lambda_i}\right) - \sqrt{x_i^2 + 2\lambda_i^2} - x_i \log\left(\frac{m_{i,\text{init}} + w_{i,\text{init}}}{m_{i,\text{init}} - w_{i,\text{init}}}\right), \tag{2}$$

where $\lambda_i = m_{i,\text{init}}^2 - w_{i,\text{init}}^2$. This corresponds the hyperbolic entropy which interpolates between $L_1$-norm ($\lambda \to 0$) and $L_2$-norm ($\lambda \to \infty$) implicit bias (Woodworth et al., 2020). Moreover, $R$ is also a Bregman function B.9, which is a property necessary for convergence.

In Example 3.2, $\lambda$ controls the relative scale. This is connected to the preserved balance by gradient flow. Similar balance equations exist for products of matrices. The small scale is associated with sparsity and with this inducing feature learning. Furthermore, the reparameterization can be used as a proxy for the key $K$ and query $Q$ matrices in attention (Tarzanagh et al., 2023; Jacobs et al., 2025b; Marcotte et al., 2025).

**Definition 3.3.** A product of parameters $m \in \mathbb{R}^n$ and $w \in \mathbb{R}^n$ is called $\lambda-$balanced iff $m^2 - w^2 = \lambda \mathbf{1}_n$, where we used the convention $m^2 = m^{\odot 2}$, i.e., element-wise multiplication and $\mathbf{1}_n$ the all one vector.

Marcotte et al. (2023) have shown that, if Definition 3.3 is satisfied, then balance is preserved under gradient flow for the more general matrix case. In other words, the parameters stay $\lambda$-balanced during training. This establishes a connection between mirror flows and the balance equation.

**Implicit bias and linear regression**  For mirror flows, the implicit bias for linear regression tasks can be characterized for general data sets. Let $\{(z_i, y_i)\}_{i=1}^k \subset \mathbb{R}^n \times \mathbb{R}$ be a dataset consisting of $k$ samples with $n$ features. The output of a linear model with parameters $x$ on the $i$-th data is $z_i^T x$. The goal is to solve the regression to predict the target vector $Y = (y_1, y_2, \ldots, y_k)^T$ based on input vector $Z = (z_1, z_2, \ldots, z_k)$. The next theorem establishes a mirror flow in this setting.

**Theorem 3.4.** *(Theorem 3.9 (Li et al., 2022)) Given $(Z, Y)$, suppose the objective $f(x)$ is of the form $f(x) = f(Zx)$ for some differentiable $f : \mathbb{R}^n \to \mathbb{R}$. Initialized at $x_0 = x_{init}$, assume that the mirror flow Eq. (1) converges to $x_\infty = \lim_{t \to \infty} x_t$, which satisfies $Zx_\infty = Y$, then*

$$D_R(x_\infty, x_0) = \min_{x \in \mathbb{R}^n} D_R(x, x_0), \text{ where } D_R(x, x_0) := R(x) - R(x_0) - \langle \nabla_x R(x_0), x - x_0 \rangle.$$

*$D_R$ is also known as the Bregman divergence (Definition B.8) with respect to $R$.*

Theorem 3.4 associates the Bregman divergence $D_R$ with the limits of a mirror flow. In Example 3.2, if $R$ is the hyperbolic entropy (Eq. (2)), a balancing constant $\lambda \to 0$ induces a feature learning regime and controls the strength of the induced sparsity bias. In conclusion, the reparameterization and $\lambda$ allow us to control the implicit bias.

**Inducing sparsity with reparameterizations**  Reparameterizations have been used to induce sparsity in deep learning architectures (Ziyin & Wang, 2022; Kolb et al., 2025; Jacobs & Burkholz, 2025) by exploiting the equivalence between the following optimization problems:

$$\min_{m,w \in \mathbb{R}^n} f(m \odot w) + \alpha \left( ||m||_{L_2}^2 + ||w||_{L_2}^2 \right) \text{ and } \min_{x \in \mathbb{R}^n} f(x) + 2\alpha ||x||_{L_1}.$$

Hence, their local minima correspond to each other, see (Theorem 2 in (Ziyin & Wang, 2022)).

## 4 THEORY: STEEPEST MIRROR FLOW AND DEEP REPARAMETERIZATIONS

To characterize the difference between modern optimizers Adam ($\simeq$ SignGF) and SGD ($\simeq$ GF), we study reparameterized steepest flows as steepest mirror flow. Our analysis is especially relevant for the finetuning setting, where small learning rates are used.

**Steepest flows**  We consider a class of algorithms that is based on steepest descent with respect to the $L_p$ norm. These are captured by the unnormalized steepest flow:

$$dx_t = -\text{sign}\left(\nabla_x f(x_t)\right) \odot |\nabla_x f(x_t)|^{q-1} dt, \qquad x_0 = x_{\text{init}}, \tag{3}$$

where $q$ satisfies $\frac{1}{p} + \frac{1}{q} = 1$. Most interesting to us are gradient flow (GF) $p = 2$ $(q = 2)$ and sign gradient flow (SignGF) $p = \infty$ $(q = 1)$, which is a proxy for Adam (see Appendix A). On a technical note, we mention that the unnormalized flow is equivalent to the normalized flow up to a time rescaling (see Appendix B). The solution to the studied ODE does not have to be unique but can be interpreted in the Filippov sense (Filippov, 1988). In this setting, Gunasekar et al. (2018) argue that a similar implicit bias characterization as in Theorem 3.4 is not possible, except for $p = 2$, which corresponds to standard GF. Accordingly, this is also not possible for reparameterizations trained by Eq. (3). However, we can still study the induced dynamics to analyze the feasibility of feature learning. Our main objective is to make qualitative statements about the dynamics such as saddle point escape, stability and the effect of decoupled weight decay.

**Steepest mirror flows**    Consider a Legendre function $R$ (Definition 3.1). A steepest mirror flow with respect to the $L_p$ norm is given by:

$$d\nabla_x R(x_t) = -\text{sign}(\nabla_x f(x_t)) \odot |\nabla_x f(x_t)|^{q-1} dt, \qquad x_0 = x_{\text{init}}. \qquad (4)$$

For this class of flows, we can show convergence using the second order condition of coercivity as in Definition 4.1, i.e. the inverse Hessian is bounded from below by a positive constant.

**Definition 4.1.** We call a function $R \in C^2(\mathbb{R}^n, \mathbb{R})$ inversely $\mu-$coercive iff there exists a constant $\mu > 0$, the coercivity constant, such that for all $x \in \mathbb{R}^n$:

$$x^T \nabla_x^2 R^{-1}(x) x \geq \mu ||x||_{L_2}^2.$$

**Theorem 4.2.** *Let $R \in C^2(\mathbb{R}^n, \mathbb{R})$ be a separable function (Definition B.9) that is inversely $\mu$-coercive (Definition 4.1). Moreover, assume that the set $\{x \in \text{Dom } R : \min f(x)\}$ is non-empty and there exists a constant $B > 0$ such that for all $t > 0$, $|\partial_i f(x_t)| \leq B$ for all $i \in [n]$. Then the loss decays and satisfies:*

$$\int_0^\infty ||\nabla_x f(x_t)||_{L_2}^2 dt \leq (f(x_0) - f(x_\infty)) / (\mu B^{2-q}).$$

*Assume that $f \in C^1(\mathbb{R}^n, \mathbb{R})$ is strongly convex. Then for the iterates of Eq. (4) converges such that we have $\lim_{t\to\infty} x_t = x^*$ where $x^*$ is the unique minimizer of $f$ with linear rate $\mu B^{q-2}\Lambda$.*

Proof. The proof follows from tracking the evolution of the loss $f$ and the observation that for strongly convex functions the sign is only zero when the minimum is reached (see Theorem E.1).

Theorem 4.2 highlights the dependence of the convergence rate on the coercivity constant. As we will show, the coercivity will effectively correspond to how hard it is to escape the saddle point set.

**Deep diagonal reparameterizations**    For the deep diagonal reparameterization given by $x = g(w) = \Pi_{i=1}^L w_i$, as in Example 3.2, we can study the steepest flow with respect to the $L_p$ norm with decoupled weight decay as in AdamW (Loshchilov & Hutter, 2017) with $\frac{1}{p} + \frac{1}{q} = 1$. The flow is described for each $i \in [L]$ by:

$$dw_{i,t} = -\text{sign}(\nabla_{w_i} f(g(w_{i,t}))) \odot |\nabla_w f(g(w_{i,t})|^{q-1} dt - \alpha_t w_{i,t} dt \qquad w_{i,0} = w_{i,\text{init}}. \qquad (5)$$

As additional result, we show that all separable steepest mirror flows have a corresponding reparameterization in Appendix G.

Deep diagonal parameterization have inherent saddle points as characterized next by Theorem 4.3.

**Theorem 4.3.** *Assume that $\nabla_x f(0) \neq 0$. Then, in addition to the saddle points of $f$, the deep diagonal reparameterization $x = g(w) = \Pi_{i=1}^L w_i$ introduces saddle points at:*

$$S := \left\{ (w_1, \ldots, w_L) : \forall_{i,j \in [n]}, w_i = w_j = 0, \ w_k \neq 0 \text{ for } k \neq i, j \text{ and } i \neq j \right\}.$$

Proof. Apply the saddle point condition from Definition D.1 (see Theorem D.2).

Theorem 4.3 implies that small initializations are close to the set $S$. Our next derivation shows how steepest mirror flows can escape such saddle points. The escape rate depends on the following balance equations, which are satisfied by the dynamics.

*Remark* 4.4. The points of the set $S$ would not be saddle points of the regularized dynamics with coupled or decoupled weight decay. However, as we will see, the metric would still be smaller for larger $q$ indicating that escaping from near the set S would be harder for GF ($q = 2$) than SignGF ($q = 1$).

**Balance equations**    The balance equations of the next lemma are needed to derive a mirror flow.

**Lemma 4.5.** *Consider steepest descent with respect to $L_p$ and weight decay, with $\frac{1}{p} + \frac{1}{q} = 1$. Then, for a deep diagonal reparameterization, i.e., $x = g(w) = \Pi_{i=1}^L, w_i$ satisfies the following balance equation for $t \geq 0$ almost everywhere:*

$$|w_{i,t}|^q - |w_{j,t}|^q = (|w_{i,0}|^q - |w_{j,0}|^q) \exp\left(-q \int_0^t \alpha_s ds\right) \text{ for all } i, j \in [L]. \qquad (6)$$

Proof. It follows from deriving the evolution of the left hand side of Equation (13) (see Lemma E.2).

Lemma 4.5 leads to the following natural extension of Definition 3.3.

**Definition 4.6.** A product of parameters $m \in \mathbb{R}^n$ and $w \in \mathbb{R}^n$ is $\lambda - L_p$-balanced with $\frac{1}{p} + \frac{1}{q} = 1$, iff

$$|m|^q - |w|^q = \lambda \mathbf{1}_n,$$

where $\mathbf{I}_n \in \mathbb{R}^n$ is the all-one vector.

We illustrate Def. 4.6 in Fig. 3. Observe that for smaller $q$, we can move faster away from the origin in both parameters, providing intuition for the saddle escape. Note, there is no analogue that holds for general deep reparameterizations, as recently shown by Marcotte et al. (2025) for $q = 1$.

*Remark* 4.7. We focus on a fixed value $\lambda$ for all $x \in \mathbb{R}^n$. However as the analysis is pointwise, therefore, we can have different values for $\lambda$ per parameter.

**Saddle escape and stability** The next theorem shows that the invariances above induce a steepest mirror flow when weight decay is turned off. This allows us to quantify the coercivity constant and also the stability of the dynamics. Furthermore, we can derive explicit expressions for the seperable Bregman functions by considering $\lambda = 0$ or $L = 2$.

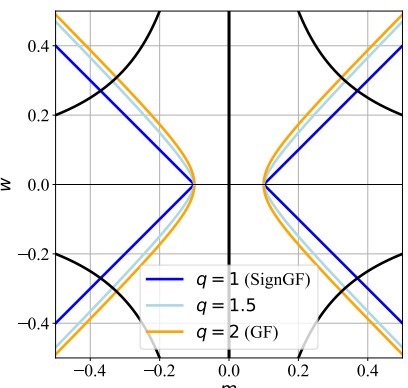

Figure 3: The balance equation for $q \in \{1, 1.5, 2\}$ and initialization $m = 0.1, w = 0$. Observe that the (curved) path away from the initialization to a point on the curve $mw = x$ with $x = \pm 0.1$ (in the plot) is shorter for smaller $q$, indicating faster saddle escape.

**Theorem 4.8.** *Initialize a deep diagonal reparameterization such that it is $\lambda - L_p$-balanced for a $\lambda > 0$ with respect to the first parameter $w_1$. Then, steepest descent satisfies a separable $L_p$-mirror flow almost everywhere:*

$$d\nabla_x R_{L_p,L}(x_t) = -sign\left(\nabla_x f(x_t)\right) \odot |\nabla_x f(x_t)|^{q-1} \, dt, \qquad x_0 = x_{init},$$

*where $R_{L_p,L} : \mathbb{R}^n \to \mathbb{R}^n$ is a seperable Bregman and Legendre function when $q\frac{L-1}{L} \leq 1$ completely characterized by the balances of Lemma 4.5. For $L = 2$, we explicitly get*

$$\nabla_x^2 R_{L_p,2}(x) := \frac{1}{\sqrt{4|x|^q + \lambda^2}}.$$

Proof. First, express the metric in terms of $|w_1|^q$ using the derived balances. Second, use the implicit function theorem to express $|w_1|^q$ as a function of $x$ and $\lambda$. For $L = 2$, we can do this analytically using the quadratic formula. To show $R_{L_p,L}$ is Bregman we use the properties of function $\nabla^2 R_{L_p,L}^{-1}$ such as being separable, bounded from below, asymptotic behavior near the boundary and being an even function. (Full proof see Theorem E.3.)

**Corollary 4.9.** *For a $\lambda - L_p$ balanced initialization, steepest descent has coercivity constant $\mu = \lambda^{L-1}$.*

Corollary 4.9 allows us to directly apply Theorem 4.2 for globally stable configurations such that $q\frac{L-1}{L} \leq 1$. Furthermore, at face value, Corollary 4.9 could indicate that all steepest descent methods have the same coercivity constant. However, the same initialization corresponds to very different $\lambda$ values for different $p$.

**Corollary 4.10.** *Initialize the reparameterization such that $w_1 = 0$ and $w_i = \mathbf{1}_n \lambda > 0$. Then, training in Eq. (5) is $\lambda^q - L_p$ balanced and $\mu = \lambda^{q(L-1)}$.*

Proof. Plug into Eq. (13) in Lemma 4.5.

Corollary 4.10 indicates that, for smaller $q$ and thus larger $p$, we indeed have a large coercivity constant and therefore can escape the saddle set $S$ faster. For small $\lambda$, the coercivity constant dominates the escape rate, as shown in Figure 1.

*Remark* 4.11. The case $p = \infty$, $L = 2$ corresponds to the same mirror map structure as smoothed sign gradient descent in (Wang & Klabjan, 2024).

For deeper insights into the dynamics, we are also interested in the shape of the Bregman function and its metric exponent, as defined next. This we can derive explicitly in case of $\lambda = 0$.

**Definition 4.12.** $m$ is called *metric exponent*, if $\lim_{|x| \to \infty} \frac{\partial^2 R^{-1}(x)}{|x|^m} = c$ for a constant $c \in (\mathbb{R}^+)^n$.

**Lemma 4.13.** *For $L \geq 2$ and $\lambda = 0$, we have:*

- *if $m = q\frac{L-1}{L} = 1$:*

$$R_{L_p, L}(x) = \frac{1}{L} \sum_{j \in [n]} (x_j log(x_j) - x_j - x_j log(x_{j,0}))$$

- *if $m = q\frac{L-1}{L} \neq 1$:*

$$R_{L_p, L}(x) = \frac{1}{L - (L-1) q} \sum_{j \in [n]} \left( \frac{|x_j|^{2 - q\frac{L-1}{L}}}{\left(\frac{q}{L} - q + 2\right)} - x_j x_{j,0} |x_{j,0}|^{q\left(\frac{1}{L} - 1\right)} \right).$$

*If $m = 1$, $R_{L_p, L}$ is a Bregman function with metric exponent $m$ on the domain $\mathbb{R}^{sign(x_{1,0})} \times \ldots \times \mathbb{R}^{sign(x_{n,0})}$. If $m < 1$, the domain is $\mathbb{R}^n$. Otherwise, $R_{L_p, L}$ is not a Bregman function.*

Proof. 1) Derive the inverse metric in terms $|x|$. 2) Integrate the metrics twice and use that $\nabla_x R(x_0) = 0$. (See proof of Lemma E.4).

Theorem 4.8 and Lemma 4.13 reveal a key distinction between GF ($\simeq$ SGD) and SignGF ($\simeq$ Adam). For GF with balanced initializations at higher depth, the smoothness condition of the Bregman function is not satisfied, but it is for SignGF. This distinction has implications for the stability of the dynamics. Accordingly, SignGF cannot escape beyond the boundaries of the Bregman function, making it globally stable which is captured by Corollary 4.14. Moreover, this corresponds to a large metric exponent ($m > 1$) as in Figure 2(b). As illustrated in the figure, the large metric exponent also leads to an initial (exponential) slow down of the convergence. Together this characterizes the stability of the dynamics. Furthermore, the gradient now may grow unbounded violating the assumptions in Theorem 4.2.

**Corollary 4.14.** *If $\lambda \geq 0$, then for $p = 2$, only $L = 2$ is a valid Bregman function. Furthermore, for $p = \infty$, $L \geq 2$ are all valid Bregman functions. For $p < 2$, there is no valid Bregman function.*

Recall that $\lambda$ needs to become very small for feature learning as it has to approximate the Bregman functions in Lemma 4.13 to induce sparsity. This we can accomplish with weight decay as shown in Lemma 4.5.

**The effect of weight decay** For gradient flow, the effect of explicit regularization can be integrated into a time-varying mirror flow (Jacobs et al., 2025b). For steepest flows, we can only study the Riemannian gradient flow, or, more specifically, the induced regularization on the manifold generated by the separable metric tensor $\nabla_x^2 R$. This informs us how regularization is affected by the geometry.

**Definition 4.15.** For the regularizer $h(x) = \sum_{i \in [n]} h_i(x_i)$ with each $h_i \in C^1(\mathbb{R}, \mathbb{R})$, the on manifold regularizer with respect to a separable $L_p$ steepest mirror descent characterized by $R$ is $M_{\text{reg}}(x) := \sum_{i \in [n]} \int^{x_i} \partial_i^2 R_i(x_i) \partial_i h_i(x_i) dx_i$, such that we have

$$d\nabla_x R(x_t) = -\text{sign} \left(\nabla_x f(x_t)\right) \odot |\nabla_x f(x_t)|^{q-1} dt - \nabla_x M_{\text{reg}}(x) dt, \qquad x_0 = x_{\text{init}}.$$

**Theorem 4.16.** *Assume a) $m = q\frac{L-1}{L} \neq 2$ or b) $m = q\frac{L-1}{L} = 2$. The manifold regularizer for decoupled weight decay with $L_p$ steepest descent on the manifold for a reparameterization of depth $L$ with balanced initialization ($\lambda = 0$) is: a) $\frac{L}{L(2-q)+q} \sum_{i \in [n]} |x_i|^{2 - q\frac{L-1}{L}}$ or b) $\sum_{i \in [n]} log(|x_i|)$.*

Proof. Use $\nabla_x^2 R$ from Corollary 4.13 and use $\partial_i h_i(x_i) = L x_i$. (See Theorem E.5.) □

*Example* 4.17. For $q = 2$ (GF) and $L = 2$, we recover $||x||_{L_1}$ as on manifold regularizer like Jacobs & Burkholz (2025). For finite depth $L$, we get a $|| \cdot ||_{L_1}$ sparsity bias for $q = \frac{L}{L-1}$, implying that for $q = 1$ (SignGF) we get $L \to \infty$.

In Theorem 4.16, we assume a balanced initialization ($\lambda = 0$). However, with sufficient amounts of weight decay, we know $\lambda \to 0$ "fast enough" during training according to Lemma 4.5. Hence, our insights generally also apply to $\lambda > 0$.

Example 4.17 establishes for SignGF ($q = 1$) that we need $L \to \infty$ to induce sparsity with explicit decoupled weight decay. This stands in stark contrast to coupled weight decay, which would induce extreme sparsity, as shown in Theorem 1 by Kolb et al. (2025). Table 1 provides an overview of the effect of weight decay on the induced regularization $M_{reg}$ for $L = 2$ and $L = \infty$.

Table 1: Comparison of the effect of coupled or decoupled weight decay ($M_{\text{reg}}$) for two reparameterization depths, namely, ($L = 2, L = \infty$). Note that the infinite depth would lead to a non-convex logarithmic regularizer (log) in the coupled case, potentially leading to instability.

|  | Coupled | Decoupled |
| --- | --- | --- |
| $q = 1$ (SignGF) | $(L_1, \log)$ | $(L_{\frac{3}{2}}, L_1)$ |
| $q = 1.5$ | $(L_1, \log)$ | $(L_{\frac{5}{4}}, L_{\frac{1}{2}})$ |
| $q = 2$ (GF) | $(L_1, \log)$ | $(L_1, \log)$ |

Note that these results imply that the respective flow cannot correspond to a time-varying steepest mirror flow, except for $q = 2$ (GF), which is covered by Jacobs et al. (2025b). This follows from Corollary E.6 in the appendix, according to which the manifold regularizer $M_{reg}$ would need to match weight decay, which is impossible for $q \neq 2$.

## 5 Experiments

The purpose of our experiments is to substantiate our theoretical findings. First, we verify our theoretical predictions on deep diagonal linear networks. Next, we show how our predictions hold in practical settings such as reparameterized sparse training and finetuning of vision and language models. In Appendix H, we study the natural invariance extension of Definition 4.6 for matrices and ablate the matrix product formed by the $Q$ query and $K$ key matrices in attention (as mentioned in Example 3.2) for a family of LLama models (Grattafiori et al., 2024). In practice, gradient flow is implemented as gradient descent with small learning rate (i.e. $\eta = 0.0001$ in Fig. 1 and $\eta = 0.01$ in Fig. 4).

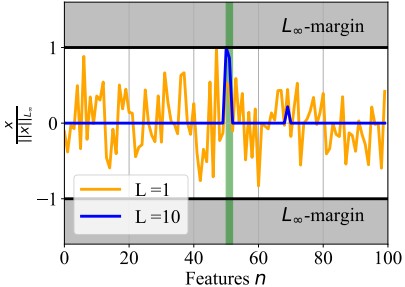

Figure 4: The $L_\infty-$margin for Adam with high and low depth $L$. The green region indicates the non-zero ground truth features. Higher depth leads to sparse ground truth recovery in line with Corollary 4.13.

**Diagonal linear network** In line with our theory, we consider a diagonal deep network $x = \Pi_{i=1}^{L} w_i$ for regression and binary classification with respect to the mean squared error or exponential loss, respectively. $x^*$ denotes the sparse ground truth. This setting corresponds to Theorem B.10 and Theorem B.12. Our initialization follows Corollary 4.10 for a small $\lambda$ close to the saddle point set $S$. For the experimental details, see Appendix I and F.

In Fig. 1, we first illustrate Theorem 4.2 by reporting the overdetermined setting for linear regression with $k = 300 > n$ samples, $n = 100$ features, and depth $L = 3$. With high probability this ensures the existence of a unique minimum, that is, strong convexity. We observe that it takes significant more time for gradient descent with small learning rate to escape the saddle point initialization and reach the global minimum. For higher depth, this effect is intensified, as can be seen in the ablations in Appendix I, where we also consider coupled versus decoupled weight decay to demonstrate Lemma 4.5 and study the effect of less data and small batch size in detail.

In the classification setting, we consider $k = 80$ samples and a sparse ground truth (see Appendix F). Fig. 4 shows how higher depth leads to sparse ground truth $L_\infty$-margin recovery. This is in line with Corollary 4.13 for SignGF ($\simeq$ Adam), where higher depth corresponds to a higher sparsity inducing Legendre function. This geometric bias was not covered before by max-margin results, as illustrated in Theorem F.1. Moreover, margins of SignGF and GF are compared in Appendix F.

**Finetuning scenario** Fig. 5(a) illustrates a mechanism by which Adam can outperform SGD in a fine-tuning vision task, despite SGD typically achieving better performance in vision pre-training scenarios. The top 50 eigenvalues of the Hessian spectrum were calculated with software from (Gol-

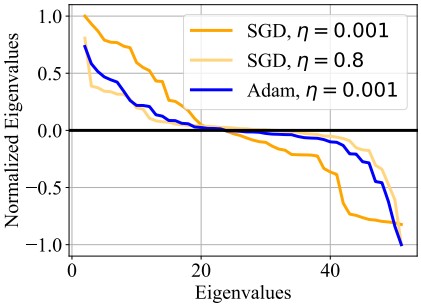 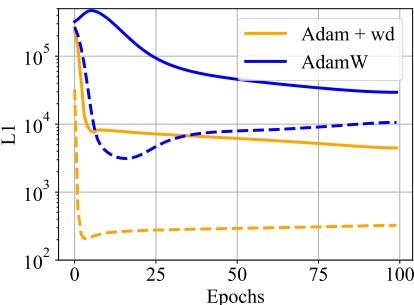

(a) Top 50 eigenvalues of Hessian at solution obtained by SGD and Adam after finetuning on CIFAR10. SGD with small learning rate has difficulty escaping the saddle point in contrast to Adam.

(b) $L_1$ norm of the weights during training for Adam with coupled weight decay strength $1e - 4$ and AdamW with $1e - 1$. The dashed lines correspond to depth $L = 10$ and solid lines to $L = 2$.

Figure 5: Eigenvalue spectra in finetuning for an ImageNet pretrained ResNet-18 on CIFAR-10 (a) and weight sparsity in reparameterized training for a ResNet-50 on Imagenet (b).

mant et al., 2018) for a ResNet-18 pretrained on ImageNet (Deng et al., 2009) after fine-tuning on CIFAR-10 (Krizhevsky, 2009). They highlight how far the optimizer has moved away from the initial saddle point. We observe that Adam exhibits fewer and weaker negative eigenvalues, indicating that it escapes saddle regions more effectively than SGD, while achieving higher performance. In Appendix K, additional ablations are provided, including additional experiments on different architectures (ViT-large, Bert-base) and datasets (Flowers (Nilsback & Zisserman, 2008), MRPC). The validation accuracy is reported in Table 2, which shows that Adam outperforms SGD for both small and (tuned) large learning rates. The specific learning rates are given in Appendix K.

Table 2: Validation accuracy for finetuning scenarios (95% confidence interval.).

| Model | Finetune | SGD (lr << 1) | SGD (lr > 0) | Adam (lr << 1) |
|---|---|---|---|---|
| ResNet-18 | CIFAR-10 | $19.15 \pm 2.82$ | $93.60 \pm 0.38$ | $\mathbf{95.19 \pm 0.21}$ |
| ResNet-18 | Flowers | $1.22 \pm 0.53$ | $62.13 \pm 1.10$ | $\mathbf{80.50 \pm 1.38}$ |
| ViT-large | CIFAR10 | $73.27 \pm 3.68$ | $99.07 \pm 0.35$ | $\mathbf{99.28 \pm 0.07}$ |
| ViT-large | Flowers | $1.03 \pm 0.82$ | $98.94 \pm 0.05$ | $\mathbf{99.37 \pm 0.08}$ |
| Bert-base | MRPC | $43.87 \pm 24.02$ | $84.80 \pm 1.00$ | $\mathbf{85.95 \pm 0.64}$ |

**Sparsification**  Next, we analyze how decoupled weight decay alters the sparsity bias in a reparameterized ResNet-50 trained on Imagenet. As shown in Figure 5(b), AdamW exhibits a sparsity-inducing effect only for very deep reparameterizations and sufficiently large weight decay, aligning with Table 1. The effects of weight decay strength and reparameterization depth are reported in Appendix J and the validation accuracy in Table 7.

## 6 DISCUSSION

We have studied training dynamics through a geometric lens that derives mirror flows for a family of steepest-descent optimizers, moving beyond gradient flow into a Banach space setting. This framework clarifies how optimizer geometry interacts with architectural choices (e.g., attention and reparameterizations). While our analysis applies to deep diagonal reparameterizations, we corroborate its relevance more broadly via fine-tuning experiments on LLM and vision tasks. The theory yields concrete, testable predictions that match practice: Compared to gradient flow GF ($\simeq$ SGD), sign gradient flow SignGF ($\simeq$ Adam) escapes saddles faster, is more stable at small learning rates, and behaves differently under decoupled weight decay, as inducing sparsity with decoupled decay requires deeper reparameterizations. These insights translate into actionable levers for efficient fine-tuning: Select optimizer geometry to control saddle escape and tune depth to target sparsity. We view this as a step toward co-design of optimizers and architectures, and a foundation for extending our analysis to non-diagonal models and discrete, stochastic training.

## ACKNOWLEDGEMENTS

The authors gratefully acknowledge the Gauss Centre for Supercomputing e.V. for funding this project by providing computing time on the GCS Supercomputer JUWELS at Jülich Supercomputing Centre (JSC). We also gratefully acknowledge funding from the European Research Council (ERC) under the Horizon Europe Framework Programme (HORIZON) for proposal number 101116395 SPARSE-ML.

## REPRODUCIBILITY STATEMENT

For the theory, detailed proofs have been provided for the main statements in Appendix E and used previously known statements have been provided in Appendix B and C. Additional derived statements are provided in Appendices D, F, and G. For the experiments, the details are provided in Appendices F, and I, J, and K.

## LLM STATEMENT

To improve fluency of the text sentence level editing has been done using large language models.

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

# A  EQUIVALENCE BETWEEN SIGNGD AND ADAM

We recall the optimization algorithms Adam (Kingma & Ba, 2017) and SignGD here to highlight their connection. Moreover the equivalence SignGD with coupled and decoupled weight decay is mentioned. We can set $\epsilon = 0$ and $\beta_1 = \beta_2 = 0$ in Algorithm 1, then we recover Algorithm 2. Similarly we recover the equivalence of AdamW (Loshchilov & Hutter, 2017) and SignGD with decoupled weight decay. Note that just setting $\epsilon = 0$ already gives us a sign like update as well. Note another related optimizer is LION which is sign gradient descent with momentum (Chen et al., 2023).

---

**Algorithm 1** Adam with Coupled ($\alpha_1$) and Decoupled ($\alpha_2$) Weight Decay

1: **Input:** parameters $x_0$, learning rate $\eta$, decay rates $\beta_1, \beta_2, \epsilon$ for stability, weight decay coefficients $\alpha_1, \alpha_2$
2: Initialize $m_0 \leftarrow 0$, $v_0 \leftarrow 0$, $t \leftarrow 0$
3: **while** not converged **do**
4:   $t \leftarrow t + 1$
5:   Compute gradient:

$$g_t \leftarrow \nabla_x f(x_{t-1}) + \alpha_1 x_{t-1}$$

6:   $m_t \leftarrow \beta_1 m_{t-1} + (1 - \beta_1) g_t$
7:   $v_t \leftarrow \beta_2 v_{t-1} + (1 - \beta_2) g_t^2$
8:   $\hat{m}_t \leftarrow m_t/(1 - \beta_1^t)$
9:   $\hat{v}_t \leftarrow v_t/(1 - \beta_2^t)$
10:  **Update rules:**
11:    *Coupled (Adam + $\alpha_1$):*

$$x_t \leftarrow x_{t-1} - \eta \frac{\hat{m}_t}{\sqrt{\hat{v}_t} + \epsilon}$$

12:    *Decoupled (AdamW + $\alpha_2$):*

$$x_t \leftarrow x_{t-1} - \eta \frac{\hat{m}_t}{\sqrt{\hat{v}_t} + \epsilon} - \eta \alpha_2 x_{t-1}$$

13: **end while**

---

**Algorithm 2** SignGD with Coupled ($\alpha_1$) and Decoupled ($\alpha_2$) Weight Decay

1: **Input:** parameters $x_0$, learning rate $\eta$, weight decay coefficients $\alpha_1, \alpha_2$
2: $t \leftarrow 0$
3: **while** not converged **do**
4:   $t \leftarrow t + 1$
5:   Compute gradient (with coupled $\alpha_1$):

$$g_t \leftarrow \nabla_x f(x_{t-1}) + \alpha_1 x_{t-1}$$

6:   **Update rules:**
7:    *Coupled (SignSGD + $\alpha_1$):*

$$x_t \leftarrow x_{t-1} - \eta \operatorname{sign}(g_t)$$

8:    *Decoupled (SignSGD + $\alpha_2$):*

$$x_t \leftarrow x_{t-1} - \eta \operatorname{sign}(\nabla_x f(x_{t-1})) - \eta \alpha_2 x_{t-1}$$

9: **end while**

---

## B    CONVEX ANALYSIS, LINEAR REGRESSION, AND CLASSIFICATION

In this section we recall definitions from convex analysis and known results from the implicit bias literature.

**Convexity an PL inequality**    For convergence to a minimizer the objective function needs to satisfy some condition. Two common ones are convexity and the PL-inequality. Note that strong convexity implies both.

**Definition B.1** (Convex Function). A function $f : \mathbb{R}^n \to \mathbb{R}$ is convex if for all $x, y \in \mathbb{R}^n$ and $\theta \in [0, 1]$,
$$f(\theta x + (1 - \theta)y) \leq \theta f(x) + (1 - \theta)f(y).$$

**Definition B.2** (Polyak–Łojasiewicz (PL) Condition). A differentiable function $f$ satisfies the *PL condition* with parameter $\Lambda > 0$ if

$$\frac{1}{2}\|\nabla_x f(x)\|_2^2 \geq \Lambda\big(f(x) - f^\star\big) \quad \text{for all } x,$$

where $f^\star = \inf_x f(x)$.

**Steepest descent**    The family of steepest descent algorithms generalizes classical gradient descent to arbitrary normed optimization geometries. We consider the same setting as in (Tsilivis et al., 2025). Given a norm $\|\cdot\|$ with dual norm $\|\cdot\|_\star$, the steepest descent update for loss $f(x)$ is defined as

$$x_{t+1} = x_t + \eta_t \Delta x_t, \quad \text{where } \Delta x_t = \arg\min_{\|u\| \leq \|\nabla_x f(x_t)\|_\star} \langle u, \nabla_x f(x_t)\rangle. \tag{7}$$

When $\|\cdot\| = \|\cdot\|_2$, this reduces to the familiar gradient descent method. More generally, the steepest flow in continuous time is given by

$$\frac{dx}{dt} \in \left\{ \arg\min_{\|u\| \leq \|g_t\|_\star} \langle u, g_t\rangle : g_t \in \partial f(x_t) \right\}, \tag{8}$$

where $\partial g(\theta_t)$ denotes Clarke's subdifferential (Definition B.4) to allow for non-differentiable activations such as ReLU. For the $L_p$ norm this reduces to:

$$dx_t = -\text{sign}(\nabla_x f(x_t)) \odot |\nabla_x f(x_t)|^{q-1} \|\nabla_x f(x_t)\|_{L_q}^{2-q} dt \qquad x_0 = x_{\text{init}},$$

where $q$ satisfies $\frac{1}{p} + \frac{1}{q} = 1$. Now define a time rescaling $\tau = \int_0^t \|\nabla_x f(x_s)\|_{L_q}^{2-q} ds$ giving:

$$dx_\tau = -\text{sign}(\nabla_x f(x_\tau)) \odot |\nabla_x f(x_\tau)|^{q-1} d\tau \qquad x_0 = x_{\text{init}}.$$

This recovers the flow investigated in the main text.

**Differential inclusion**    In order to study these flows we need to introduce what a Clarke subdifferential is and a differential inclusion. This is needed as the flow can not be interpreted in the classic sense where there exists a unique solution. Instead we can use a set valued interpretation.

**Definition B.3** (Differential Inclusion). A *differential inclusion* is a generalized ODE:

$$\frac{dx_t}{dt} \in F(x_t), \qquad t \geq 0,$$

where $F : \mathbb{R}^n \rightrightarrows \mathbb{R}^n$ is set-valued.

**Definition B.4** (Clarke Subdifferential). For a locally Lipschitz function $f : \mathbb{R}^n \to \mathbb{R}$, the *Clarke subdifferential* at $x$ is

$$\partial^\circ f(x) = \text{conv}\left\{ \lim_{k\to\infty} \nabla_x f(x_k) : x_k \to x, \ f \text{ differentiable at } x_k \right\}.$$

*Remark B.5.* Gradient flows for nonsmooth convex functions can be written as $\dot{x}(t) \in -\partial f(x(t))$ (using the convex subdifferential), and more generally for Lipschitz functions using the Clarke subdifferential.

*Remark* B.6 (Clarke subdifferential viewpoint on sign descent). Let $g(u) = \|u\|_1$. Its Clarke subdifferential is

$$\partial^\circ g(u) = \big\{ s \in \mathbb{R}^n : s_i = \text{sign}(u_i) \text{ if } u_i \neq 0, \ s_i \in [-1,1] \text{ if } u_i = 0 \big\}.$$

Hence, for any differentiable $f$, the *set-valued sign map* satisfies

$$\text{sign}\big(\nabla_x f(x)\big) = \partial^\circ \| \nabla_x f(x) \|_1.$$

Consequently, the sign gradient flow can be written as the differential inclusion

$$\frac{dx_t}{dt} \in -\partial^\circ \big\| \nabla_x f\big(x_t\big) \big\|_1,$$

which is well-posed in the sense of Filippov for locally Lipschitz right-hand sides (Clarke, 1990; Filippov, 1988).

To avoid notation overload, we will use the classical notation for steepest descent and write:

$$dx_t = -\text{sign}(\nabla_x f(x_t)) \odot |\nabla_x f(x_t)|^{q-1} dt, \qquad x_0 = x_{\text{init}}.$$

**Mirror flow**   A mirror flow can be defined in the classical sense:

$$d\nabla_x R(x_t) = -\nabla_x f(x_t) dt, \qquad x_0 = x_{\text{init}}. \tag{9}$$

where $R$ is a Legendre function (Defintion B.7). The overparameterization in deep linear networks can be interpreted as mirror flow as we will see in Appendix C.

**Definition B.7.** (Legendre function Definition 3.8 ((Li et al., 2022))) Let $R : \mathbb{R}^n \to \mathbb{R} \cup \{\infty\}$ be a differentiable convex function. We say $R$ is a Legendre function when the following holds:

- $R$ is strictly convex on $\text{int}(\text{dom}R)$.

- For any sequence $\{x_i\}_{i=1}^\infty$ going to the boundary of $\text{dom}R$, $\lim_{i \to \infty} \|\nabla_x R(x_i)\|_{L_2}^2 = \infty$.

For convergence of the itterates of the mirror flow as in Theorem 4.14 of (Li et al., 2022) the function $R$ also needs to be a Bregman divergence and function, which we define in Definitions B.8 and B.9.

**Definition B.8.** A Bregman divergence for a generator function $R : \mathbb{R}^n \to \mathbb{R}$ is defined for two points $x_1, x_2 \in \text{dom}R$:

$$D_R(x_1, x_2) = R(x_1) - R(x_2) - \langle \nabla_x R(x_2), x_1 - x_2 \rangle$$

**Definition B.9.** (Bregman function Definition 4.1 (Alvarez et al., 2004)) A function $R$ is called a Bregman function if it satisfies the following properties:

- $\text{dom}R$ is closed. $R$ is strictly convex and continuous on $\text{dom}R$. $R$ is $C^1$ on $\text{int}(\text{dom}R)$.

- For any $x \in \text{dom}R$ and $\gamma \in \mathbb{R}$, $\{y \in \text{dom}R | D_R(x,y) \leq \gamma\}$ is bounded.

- For any $x \in \text{dom}R$ and sequence $\{x_i\}_{i=1}^\infty \subset \text{int}(\text{dom}R)$ such that $\lim_{i \to \infty} x_i = x$, it holds that $\lim_{i \to \infty} D_R(x, x_i) \to 0$.

**Implicit bias in linear regression**   We recall a known result for the linear regression setup as also highlighted in Theorem B.10. We denote the data matrix with $Z$ and outputs with $Y$. This includes, gradient flow, sign gradient flow and mirror flow. Note the mirror flow case covers the gradient flow case as it corresponds to $R(x) := \frac{1}{2}\|x\|_{L_2}^2$.

**Theorem B.10** (Implicit bias of gradient and mirror flow). *(Gunasekar et al., 2018) Let $R$ be a Legendre function and initialize $x_0 = x_{init}$. Assume that the set $\{x \in \text{dom}R : Zx = Y\}$ is nonempty and that $f : \mathbb{R}^n \to \mathbb{R}$ is convex and or satisfies the PL-inequality. Among interpolants, the mirror-flow limit (when it exists) minimizes Bregman divergence to $x_{init}$:*

$$x^\star = argmin D_R(x, x_{init}) \text{ such that } Zx = Y.$$

*Remark* B.11. As shown in (Gunasekar et al., 2017) steepest descent algorithms do not nessecary allows a similar characterization for linear regression as in Theorem B.10.

**Implicit bias for classification** For steepest descent there is a recent result on seperable data for binary classification (Tsilivis et al., 2025). Similarly a result for general mirror flow exists (Pesme et al., 2024), not steepest mirror flows. We focus on the steepest descent result here as this includes our steepest descent reparameterization as well (it is a homogeneous network). By exploiting invariances, we can show that the margin has to satisfy additional constraints for deep diagonal networks. Their analysis relies on the following assumptions, which are satisfied by many practical neural network architectures and our reparameterization:

1. **Local Lipschitzness:** For any $z_i \in \mathbb{R}^d$, the mapping $x \mapsto f(z_i; x)$ is locally Lipschitz.

2. $L$**-Homogeneity:** The network $f$ is homogeneous of degree $L$, i.e. $f(\cdot; cx) = c^L f(\cdot; x)$ for any $c > 0$.

3. **Realizability:** There exists $t_0 > 0$ such that $L(x_{t_0}) < 1$, ensuring that perfect training accuracy is eventually achieved.

We now recall the main result of the paper regarding the implicit bias of steepest descent.

**Theorem B.12** (Convergence to KKT Points (Tsilivis et al., 2025, Theorem 3.4)). *Under assumptions (1)–(3), consider steepest flow with respect to a norm $\|\cdot\|$ on the exponential loss*

$$L(x) = \sum_{i \in [m]} e^{-y_i f(z_i; x)}.$$

*Then, any limit point $\bar{x}$ of the normalized trajectory $\left\{\frac{x_t}{\|x_t\|}\right\}_{t \geq 0}$ lies in the direction of a Karush–Kuhn–Tucker (KKT) point of the margin maximization problem*

$$\min_{x \in \mathbb{R}^p} \tfrac{1}{2} \|x\|^2 \quad s.t. \quad y_i f(z_i; x) \geq 1, \ \forall i \in [m]. \tag{10}$$

This theorem establishes that steepest descent algorithms implicitly bias the solution towards maximizing a geometry-dependent margin.

## C  REPARAMETERIZATIONS AS MIRROR FLOW

This section recaps the general results for reparameterizations and mirror flows and is based on Appendix A in (Jacobs et al., 2025b). For gradient flow we present the existing results for the mirror flow framework and time varying mirror flow framework. Consider an objective function $f : \mathbb{R}^n \to \mathbb{R}$

$$\min_{x \in \mathbb{R}^n} f(x).$$

We can use the implicit bias framework to study the effect of overparameterization. An overparameterization can be accomplished by introducing a function $g : M \to \mathbb{R}^n$, with $M$ a smooth manifold. For particular $g$, the reparameterization of the loss function $f$ leads to a mirror flow. The general framework is given in (Li et al., 2022) and extended in (Jacobs et al., 2025b) to study the implicit bias in terms of a mirror flow. (Li et al., 2022) provide a sufficient condition for the reparameterization $g$ such that it induces a mirror flow Eq. (9). The Legendre function $R$, see Definition B.7, controls the implicit bias and steers the trajectory of the dynamics.

In order to recover the convergence result in Theorem 4.14 in (Li et al., 2022) the function $R$ also needs to be a Bregman function, which is defined in Definition B.9. For a reparameterization to induce a mirror flow with a corresponding Legendre function we first have to give two definitions. Furthermore, we define $\partial g$ as the Jacobian of the function $g$.

**Definition C.1.** (Regular Parmeterization Definition 3.4 (Li et al., 2022)) Let $M$ be a smooth submanifold of $\mathbb{R}^D$. A regular parameterization $g : M \to \mathbb{R}^n$ is a $C^1$ parameterization such that $\partial G(w)$ is of rank $n$ for all $w \in M$.

For the second definition, we first need to define what a Lie bracket is.

**Definition C.2.** (Lie bracket Definition 3.4 (Li et al., 2022)) Let $M$ be a smooth submanifold of $\mathbb{R}^D$. Given two $C^1$ vector fields $X, Y$ on $M$, we define the Lie Bracket of $X$ and $Y$ as $[X, Y](w) := \partial Y(w)X(w) - \partial X(w)Y(w)$.

**Definition C.3.** (Commuting Parameterization Definition 4.1 (Li et al., 2022)) Let $M$ be a smooth submanifold of $\mathbb{R}^D$. A $C^2$ parameterization $g : M \to \mathbb{R}^d$ is commuting in a subset $S \subset M$ iff for any $i, j \in [n]$, the Lie bracket $\left[\nabla_w g_i, \nabla_w g_j\right](w) = 0$ for all $w \in S$. Moreover, we call $g$ a commuting parameterization if it is commuting in the entire $M$.

One additional assumption is need ed on the flow of the solution. We define the solution of the gradient (descent) flow of a function $f : M \to \mathbb{R}^n$ initialized at $x \in M$

$$dx_t = -\nabla_x f(x_t)dt \qquad x_0 = x \tag{11}$$

as $x_t = \phi_x^t(x)$ which is well defined if the solution exists. Using this we can make the following assumption.

**Assumption C.4.** (Assumption 3.5 (Li et al., 2022)) Let $M$ be a smooth submanifold of $\mathbb{R}^D$ and $g : M \to \mathbb{R}^n$ be a reparameterization. We assume that for any $w \in M$ and $i \in [n]$, $\phi_{g_i}^t(w)$ is well-defined for $t \in (T_-, T_+)$ such that either $\lim_{t \to T_+} ||\phi_{g_i}^t(w)||_{L_2} = \infty$ or $T_+ = \infty$ and similarly for $T_-$. Also, we assume that for any $w \in M$ and $i, j \in [n]$, it holds that for $(t, s) \in \mathbb{R}^2$ that $\phi_{g_i}^s \circ \phi_{g_j}^t(w)$ is well-defined iff $\phi_{g_j}^t \circ \phi_{g_i}^s(w)$

Using these definitions we state the known result for mirror flow.

**Theorem C.5.** *(Theorem 4.9 (Li et al., 2022)) Let $M$ be a smooth submanifold of $\mathbb{R}^D$ and $g : M \to \mathbb{R}^n$ be a commuting and regular parameterization satisfying Assumption C.4. For any initialization $w_{init} \in M$, consider the gradient flow for any objective $f : \mathbb{R}^n \to \mathbb{R}$:*

$$dw_t = -\nabla_w f(g(w_t))dt, \qquad w_0 = w_{init}.$$

*Define $x_t = g(w_t)$ for all $t \geq 0$, then the dynamics of $x_t$ is a mirror flow with respect to the Legendre function $R$ given by Lemma 4.8 in (Li et al., 2022), i.e.,*

$$d\nabla_x R(x_t) = -\nabla_x f(x_t)dt, \qquad x_0 = g(w_{init}).$$

*Moreover, this $R$ only depends on the initialization $w_{init}$ and the reparameterization $g$, and is independent of the loss function $L_t$.*

**Explicit regularization**   The above framework got extended recently in (Jacobs et al., 2025b) including explicit regularization. Consider the optimization problem:

$$\min_{w \in M} f(g(w)) + \alpha h(w).$$

Then the dynamics becomes a time varying mirror flow as described in Theorem C.6.

**Theorem C.6.** *Let $(g, h)\colon M \to \mathbb{R}^{n+1}$ be regular and commuting reparameterization satisfying Assumption C.4. Then there exists a time-dependent Legendre function $R_a$ such that*

$$d\nabla_x R_{a_t}(x_t) = -\nabla_x f(x_t)dt, \qquad x_0 = g(w_{init}), \tag{12}$$

*where $a_t = -\int_0^t \alpha_s ds$. Moreover, $R_{a_t}$ only depends on the initialization $w_{init}$ and the reparameterization $g$ and regularization $h$, and is independent of the loss function $f$.*

The deep diagonal linear reparameterizations do not satisfy a time varying steep mirror flow as shown in Corollary E.6.

## D    DEEP DIAGONAL LINEAR REPARAMETERIZATIONS AND SADDLE POINTS

We characterize the saddle points induced by the deep diagonal linear reparameterization. For this we first define what a saddle points is in Definition D.1.

**Definition D.1.** A saddle point $x \in \mathbb{R}^n$ of an objective function $f \in C^2(\mathbb{R}^n, \mathbb{R})$ is characterized by:

$$\nabla_x f(x) = 0 \text{ and } \nabla_x^2 f(x) \not\succeq 0$$

i.e. it is a critical point while the Hessian is not positive semidefinite.

Consider the product of parameters, $w_1, \ldots, w_L \in \mathbb{R}^n$ as in the main text. Then the loss landscape of an objective function $f(x)$ with $x = \Pi_{i=1}^{L} w_i$ has additional saddle points as characterized by the set $S$ in Theorem D.2.

**Theorem D.2.** *Assume that $\nabla_x f(0) \neq 0$. Then, in addition to the saddle points of $f$, the deep diagonal reparameterization $x = g(w) = \Pi_{i=1}^{L} w_i$ introduces saddle points at:*

$$S := \left\{ (w_1, \ldots, w_L) : \forall_{i,j \in [n]}, w_i = w_j = 0, \ w_k \neq 0 \text{ for } k \neq i, j \text{ and } i \neq j \right\}.$$

First we calculate the resulting gradient and Hessian using the chain rule:

$$\nabla_w f(x) = \left( \sum_{i \in [L]} \Pi_{j \neq i} w_j \right) \nabla_x f(x).$$

This implies that at least two $w_i = 0$ to induce a critical point. Assume now that exactly two are indeed zero, then for the Hessian term depending $\nabla_x^2 f$ does not contribute and we get

$$\nabla_w^2 f(0) = \nabla_x f(0) \otimes H_x$$

where $H_x$ is Hessian of $x = \Pi_{i=1}^{L} w_i$ i.e. block matrices for every coordinate of $x$.

Every block matrix has two nonzero entries i.e. we have:

$$H_{x,k,m} := \begin{cases} \Pi_{\ell \neq i,j} w_\ell & \text{if } (k,m) = (i,j) \text{ or } (j,i) \\ 0 & \text{else} \end{cases}$$

This matrix is indefinite with eigenvalues $\pm \sqrt{\Pi_{\ell \neq i,j} w_\ell}$. Since $\nabla_x f(0) \neq 0$ there is at least one negative eigen value. $\square$

Theorem D.2 highlights that if already one coordinate vector $w_i$ for $i \in [L]$ is zero, the model is already close to a saddle point. This highlights that for the $\lambda-$balance, for small $\lambda$, we are very close to a saddle point.

# E    MAIN RESULTS: STEEP MIRROR FLOW AND INVARIANCE

We provide proofs here for the main results in the main text. The correspondence is:

- Theorem E.1 is Theorem 4.2.
- Lemma E.2 is Lemma 4.5.
- Theorem E.3 is Theorem 4.8.
- Corollary E.4 is Corollary 4.13.
- Theorem E.5 is Theorem 4.16.

**Theorem E.1.** *Let $R \in C^2(\mathbb{R}^n, \mathbb{R})$ be a separable function (Definition B.9) that is inversely $\mu$-coercive (Definition 4.1). Moreover, assume that the set $\{x \in \text{Dom } R : \min f(x)\}$ is non-empty and there exists a constant $B > 0$ such that for all $t > 0$, $|\partial_i f(x_t)| \leq B$ for all $i \in [n]$. Then the loss decays and satisfies:*

$$\int_0^\infty ||\nabla_x f(x_t)||_{L_2}^2 dt \leq (f(x_0) - f(x_\infty)) / (\mu B^{2-q}) .$$

*Assume that $f \in C^1(\mathbb{R}^n, \mathbb{R})$ is strongly convex. Then for the iterates of Eq. (4) converges such that we have $\lim_{t \to \infty} x_t = x^*$ where $x^*$ is the unique minimizer of $f$ with linear rate $\mu B^{q-2}\Lambda$.*

Proof. The proof follows from tracking the evolution of the loss $f$ and the observation that for strongly convex functions the sign is only zero when the minimum is reached.

First note the loss is decreasing:

$$df(x_t) = -\langle \nabla_x f(x_t), \nabla_x^2 R^{-1}(x_t) \text{sign}(\nabla_x f(x_t))|\nabla_x f(x_t)|^{q-1}\rangle dt$$
$$\leq -\mu ||\nabla_x f(x_t)||_{L_q}^q dt$$
$$\leq 0$$

where we used that $\nabla_x^2 R^{-1}$ is $\mu-$coercive and that it is separable. Rewriting the above equation gives us:

$$\int_0^\infty ||\nabla_x f(x_t)||_{L_q}^q \leq (f(x_0) - f(x_\infty)) / \mu < \infty.$$

This resembles the classic sufficient descent lemma for $L-$smooth functions. Moreover we have that:

$$\int_0^\infty ||\nabla_x f(x_t)||_{L_2}^2 dt \leq \int_0^\infty B^{2-q}||\nabla_x f(x_t)||_{L_q}^q dt$$

implying that

$$\int_0^\infty ||\nabla_x f(x_t)||_{L_2}^2 dt \leq (f(x_0) - f(x_\infty)) / (\mu B^{2-q})$$

Note that if $f$ is strongly convex then it satisfies the PL-inequality and we have:

$$df(x_t) = -\langle \nabla_x f(x_t), \nabla_x^2 R^{-1}(x_t) \text{sign}(\nabla_x f(x_t))|\nabla_x f(x_t)|^{q-1}\rangle dt$$
$$\leq -\mu ||\nabla_x f(x_t)||_{L_q}^q dt$$
$$\leq -\mu B^{q-2}||\nabla_x f(x_t)||_{L_2}^2 dt$$
$$\leq -\mu B^{q-2}\Lambda (f(x_t) - f(x^*)) dt$$

where we use the bounded gradients and the fact that $y^q \geq B^{q-2}y^2$ for for $y \in \mathbb{R}^+$. Then by Grönwall Lemma we have that:

$$f(x_t) - f(x^*) \leq (f(x_0) - f(x^*)) \exp\left(-t\mu B^{2-q}\Lambda\right),$$

recovering linear convergence depending on $\mu$ and $\Lambda$. We now can use that for $\Lambda$-strongly convex functions we have for all $x \in \mathbb{R}^n$ and the unique minimizer $x^*$:

$$||x - x^*||_{L_2}^2 \leq \Lambda (f(x) - f(x^*)),$$

using this we also have:

$$||x_t - x^*||_{L_2}^2 \leq \Lambda \exp\left(-t\mu B^{q-2}\Lambda\right)$$

This concludes the proof. $\square$

**Lemma E.2.** *Consider steepest descent with respect to $L_p$ and weight decay, with $\frac{1}{p} + \frac{1}{q} = 1$. Then, for a deep diagonal reparameterization, i.e., $x = g(w) = \Pi_{i=1}^{L}, w_i$ satisfies the following balance equation for $t \geq 0$ almost everywhere:*

$$|w_{i,t}|^q - |w_{j,t}|^q = (|w_{i,0}|^q - |w_{j,0}|^q) \exp\left(-q \int_0^t \alpha_s ds\right) \text{ for all } i,j \in [L]. \quad (13)$$

Proof. This can be checked by deriving the flow of the left hand side:

$$d\left(|w_{i,t}|^q - |w_{j,t}|^q\right) = q \, \text{sign}(w_{i,t})|w_{i,t}|^{q-1}dw_{i,t} - q \, \text{sign}(w_{j,t})|w_{j,t}|^{q-1}dw_{j,t}$$
$$= -\text{sign}(w_{i,t})|w_{i,t}|^{q-1}\text{sign}\left(\nabla_{w_i} f(x_t)\right)|\nabla_{w_i} f(x_t)|^{q-1}dt$$
$$+ \; \text{sign}(w_{j,t})|w_{j,t}|^{q-1}\text{sign}\left(\nabla_{w_j} f(x_t)\right)|\nabla_{w_j} f(x_t)|^{q-1}dt$$
$$- \; q \, \alpha_t \left(|w_{i,t}|^q - |w_{j,t}|^q\right) dt$$

It remains to be shown that the first terms cancel out. We can use the decompositions of signs and absolute values i.e. $\text{sign}\,(ab) = \text{sign}\,(a)\,\text{sign}\,(b)$ and $|ab| = |a||b|$. Using this we get for all $i \in [L]$:

$$\text{sign}(w_{i,t})|w_{i,t}|^{q-1}\text{sign}\left(\nabla_{w_i} f(x_t)\right)|\nabla_{w_i} f(x_t)|^{q-1} =$$
$$\text{sign}(w_{i,t})\text{sign}\left(\Pi_{j\in[L]\setminus\{i\}} w_{j,t}\right)|\Pi_{j\in[L]\setminus\{i\}} w_{j,t}|^{q-1}\text{sign}\left(\nabla_x f(x_t)\right) =$$
$$\text{sign}(x_t)|x_t|^{q-1}\text{sign}\left(\nabla_x f(x_t)\right)|\nabla_x f(x_t)|^{q-1},$$

which holds for all absolutely continuous solutions $w_t$. Therefore, we have that the evolution is given by:

$$d\left(|w_{i,t}|^q - |w_{j,t}|^q\right) = -q \, \alpha_t \left(|w_{i,t}|^q - |w_{j,t}|^q\right) dt.$$

This is linear ODE of the form $dz_t = -q \, \alpha_t z_t dt$ which has solution $z_t = z_0 \exp\left(-q \int_0^t \alpha_s ds\right)$. Plugging in $z_t := |w_{i,t}|^q - |w_{j,t}|^q$ yields the result. Note that this result has to be interpreted in the Filippov sense i.e. for all absolutely continuous solutions this holds almost everywhere. $\square$

**Theorem E.3.** *Initialize a deep diagonal reparameterization such that it is $\lambda - L_p$-balanced for a $\lambda \geq 0$ with respect to the first parameter $w_1$. Then, steepest descent satisfies a separable $L_p$-mirror flow almost everywhere:*

$$d\nabla_x R_{L_p,L}(x_t) = -\text{sign}\left(\nabla_x f(x_t)\right) \odot |\nabla_x f(x_t)|^{q-1} dt, \qquad x_0 = x_{init},$$

*where $\nabla_x R_{L_p,L}(x)$ is a seperable Bregman function completely characterized by the balances of Lemma 4.5. For $L = 2$, we explicitly get*

$$\nabla_x^2 R_{L_p,2}(x) := \frac{1}{\sqrt{4|x|^q + \lambda^2}}.$$

Proof. First we derive an expression for the metric in terms of $w_i$ for $i \in [L]$. We then use Lemma 4.5 to characterize $\nabla_x^2 R^{-1}(x)$. From the chain rule and decomposition of signs and absolute values it follows that:

$$dx_t = -\left(\sum_{i\in[L]} |\Pi_{j\in[L]\setminus\{i\}} w_j|^q\right) \text{sign}\left(\nabla_x f(x_t)\right)|\nabla_x f(x_t)|^{q-1}dt.$$

Now using the invariance and balance assumption with respect to the first parameter $w_1$ that holds a.e.:

$$|w_{j,t}|^q - |w_{1,t}|^q = \lambda \text{ for all } j \in [L] \setminus \{1\},$$

we can express the inverse metric in terms of $|w_{1,t}|^q$ and $\lambda$:

$$\nabla_x^2 R^{-1}(x) = \text{diag}\left((|w_1|^q + \lambda)^{L-1} + (L-1)|w_1|^q (|w_1|^q + \lambda)^{L-2}\right) \quad (14)$$

This is a continuous differentiable function in $|w_1|^q$. Moreover, we have that:

$$|x|^q = |w_1|^q (|w_1|^q + \lambda)^{L-1}$$

By the implicit function theorem from calculus we know there exists a continuous function $w_1(x, \lambda)$ for all $x \in \mathbb{R}^n$ and $\lambda > 0$. For this we need to have that there exists a unique positive solution to the polynomial equation of the form:

$$|x|^q = z(z + \lambda)^{L-1},$$

where the left hand side is a non-negative constant. We can show that the right hand side is increasing for $z \geq 0$ implying a unique solution:

$$\frac{d}{dz}\left(z(z + \lambda)^{L-1}\right) = (z + \lambda)^{L-1} + (L - 1)z(z + \lambda)^{L-2} > 0$$

for $\lambda > 0$. Thus there is a unique solution. In case $\lambda = 0$ we have that

$$z = |x|^{\frac{q}{L}}.$$

Therefore in the case $\lambda > 0$ we can guarantee using the implicit function theorem that we can express $w_1$ in terms of $x$ and $\lambda$. Moreover, for $\lambda = 0$ an explicit expression is available. Plugging this into Eq. (14) yields the result.

For $L = 2$ we have that

$$|x|^q = |w_1|^q \left(|w_1|^q + \lambda\right).$$

This is a quadratic equation in terms of $|w_1|^q$. We need to select the sole nonnegative solution, giving:

$$|w_1|^q = \frac{-\lambda + \sqrt{\lambda^2 + 4|x|^q}}{2}.$$

We can plug this into $\nabla_x^2 R^{-1}(x)$ giving

$$\nabla_x^2 R^{-1}(x) = 2|w_1|^q + \lambda = \sqrt{\lambda^2 + |x|^q}.$$

This concludes the first part.

It remains to be shown that the implicit constructed mirror map is a separable Bregman function. We will use the connection between Legendre functions and Bregman functions to show this. We use that if the domain of a Legendre functions $R$ is $\mathbb{R}^n$ and its convex dual $R^*$ has this as its domain as well then $R$ is a Bregman function according to Theorem 4.7 in (Alvarez et al., 2004). Therefore, we need to show $R_{L_p,L}$ is a Legendre function and characterize the domains.

We first note that it separable by construction. This allows us to focus on the one dimensional case. By construction, we know that $\nabla_i^2 R_{L_p,L}^{-1}$ has domain $\mathbb{R}$ and range $[\lambda^{q(L-1)}, \infty)$. Therefore, $\nabla_i^2 R_{L_p,L}$ has domain $\mathbb{R}$ and range $(0, \lambda^{-q(L-1)}]$. This holds for all $i \in [n]$. This implies that $R$ is strictly convex and $C^2(\mathbb{R}^n, (0, \lambda^{-q(L-1)}]^n)$ proving the first condition of being a Legendre function. For the essential smooth condition, we can use the asymptotic behavior near the boundary of the domain of $\nabla_i^2 R_{L_p,L}$. This provides a lower bound on $|\nabla_i R_{L_p,L}|$. Concretely we use the triangle inequality and lower bound the growth of $\nabla_i^2 R_{L_p,L}$:

$$\begin{aligned}
|\nabla_i R_{L_p,L}(x)|^2 &= \left|\int^{x_i} \nabla_i^2 R_{L_p,L}(y)dy\right|^2 \\
&\geq \left(\int^{x_i} |\nabla_i^2 R_{L_p,L}(y)|dy\right)^2 \\
&\geq \left(\int^{x_i} |y|^{-q\frac{L-1}{L}}dy\right)^2 \\
&= \left(\frac{1}{1 - q\frac{L-1}{L}}\right)^2 |x_i|^{2 - 2q\frac{L-1}{L}}
\end{aligned}$$

The right hand side only diverges if and only if $q\frac{L-1}{L} \leq 1$. Hence $R_{L_p,L}$ is a Legendre function. In order to show $R_{L_p,L}$ is Bregman we use the following two observations. 1) The anti-derivative of an even function is odd 2) $\nabla_i^2 R_{L_p,L}^{-1}$ is an even function. It follows from 2) that also the reciprocal $\nabla_i^2 R_{L_p,L}$ is even. Now we integrate and this implies that $\nabla_i R_{L_p,L}$ is odd. Now using continuity and essential smoothness imply that the range of $\nabla_i R_{L_p,L}$ is $\mathbb{R}$. Therefore, the domain of the $\nabla_i R_{L_p,L}^*$ is $\mathbb{R}$. This implies $R_{L_p,L}^*$ has domain $\mathbb{R}^n$. Hence $R_{L_p,L}$ is a Bregman function accordingly. $\square$

**Lemma E.4.** *For $L \geq 2$ and $\lambda = 0$, candidates for the Legendre function are given by:*

- *if $m = q\frac{L-1}{L} = 1$:*

$$R_{L_p,L}(x) = \frac{1}{L} \sum_{j \in [n]} \left( x_j log(x_j) - x_j - x_j log(x_{j,0}) \right)$$

- *if $m = q\frac{L-1}{L} \neq 1$:*

$$R_{L_p,L}(x) = \frac{1}{L - (L-1)q} \sum_{j \in [n]} \left( \frac{|x_j|^{2-q\frac{L-1}{L}}}{\left(\frac{q}{L} - q + 2\right)} - x_j x_{j,0} |x_{j,0}|^{q\left(\frac{1}{L}-1\right)} \right).$$

*If $m = 1$, $R_{L_p,L}$ is a Legendre function with metric exponent $m$ on the domain $\mathbb{R}^{sign(x_{1,0})} \times \ldots \times \mathbb{R}^{sign(x_{n,0})}$. If $m < 1$, the domain is $\mathbb{R}^n$. Otherwise, $R_{L_p,L}$ is not a Legendre function.*

Proof. Plug in $\lambda = 0$ and calculate $w_1(x, 0)$. This gives an explicit expression for the inverse metric:

$$\nabla_x^2 R^{-1}(x) = L|x|^{q\frac{L-1}{L}}.$$

We now integrate the metric to get the Legendre function, to keep notation clean we omit the summing over $x_i \in [n]$ as the calculation is the same for all. Integrating the inverse twice and using that $\nabla_x R(x_0) = 0$ gives: If $q\frac{L-1}{L} = 1$ we have that

$$\int^x \int^u \nabla_x^2 R(v) dv du = \int^x \int^u \frac{1}{L|v|} dv du$$

$$= \frac{1}{L} \int^x log(u) - log(x_0) du$$

$$= \frac{1}{L} \left( x log(x) - x - x log(x_0) \right).$$

Moreover, if $q\frac{L-1}{L} \neq 1$ we have that:

$$\int^x \int^u \nabla_x^2 R(v) dv du = \int^x \int^u \frac{1}{L} |v|^{-q\frac{L-1}{L}} dv du$$

$$= \int^x -\frac{u|u|^{\frac{q}{L}-q}}{(L-1)q - L} + \frac{x_0|x_0|^{\frac{q}{L}-q}}{(L-1)q - L} du$$

$$= -\frac{|x|^{\frac{q}{L}-q+2}}{\left(\frac{q}{L} - q + 2\right)((L-1)q - L)} + x\frac{x_0|x_0|^{\frac{q}{L}-q}}{(L-1)q - L}$$

$$= \frac{1}{L - (L-1)q} \left( \frac{|x|^{q\left(\frac{1}{L}-1\right)+2}}{\left(\frac{q}{L} - q + 2\right)} - x x_0 |x_0|^{q\left(\frac{1}{L}-1\right)} \right)$$

This concludes the result. In order for $R_{L_p,L}$ to be strictly convex we need $q\frac{L-1}{L} < 1$ the other conditions to be Legendre function such as essentially smooth are then also satisfied. The domains follow from the derived Legendre function cases. $\square$

**Theorem E.5.** *Assume a) $m = q\frac{L-1}{L} \neq 2$ or b) $m = q\frac{L-1}{L} = 2$. The manifold regularizer for decoupled weight decay with $L_p$ steepest descent on the manifold for a reparameterization of depth $L$ with balanced initialization ($\lambda = 0$) is: a) $\frac{L}{L(2-q)+q} \sum_{i \in [n]} |x_i|^{2-q\frac{L-1}{L}}$ or b) $\sum_{i \in [n]} log(|x_i|)$.*

Proof. The regularization rebalances the balance equation leading to the balance with $\lambda = 0$. We can use Corollary 4.13 to derive the metric. A key difference now is that the regularization is still on so we have a dynamics of the form:

$$dx_t = -L|x_t|^{q\frac{L-1}{L}} \left( \text{sign}(\nabla_x f(x_t)) \odot |\nabla_x f(x_t)|^{q-1} \right) - Lx_t dt, \qquad x_0 = x_{\text{init}}.$$

This can be equivalently written as:

$$dx_t = -L|x_t|^{q\frac{L-1}{L}} \left( \text{sign}(\nabla_x f(x_t)) \odot |\nabla_x f(x_t)|^{q-1} + x_t |x_t|^{-q\frac{L-1}{L}} \right) dt, \qquad x_0 = x_{\text{init}}.$$

Similarly this can written as the mirror flow due to the equivalence of Riemannian GF and mirror flow:

$$d\nabla_x R_{L_p,L}(x_t) = -\left(\text{sign}(\nabla_x f(x_t)) \odot |\nabla_x f(x_t)|^{q-1} + x_t |x_t|^{-q\frac{L-1}{L}}\right) dt$$

Therefore, the on manifold regularization is the $M_{\text{reg}}(x)$:

$$M_{\text{reg}}(x) = \sum_{i \in [n]} \int^{x_i} |x_i|^{-q\frac{L-1}{L}} x_i dx_i = \begin{cases} \frac{L}{L(2-q)+q} \sum_{i \in [n]} |x_i|^{2-q\frac{L-1}{L}} & \text{if } q\frac{L-1}{L} \neq 2 \\ \sum_{i \in [n]} \log(|x_i|) & \text{if } q\frac{L-1}{L} = 2. \end{cases}$$

This concludes the result.$\square$

**Corollary E.6.** *Iff $q = 2$, weight decay is equal to the on manifold regularization $M_{reg}$ for $\lambda = 0$.*

Proof. Since $\lambda = 0$, the weight decay is given by

$$\frac{1}{2}||w||_{L_2}^2 = \frac{L}{2} \sum_{i \in [n]} |x_i|^{\frac{2}{L}}$$

We can match this with $M_{\text{reg}}(x)$. For this we need to have:

$$\frac{L}{2} = \frac{L}{L(2-q)+q} \Leftrightarrow L(2-q)+q = 2 \Leftrightarrow q(1-L) = 2(1-L)$$

which is true if and only if $q = 2$. $\square$

Corollary E.6 highlights that Theorem C.6 can not be extended directly to steeper flows. This is due to the fact that the possible limiting regularization $M_{\text{reg}}$ on the manifold mismatches with the weight decay i.e. $\lambda = 0$, so in the end of training the time-varying mirror flow has to break down. Furthermore, the result Theorem C.6 already breaks for $L > 2$ as mentioned in (Jacobs et al., 2025b).

## F  IMPLICIT BIAS OF STEEP MIRROR DESCENT FOR BINARY SEPARABLE CLASSIFICATION

We present a margin characterization for SignGF using a recent result from (Tsilivis et al., 2025). We observe that the margin should be independent of depth $L$. The margin now becomes dependent on maximum of $|x_\ell|^{\frac{2}{L}}$ but this is an increasing function with the magnitudes as input thus the maximum would not change. In other words, the margin does not see what happens at zero. However, our mirror flow analysis suggests that the movement speed of the parameters near initialization will influence the solution reached by slowing down movement near zero and accelerating it further away. This helps with sparse ground truth recovery.

**Theorem F.1.** *Consider a $\lambda$-balanced deep diagonal linear networks trained in the linear separable classification setting as in Theorem B.12 with sign descent then $\tilde{x}_t := \frac{x_t}{||x_t||_{L_\infty}}$ limit point lies in the direction of a KKT point of margin maximization problem:*

$$\min_{x \in \mathbb{R}^n} \max_{\ell \in [d]} |x_\ell|^{\frac{2}{L}} \text{ such that } y_j \langle x, z_i \rangle \geq 1 \text{ for all } i \in [k]$$

Proof.

It follows from Theorem B.12 that $\tilde{w}_t := \frac{w_t}{||w_t||_{L_\infty}}$ is in the direction of a KKT point:

$$\min_{w_1,\dots,w_L \in \mathbb{R}^n} \frac{1}{2} ||w_1, \dots, w_L||_{L_\infty}^2 \text{ such that } y_j \langle g(w), z_i \rangle \geq 1 \text{ for all } i \in [k]$$

where $g(w) = \Pi_{j=1}^L w_j$. In addition, we know the iterates $||w||_{L_\infty} \to \infty$. Combining this with Lemma 4.5 it follows that for all $i, j \in [L]$:

$$|\tilde{w}_{t,i}| - |\tilde{w}_{t,j}| = \frac{\lambda}{||w||_{L_\infty}} \to 0$$

These additional constraints reduce the optimization problem to:

$$\min_{w_1,\dots,w_L \in \mathbb{R}^n : \Pi_{j=1}^L w_j = x} \frac{1}{2} \max_{\ell \in [d]} |x_\ell|^{\frac{2}{L}} \text{ such that } y_j \langle x, z_i \rangle \geq 1 \text{ for all } i \in [k]$$

It is easy to show that $\tilde{x}_t = \frac{x_t}{||x_t||_{L_\infty}}$ satisfies the KKT conditions above as well by using that in the limit $||w||_{L_\infty} = \max |x|^{\frac{1}{L}}$ and $\Pi_{j=1}^L w_j = x$ we have that:

$$\lim_{t \to \infty} \tilde{x}_t := \lim_{t \to \infty} \frac{x_t}{||x_t||_{L_\infty}} = \lim_{t \to \infty} \frac{\Pi_{j=1}^L w_{j,t}}{||w_t||_{L_\infty}^L} = \lim_{t \to \infty} \Pi_{j=1}^L \tilde{w}_{j,t},$$

where the middle equality follows from the invariance relationship. This concludes the proof. $\square$

**Experimental illustration**  We conduct an experiment on binary classification with an exponential loss as described above. The main goal is to illustrate the effect of depth which would not have an influence according to Theorem F.1. However, our dynamics description would predict that higher depth will lead to a relative slow down near zero of the dynamics effectively creating a sparsity bias.

We generate a sparse ground truth $x^* = (1, 1, 0, \dots, 0) \in \mathbb{R}^{100}$ and $k = 80$ data samples from a random Gaussian such that $Z_{i,j} \sim N(0,1)$ with $i, j \in [100, 80]$. The labels are then determined by the classifier groundtruth i.e. $y_j := \text{sign}(z_j^T x^*)$. Then we initialize at zero with $\lambda = 0.1$. We train for 10000 steps with learning rate $\eta = 0.01$. The optimizers used are SignGD, GD and Adam.

We report the final margin in Figure 6. Observe that for higher depth the margin is much sparser than for low depth. This highlights a new implicit bias mechanism caused by depth, leading to feature learning. Note that for GD depth $L = 10$, did not converge, as expected. This explains the spiky nature of the $L_\infty$ margin.

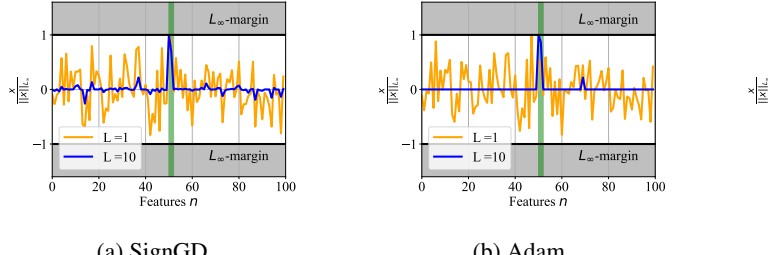

(a) SignGD             (b) Adam.             (c) GD.

Figure 6: Resulting $L_\infty$ margins for optimizers SignGD, Adam and GD, where the green strip indicates the contributing ground truth features. Observe the similarity between Adam and SignGD for all depth.

## G  SEPARABLE MIRROR REPARAMETERIZATION CONSTRUCTION

For completeness, we show how each separable steepest mirror flow can be seen as a reparameterization of steepest gradient flow. This is done by construction.

**Theorem G.1.** *Consider a one dimensional steepest mirror flow with Legendre function $R$ and is $\mu$-coercive. Then there exists a reparameterization $g : \mathbb{R} \to \mathbb{R}$ such that we have $x = g(w)$.*

Proof. We can show this by construction in the one dimensional case.

A valid invertible reparameterization is (using $\mu$-coercive):

$$z = \int^x \left(\partial^2 R(x)\right)^{\frac{1}{q}} dx,$$

to see this we can calculate the evolution of $z$:

$$dz_t = -\left(\partial^2 R(x)\right)^{\frac{1}{q}} dx_t = -\left(\partial^2 R(x)\right)^{\frac{1}{q}-1} \operatorname{sign}\left(\partial_x f(x_t)\right) |\partial_x f(x_t)|^{q-1} dt.$$

Now we use the implicit function theorem for the derivative of $f$ with respect to $z$:

$$\partial_z f(x) = \left(\partial^2 R(x)\right)^{-\frac{1}{q}} \partial_x f(x).$$

Plugging this in gives us:

$$dz_t = -\left(\partial^2 R(x)\right)^{\frac{1}{q}-1-\frac{q-1}{q}} \operatorname{sign}\left(\partial_z f(x_t)\right) |\partial_z f(x_t)|^{q-1} dt = -\operatorname{sign}\left(\partial_z f(x_t)\right) |\partial_z f(x_t)|^{q-1} dt.$$

Therefore $x$ can be seen as the inverse of $z$. Hence there exists a steep gradient flow with respect to the reparameterization $z^{-1}$ that corresponds to a chosen mirror flow by construction. □

*Remark* G.2. The proof in the one-dimensional case is quite simple as it is by construction. However, the proof in higher dimensions for standard mirror flow already relies on the Nash embedding theorem (Li et al., 2022) which is not constructive.

Table 3: Parameter sign flips per group type, overall, and average L1/L2 differences for LLaMA models. We also indicate with $< \%$ the percentage of the layers that have a negative delta

| Model | Q (%) | K (%) | Total (%) | Avg $\Delta_{L1}$ | $< \%, L_1$ | Avg $\Delta_{L2}$ | $< \%, L_2$ |
|---|---|---|---|---|---|---|---|
| LLaMA-3.1 8B | 1.25 | 0.87 | 1.57 | -624.13 | 100 | -20.28 | 100 |
| LLaMA-3.2 3B | 4.37 | 3.50 | 5.04 | -1757.04 | 100 | -101.71 | 100 |
| LLaMA-3.2 1B | 4.73 | 3.24 | 6.11 | -891.76 | 100 | -66.15 | 100 |

## H  INVARIANCE ISSUE FOR STEEPEST DESCENT FOR MATRIX INVARIANCES

The main hurdle for a more general balance equation to hold is that the sign operator does not distribute over matrices. In other words for two matrices $W_1$ and $W_2$ we do not have

$$\text{sign}\,(W_1 W_2) = \text{sign}\,(W_1)\,\text{sign}\,(W_2)$$

If this condition would hold plus the same condition with respect to the gradient then we would expect for a reparameterization $g(W_1, W_2) = W_1 W_2$ trained with a sign gradient flow the following to hold during training:

$$||W_{1,t}||_{L_1} - ||W_{2,t}||_{L_1} = (||W_{1,0}||_{L_1} - ||W_{2,0}||_{L_1}) \exp\left(-\int_0^t \alpha_s ds\right)$$

This would then hold instead of the balance equation for gradient flow:

$$||W_{1,t}||_{L_2}^2 - ||W_{2,t}||_{L_2}^2 = \left(||W_{1,0}||_{L_2}^2 - ||W_{2,0}||_{L_2}^2\right) \exp\left(-2\int_0^t \alpha_s ds\right),$$

which is known to hold for gradient flow. To see this, we compare for a family of LLama models the base version with their tuned instruct version. Their tuning (partially) has been done with AdamW. Even tough, sign flips occur during training, effectively ruining the balance for wider reparameterizations. We empirically observe that for finetuning a setting with small learning rate, less sign flips occur, making the insights from our example potentially relevant to larger scale finetuning. We track the direct generalization of the balance as in Definition 4.6 for the matix product of the $Q$ query and $K$ key matices in the attention mechanism:

$$\Delta_{L_p} := \left| ||Q_{\text{ft}}||_{L_q}^q - ||K_{\text{ft}}||_{L_q}^q \right| - \left| ||Q_{\text{pre}}||_{L_q}^q - ||K_{\text{pre}}||_{L_q}^q \right|.$$

In Table H we observe that indeed the $L_1$ balance is minimized more than the $L_2$ balance which is an indication that our balance result might be able to generalize to the fine tuning setting where AdamW is used. In addition, we observe for finetuning scenarios, that the signs of parameters change minimally. This we can capture by Definition H.1, which could lead to a bound on the invariance. However, this needs further assumptions on the nature of the gradients and how they evolve.

**Definition H.1.** Let $g : \mathbb{R}^{n \times m} \times \mathbb{R}^{m \times k} \to \mathbb{R}^{n \times k}$ be a reparameterization defined by $g(W_1, W_2) := W_1 W_2$. Then it is called sign stable during training if for $t \geq 0$,

$$\text{sign}(W_{\ell,t}) = \text{sign}(W_{\ell,0}) \qquad \text{for } \ell \in [2].$$

# I ADDITIONAL EXPERIMENTS ON DIAGONAL DEEP LINEAR NETWORKS

For linear regression with mean squared error we set the groundtruth to $(1, 1, 1, 1, 1, 0, \ldots, 0) \in \mathbb{R}^{100}$ and sample $Z_{i,j} \sim N(0, 1)$ for $i \in [100]$, $j \in [k]$. For our experiments we will train with steepest descent i.e. the discretization of Eq. (3) and train with learning rate $\eta = 1e - 4$ for $1e + 6$ steps. For our experiments in the main text we will set $w_0 = 0$ and $w_i = \lambda$ for $i \in [L] \setminus \{1\}$, with $\lambda = 0.1$. This ensures we start close to a saddle point as described in Appendix D. Moreover, we vary the parameters $q \in [1, 1.5, 2]$, $k \in [300, 80]$, $L \in [1, 2, 3, 10]$, and study the effect of coupled and decoupled weight decay.

First we consider the underdetermined case with $k = 80$, to illustrate the different implicit biases at each depth $L$. In Figure 7 we see that for high depth ($L = 10$) sign gradient descent recovers the sparse ground truth and gradient flow can not escape the saddle, which is in line with our dynamical description. Moreover, for $L = 2$, we see that gradient flow gets close to the ground truth which is in line with the implicit bias of the hyperbolic entropy see Example 3.2.

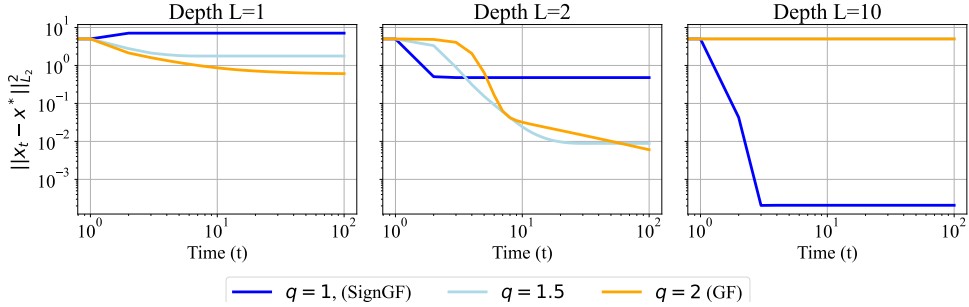

Figure 7: Underdetermined linear regression ($k = 80$), for depth $L = 1$ we do net get close to the ground truth in all cases, for $L = 2$ gradient flow gets close to the ground truth as in line with Theorem B.10 and in for higher depth $L = 10$ the sign gradient flow (SignGF) converges close to ground truth which we would expect based on the dynamic reformulation.

Next we observe in Figure 8 and 10 that smaller batch size is beneficial for feature learning when the depth $L$ plus steepest descent method $q$ leads close to an $L_1$ bias. Furthermore, in Figure 9 with less data, the implicit bias argument does not prevail and we do not observe feature learning. This highlight that there is no guarantee for feature learning. However, it seems to be possible to remedy it with smaller batch size.

Moreover, we conduct an additional experiment for sign gradient descent with coupled and decoupled weight decay of which the results are reported in Table 4. We use the same setting as described in the main text with $k = 80$ data samples and the same ground truth. We report the average $L_1$ distance to the theoretical predict balance value at the end of training which denote with Balance Distance. Observe that for coupled weight decay ($\alpha_2$) the distance increases while for decoupled weight decay ($\alpha_1$) we stay close to the theoretical predicted value. To add to this, high depth and decoupled regularization leads to recovering the ground truth the best.

**The benefit of noise** The benefit of noise for feature learning could be seen from re-purposing the majority voting interpretation in (Bernstein et al., 2019) where it is used for convergence guarantees. If a parameter needs to be zero to reach the ground truth and starts at zero, the gradient is potentially small, however, it still has a sign direction which might pull it away from the ground truth. Nevertheless, if we train with stochastic estimates we might be equally moved in either direction. This is captured by the following thought experiment, consider the gradient and stochastic gradient estimate:

$$\nabla f(x) = 0.01 \text{ and } g(x) = \begin{cases} -0.01 \text{ w.p. } \frac{1}{2} \\ 0.03 \text{ w.p. } \frac{1}{2} \end{cases}.$$

These estimators would have the same gradient expectation but the sign expectation is different i.e. we have

$$\text{sign}(\nabla f(x)) = 1 \text{ and } \mathbb{E}\left[\text{sign}(g(x))\right] = 0.$$

This indicates we need a stronger pull away from zero to actually move in the stochastic case. In other words, a larger majority of the gradients need to vote for a certain direction.

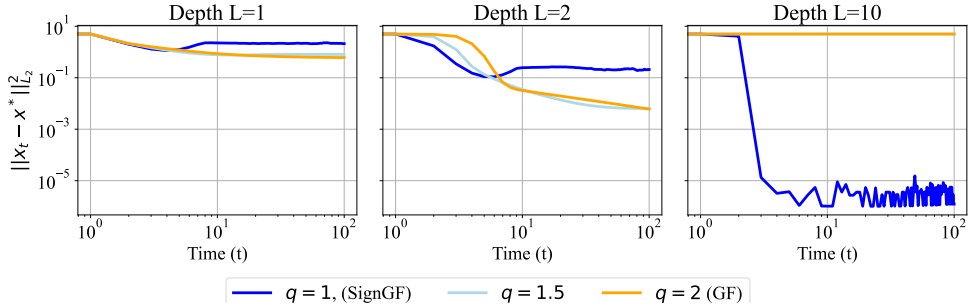

Figure 8: Recovering the ground truth with small batch size 5 for underdetermined regression with $k = 80$.

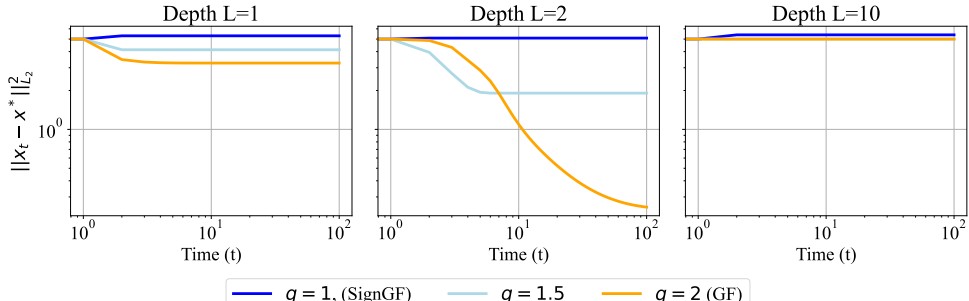

Figure 9: Recovering the ground truth with full batch for underdetermined regression with $k = 40$.

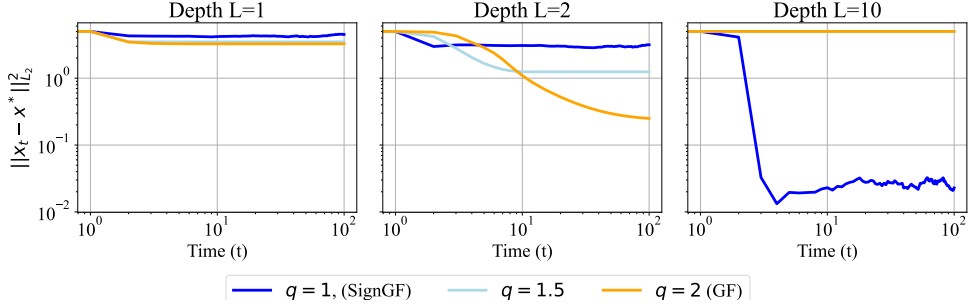

Figure 10: Recovering the ground truth with small batch size 5 for underdetermined regression with $k = 40$.

Table 4: Effect of Regularization Strengths on theoretical balance according to Lemma 4.5 and the distance the ground truth. For the decoupled weight decay ($\alpha_1$) indeed stays close the theoretical predicted balance.

| Depth $L$ | $\alpha_1$ | $\alpha_2$ | Balance Dist. | Groundtruth Dist. |
|---|---|---|---|---|
| 1 | 0 | 1e-4 | 0 | 7.1 |
| | 1e-4 | 0 | 0 | 7.0 |
| | 0 | 1e-3 | 0 | 7.1 |
| | 1e-3 | 0 | 0 | 5.8 |
| | 0 | 1e-2 | 0 | 7.1 |
| | 1e-2 | 0 | 0 | 1.0 |
| | 0 | 0 | 0 | 7.1 |
| 2 | 0 | 1e-4 | 5.3e-4 | 4.8e-1 |
| | 1e-4 | 0 | 7.3e-5 | 4.7e-1 |
| | 0 | 1e-3 | 5.1e-3 | 4.7e-1 |
| | 1e-3 | 0 | 1.8e-3 | 4.1e-1 |
| | 0 | 1e-2 | 3.5e-2 | 4.7e-1 |
| | 1e-2 | 0 | 6.7e-4 | 4.8e-1 |
| | 0 | 0 | 1.1e-4 | 4.8e-1 |
| 10 | 0 | 1e-4 | 1.2e-1 | 2.5e-4 |
| | 1e-4 | 0 | 1.5e-4 | 1.4e-4 |
| | 0 | 1e-3 | 3.9e-1 | 2.9e-4 |
| | 1e-3 | 0 | 2.9e-4 | 4.9e-5 |
| | 0 | 1e-2 | 7.8e-1 | 3.0e-3 |
| | 1e-2 | 0 | 1.6e-3 | 7.9e-6 |
| | 0 | 0 | 1.5e-4 | 2.1e-4 |

## J   SPARSITY EXPERIMENT

In this section we provide additional experiments for the reparameterized sparsity bias. Moreover, we provide additional experimental details in Table 5. The tunable parameters are depth $L \in \{2, 4, 10\}$ and weight decay strength $\alpha \in \{1e-1, 1e-4\}$. In the case for coupled weight decay we are effectively optimizing:

$$\min_{w_1,\ldots,w_L \in \mathbb{R}^n} f(\Pi_{i=1}^L w_i) + \alpha \sum_{i \in [L]} ||w_i||_{L_2}^2$$

or equivalently

$$\min_{x \in \mathbb{R}^n} f(x) + L\alpha \sum_{i \in [L]} ||x||_{L_{2/L}}^{2/L}$$

see Theorem 1 in (Kolb et al., 2025). The code used is based on Turboprune (Nelaturu et al.). The initialization of the depth 2 reparameterization is based on (Gadhikar et al., 2025) and for deeper reparameterizations we use the balancing equation to inform our initialization i.e. we use $w_1 = x$ and $w_i = 1$ for $i \neq 1$. This is closely related to the closed form formula for initialization of depth 2:

$$m_0 = \frac{v + \frac{\gamma}{v}}{\sqrt{2}} \text{ and } w_0 = \frac{v - \frac{\gamma}{v}}{\sqrt{2}}$$

where $v = \sqrt{x + \sqrt{x^2 + \gamma^2}}$ with $\gamma = \frac{1}{2}$. We can see this from a Taylor approximation around $x = 0$. Then we have $v \simeq \frac{1}{\sqrt{2}}\left(1 + x + \frac{x^2}{2}\right)$ and then $1/v \simeq \sqrt{2}\left(1 - x + \frac{x^2}{2}\right)$, putting this together give:

$$m_0 = 1 + \frac{x^2}{2} \text{ and } w_0 = x.$$

So when $x^2$ is negligible it matches our proposed initialization for deeper reparameterization.

In Figure 11 and 12, we show the $L_1$ norm during training for Adam with coupled weight decay and AdamW. Moreover, we compare them directly in Figure 15. Observe that for coupled weight decay we see that for both little and strong weight decay, the sparsity bias becomes more when the depth increases. In contrast, with less weight decay, AdamW for higher depth, the $L_1$-norm increases more. This is in line with the prediction for SignGF, which has the stationarity condition $||x||_{L_\infty} \leq \frac{1}{\alpha}$. Therefore, the parameter $x$ can move more freely and the geometry has less effect. However when the weight decay is increased we observe the opposite: we see a higher sparsity bias for deeper reparameterization. Furthermore, we report the corresponding validation accuracies in Table 6. Observe the significant accuracy drops for Adam with coupled weight decay for increasing the regularization, an indication for extreme sparsity.

We conduct the same experiment for a ResNet-50 on Imagenet (Deng et al., 2009). We report for depth $L = 2, 10$ the $L_1$ norm during training for both Adam with coupled weight decay and AdamW in Figures 13 and 14. Validation accuracy values are reported in Table 7. We observe the same behavior as for ResNet-20 on CIFAR-10, coupled weight decay leads to sparsity faster and with that a drop in generalization performance.

Table 5: Training details for all experiments presented on sparse reparameterizations.

| Dataset | Model | LR | Epochs | Batch Size | Optim | Schedule |
|---------|-------|-----|--------|------------|-------|----------|
| CIFAR-10 | ResNet-20 | 0.001 | 150 | 512 | Adam, AdamW | Triangular |
| Imagenet | ResNet-50 | 0.001 | 100 | 1024 | Adam, AdamW | Triangular |

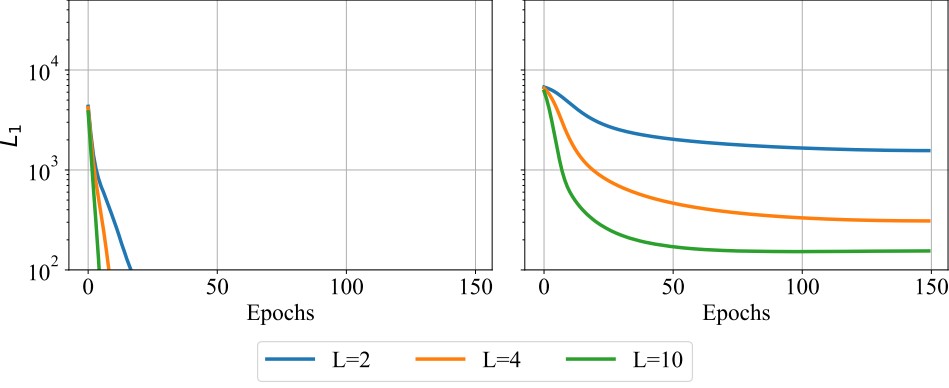

Figure 11: Adam with coupled weight decay trained with various depth reparameterizations for ResNet-20 on CIFAR-10. On the left is high regularization $1e-1$ and on the right is less regularization $1e-4$.

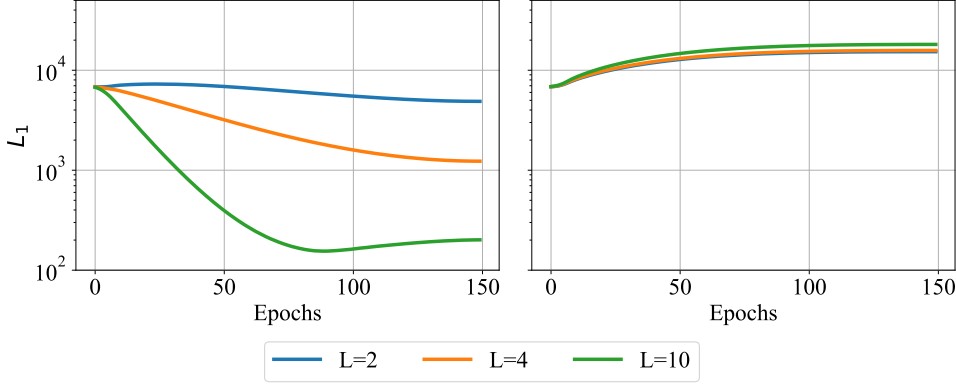

Figure 12: AdamW (decoupled weight decay) trained with various depth reparameterizations for ResNet-20 on CIFAR-10. On the left is high regularization $1e-1$ and on the right is less regularization $1e-4$.

Table 6: Test Accuracy (%) $\pm$ 95% CI for AdamW and Adam+wd across depths and weight decays training a ResNet-20 on CIFAR-10.

| Optimizer | Depth | Weight Decay | Accuracy $\pm$ CI |
|-----------|-------|--------------|-------------------|
| AdamW | 2 | $1e-1$ | $89.75 \pm 0.20$ |
| Adam+wd | 2 | $1e-1$ | $64.36 \pm 2.70$ |
| AdamW | 2 | $1e-4$ | $89.29 \pm 0.28$ |
| Adam+wd | 2 | $1e-4$ | $88.27 \pm 0.08$ |
| AdamW | 4 | $1e-1$ | $89.73 \pm 0.18$ |
| Adam+wd | 4 | $1e-1$ | $58.23 \pm 4.98$ |
| AdamW | 4 | $1e-4$ | $89.38 \pm 0.35$ |
| Adam+wd | 4 | $1e-4$ | $86.55 \pm 0.25$ |
| AdamW | 10 | $1e-1$ | $89.33 \pm 0.23$ |
| Adam+wd | 10 | $1e-1$ | $43.13 \pm 3.73$ |
| AdamW | 10 | $1e-4$ | $89.49 \pm 0.06$ |
| Adam+wd | 10 | $1e-4$ | $81.99 \pm 0.05$ |

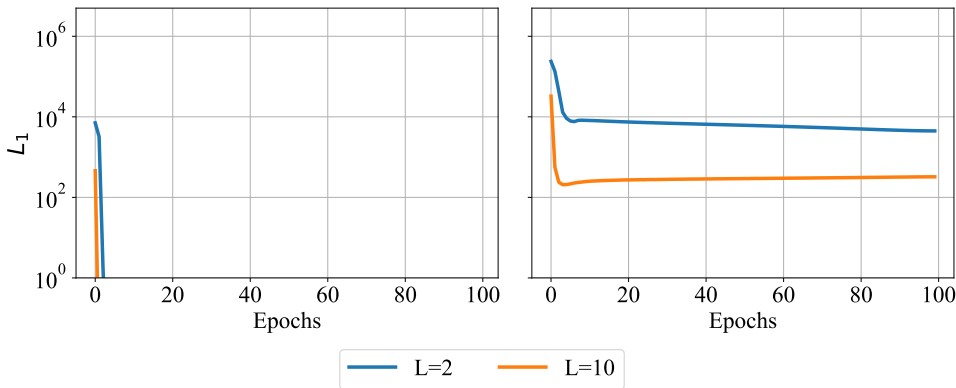

Figure 13: Adam with coupled weight decay trained with various depth reparameterizations for ResNet-50 on Imagenet. On the left is high regularization $1e-1$ and on the right is less regularization $1e-4$.

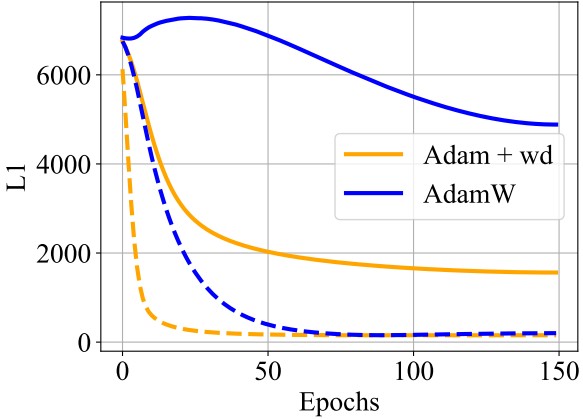

Figure 15: $L_1$ norm of the weights during training for Adam with coupled weight decay strength $1e-4$ and AdamW with $1e-1$. The dashed lines correspond to depth $L = 10$ and solid lines to $L = 2$. The training setup is ResNet-20 on CIFAR-10

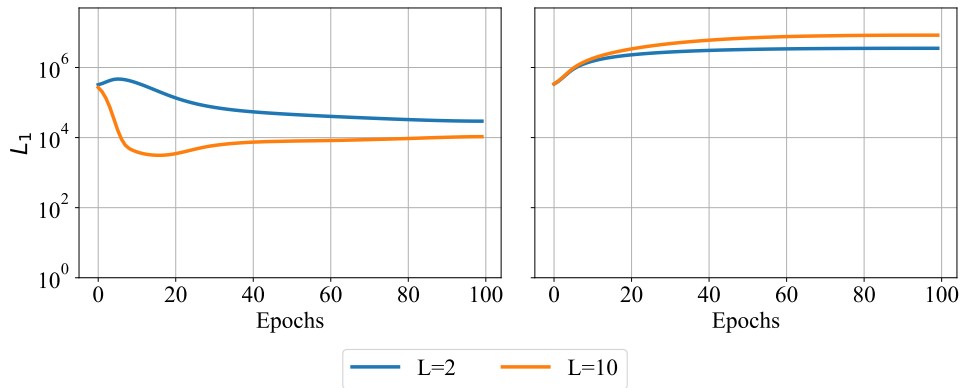

Figure 14: AdamW (decoupled weight decay) trained with various depth reparameterizations for ResNet-50 on Imagenet. On the left is high regularization $1e-1$ and on the right is less regularization $1e-4$.

Table 7: Test Accuracy (%) $\pm$ 95% CI for AdamW and Adam+wd across depths and weight decays training a Resnet 50 on Imagenet.

| Optimizer | Depth | Weight Decay | Accuracy $\pm$ CI |
|-----------|-------|--------------|-------------------|
| AdamW | 2 | $1e-1$ | $76.23 \pm 0.07$ |
| Adam+wd | 2 | $1e-1$ | $1.95 \pm 0.48$ |
| AdamW | 2 | $1e-4$ | $73.32 \pm 0.11$ |
| Adam+wd | 2 | $1e-4$ | $73.35 \pm 0.05$ |
| AdamW | 10 | $1e-1$ | $62.20 \pm 0.25$ |
| Adam+wd | 10 | $1e-1$ | $0.58 \pm 0.06$ |
| AdamW | 10 | $1e-4$ | $73.19 \pm 0.04$ |
| Adam+wd | 10 | $1e-4$ | $9.78 \pm 0.94$ |

## K    SADDLE ESCAPE FOR FINETUNING

In this section we present the saddle escape experiment for finetuning. We finetune a ResNet-18 that was pretrained on ImageNet on CIFAR-10 and Flowers. To do this, we have to replace the classifier layer with a new randomly initialized one. We finetune the model with two different optimizers: SGD and Adam. Both cases are run for 15 epochs with the best learning rate selected after a sweep for both Adam and SGD. The learning rates are selected from a preliminary sweep for Adam $\eta \in \{8e-4, 1e-3, 2e-3, 3e-3\}$ and SGD $\eta \in \{1e-2, 5e-2, 1e-1, 2e-1, 3e-1, 4e-1, 5e-1, 6e-1, 7e-1, 8e-1, 9e-1\}$. We also run the best learning rate for Adam for SGD to illustrate our main point of the saddle point escape. Note that for vision tasks, SGD usually outperforms Adam. However, in finetuning we observe the opposite. We track the top-50 largest eigenvalues during finetuning. For the experiment presented in the main text, we show the final eigenvalue distribution for the corresponding best validation accuracy.

In Table 8 and 9, the validation accuracy for both the CIFAR-10 and Flowers finetuning scenario are reported. Observe that Adam outperforms SGD in both cases. In addition, we report the distance traveled by all parameters (including the classification layer) in terms of the $L_1$ and $L_2$ norm. Adam has a much larger $L_1$ norm indicating more uniform movement of the parameters. In other words, the adaptiveness of Adam allows all parameters to move more, which is as expected. In Figures 17, 18 ,19, and 20 we report the top 50 eigenvalues for each seed, not normalized and similar for the Flowers finetuning in Figures 21, 22 ,23, and 24. We observe that the difference between the seeds is quite large. We believe that this is due to the randomly initialized classification layer. Furthermore, we report the normalized eigenvalues for each best seed also for Flowers finetuning in Figure 16. We observe less negative eigen values for Adam. Note that here we used standard SGD and Adam, that is, we are not using parameter efficient versions such as in (Zhou et al., 2025; Modoranu et al., 2024; Rios et al., 2025).

Table 8: Validation accuracy and parameter distance traveled in terms of $L_1$ and $L_2$ norm for finetuning ResNet18 on CIFAR-10.

| Metric | SGD ($\eta = 0.001$) | SGD ($\eta = 0.8$) | Adam ($\eta = 0.001$) |
|--------|--------|--------|--------|
| Val Acc | $19.15 \pm 2.82$ | $93.60 \pm 0.38$ | $95.19 \pm 0.21$ |
| $L_1$ | $424911.48 \pm 34308.92$ | $477750.60 \pm 10343.88$ | $693101.67 \pm 13509.59$ |
| $L_2$ | $29640.98 \pm 985.56$ | $28409.58 \pm 219.53$ | $27833.50 \pm 494.50$ |

Table 9: Validation accuracy and parameter distance traveled in terms of $L_1$ and $L_2$ norm for finetuning ResNet18 on Flowers.

| Metric | SGD ($\eta = 0.002$) | SGD ($\eta = 0.4$) | Adam ($\eta = 0.002$) |
|--------|--------|--------|--------|
| Val Acc | $1.22 \pm 0.53$ | $62.13 \pm 1.10$ | $80.50 \pm 1.38$ |
| $L_1$ | $206325.76 \pm 1327.51$ | $173882.76 \pm 2967.69$ | $618592.47 \pm 3445.18$ |
| $L_2$ | $10124.38 \pm 76.52$ | $7015.04 \pm 226.30$ | $11432.11 \pm 350.38$ |

### K.1    ADDITIONAL VISION FINETUNING EXPERIMENTS

We now present finetuning experiments using a large-scale transformer architecture, ViT-Large. We finetune a ViT-Large pretrained on ImageNet on CIFAR-10 for 30 epochs and on Flowers for 15 epochs. As is standard in finetuning, the original classifier head is replaced with a newly initialized one. We evaluate two optimizers—SGD and Adam—with learning rates selected via a sweep: $\eta \in \{9e-5, 1e-4, 1e-4, 5e-4\}$ for Adam and $\eta \in \{1e-3, 5e-3, 1e-2, 5e-2, 1e-1\}$ for SGD. Additionally, we run SGD with the best Adam learning rate to further illustrate our observations on saddle escape. All experiments use batch size 128, weight decay 0, cosine annealing learning rate scheduling, and label smoothing of 0.1. Because of the large model size and limited compute, we track only the top-25 eigenvalues.    Table. 10 and 11 report the validation accuracy on CIFAR-10 and Flowers, along with the $L_1$ and $L_2$ parameter distance traveled (including the classifier layer). Adam consistently achieves higher validation accuracy than SGD on both tasks. As in our earlier experiments, Adam induces a larger $L_1$ parameter shift, reflecting its more uniform adaptive updates. Figure. 25, 26, 27, 28, 29, 30 show the eigenvalue spectra across seeds and tasks. We additionally

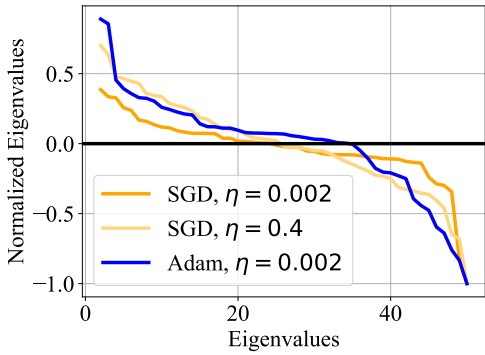

Figure 16: Normalized top-50 eigenvalues for a ResNet-18 finetuned on Flowers.

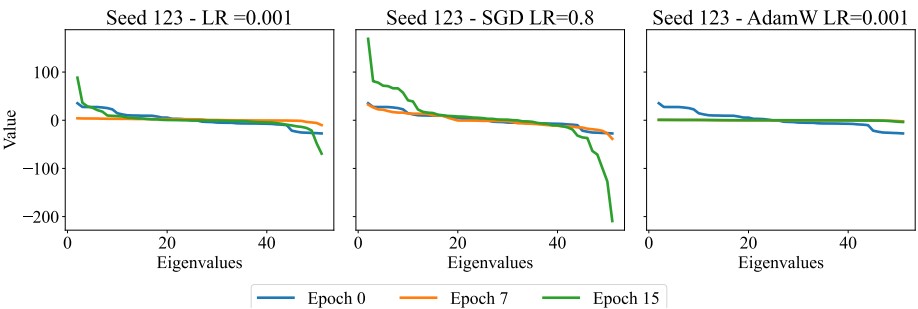

Figure 17: The eigen value evolution for seed 123 on CIFAR-10.

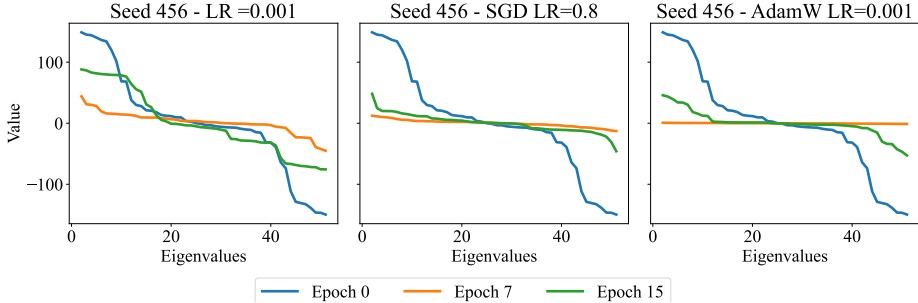

Figure 18: The eigen value evolution for seed 456 on CIFAR-10.

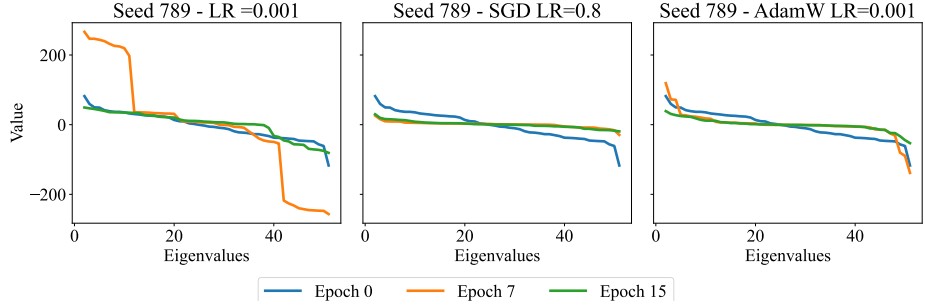

Figure 19: The eigen value evolution for seed 789 on CIFAR-10.

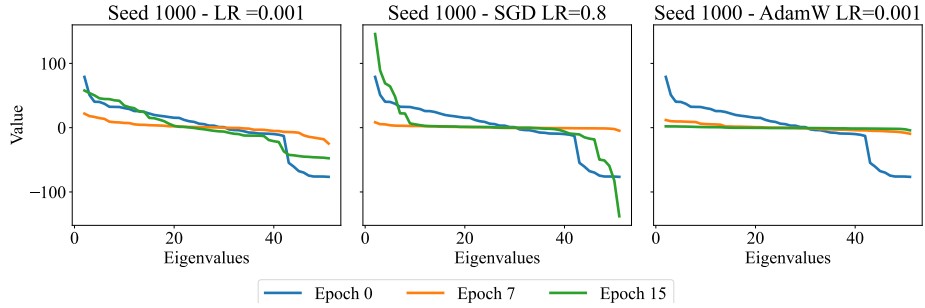

Figure 20: The eigen value evolution for seed 1000 on CIFAR-10.

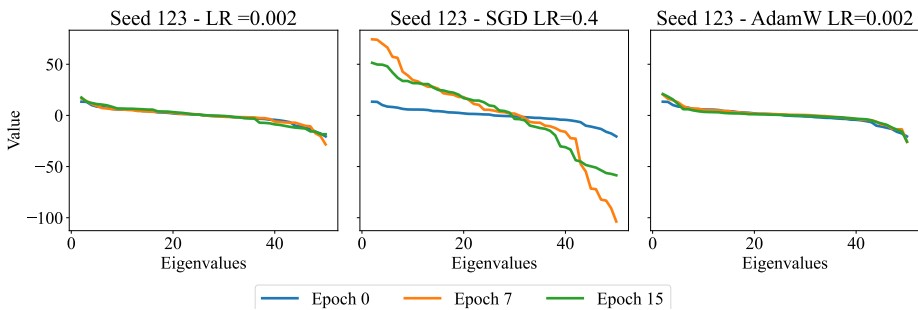

Figure 21: The eigen value evolution for seed 123 on Flowers.

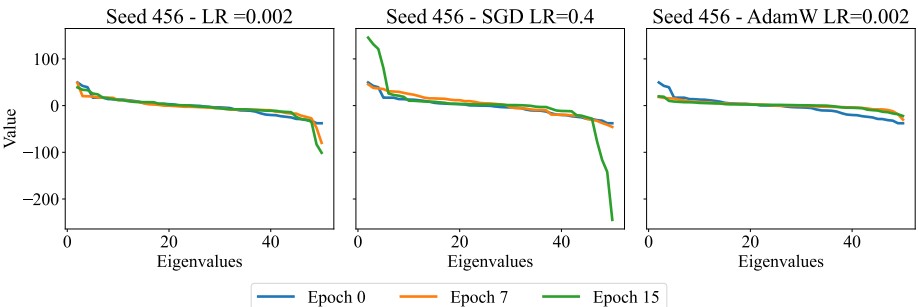

Figure 22: The eigen value evolution for seed 456 on Flowers.

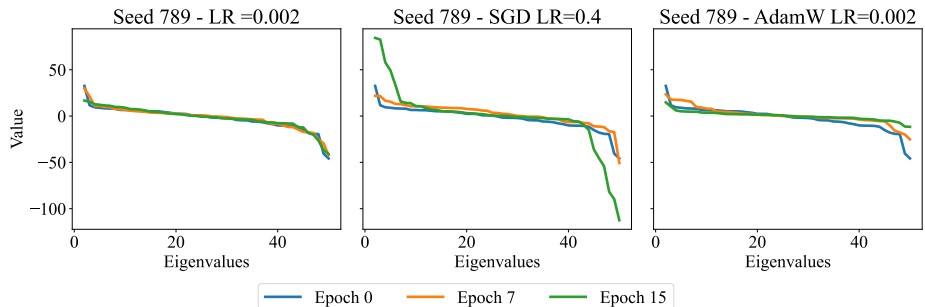

Figure 23: The eigen value evolution for seed 789 on Flowers.

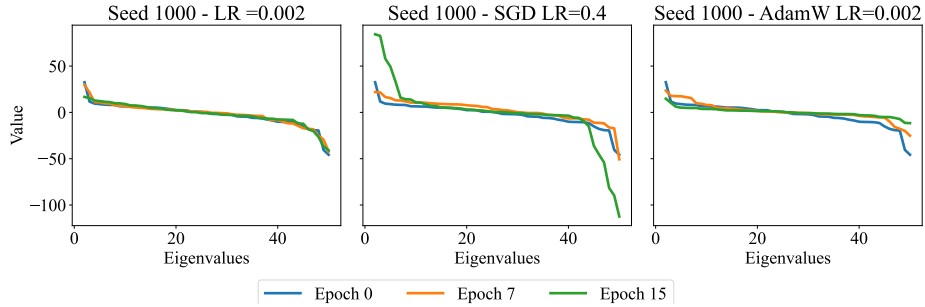

Figure 24: The eigen value evolution for seed 1000 on Flowers.

provide unnormalized and normalized spectra in Figure 31 and Figure 32 for different tasks. In the unnormalized CIFAR-10 spectra (Figure. 31a), SGD with a learning rate of $1e-4$ produces substantially larger eigenvalues than the other configurations, obscuring the trends for Adam and SGD with $1e-2$. Removing this outlier (Figure. 31b) reveals that Adam exhibits fewer negative eigenvalues. The same behavior holds for finetuning ViT-Large on Flowers.

Table 10: Validation accuracy and parameter distance traveled in terms of $L_1$ and $L_2$ norm for finetuning ViT-Large on CIFAR-10.

| Metric | SGD ($\eta = 0.0001$) | SGD ($\eta = 0.01$) | Adam ($\eta = 0.0001$) |
|---|---|---|---|
| Val Acc | $73.27 \pm 3.68$ | $99.07 \pm 0.35$ | $99.28 \pm 0.07$ |
| $L_1$ | $460.47 \pm 219.86$ | $24617.83 \pm 14406.4$ | $453934.906 \pm 22278.43$ |
| $L_2$ | $0.48 \pm 0.059$ | $6.25 \pm 4.29$ | $39.47 \pm 1.01$ |

Table 11: Validation accuracy and parameter distance traveled in terms of $L_1$ and $L_2$ norm for finetuning ViT-Large on Flowers.

| Metric | SGD ($\eta = 0.0001$) | SGD ($\eta = 0.01$) | Adam ($\eta = 0.0001$) |
|---|---|---|---|
| Val Acc | $1.03 \pm 0.82$ | $98.94 \pm 0.05$ | $99.37 \pm 0.08$ |
| $L_1$ | $25.71 \pm 30.48$ | $4655.83 \pm 576.49$ | $108583.62 \pm 2078.48$ |
| $L_2$ | $0.04 \pm 0.02$ | $1.50 \pm 0.12$ | $8.35 \pm 0.16$ |

## K.2 ADDITIONAL LANGUAGE FINETUNING EXPERIMENTS

In addition to our experiments on vision tasks, we conduct a parallel study on language models. Specifically, we fine-tune a pretrained BERT-base model on the MRPC task from the GLUE benchmark, following the setup in Zhou et al. (2025). The model is fine-tuned for 5 epochs using both SGD and Adam. Learning rates are selected via a sweep: $\eta \in \{5 \times 10^{-5}, 7 \times 10^{-5}, 9 \times 10^{-5}\}$ for Adam, and $\eta \in \{10^{-2}, 5 \times 10^{-2}, 10^{-1}, 5 \times 10^{-1}\}$ for SGD. We additionally evaluate SGD using the best learning rate obtained for Adam. As before, we track the top-50 eigenvalues throughout training. Table 12 reports the validation accuracy along with the parameter displacement measured in $L_1$ and $L_2$ norms. Figures 33, 34, and 35 show the evolution of eigenvalues across different random seeds. Figure 36 presents the unnormalized and normalized eigenvalue spectra for the model achieving the best validation performance. The conclusions mirror those observed in our vision experiments.

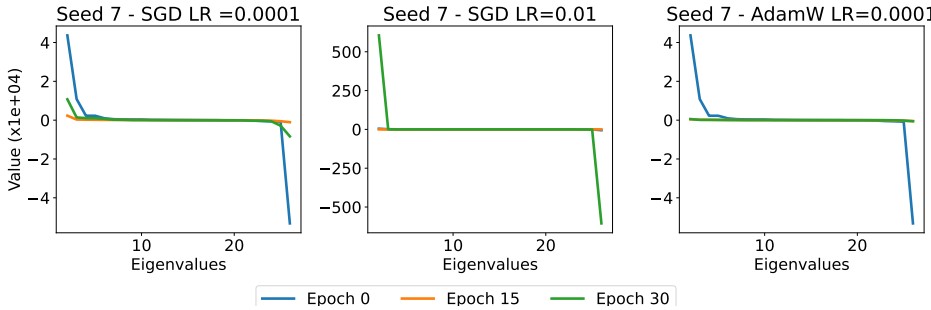

Figure 25: The eigen value evolution for seed 7 on finetuning ViT-Large on CIFAR-10.

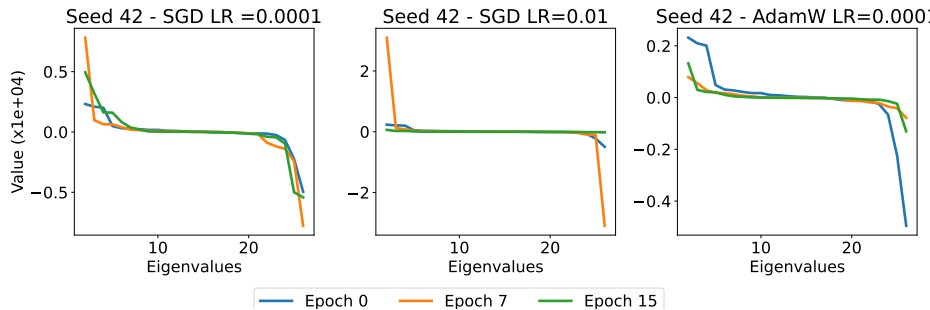

Figure 26: The eigen value evolution for seed 42 on finetuning ViT-Large on CIFAR-10.

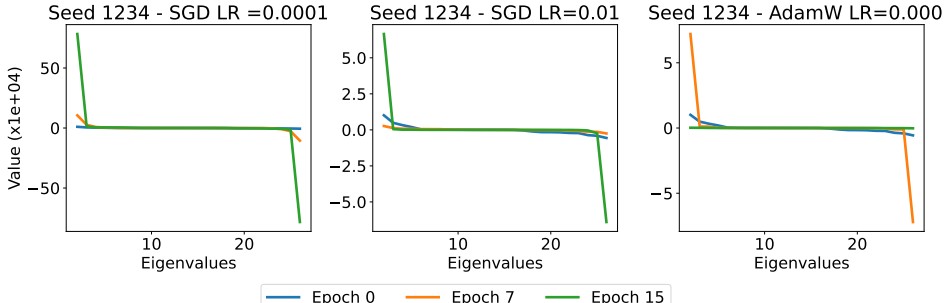

Figure 27: The eigen value evolution for seed 1234 on finetuning ViT-Large on CIFAR-10.

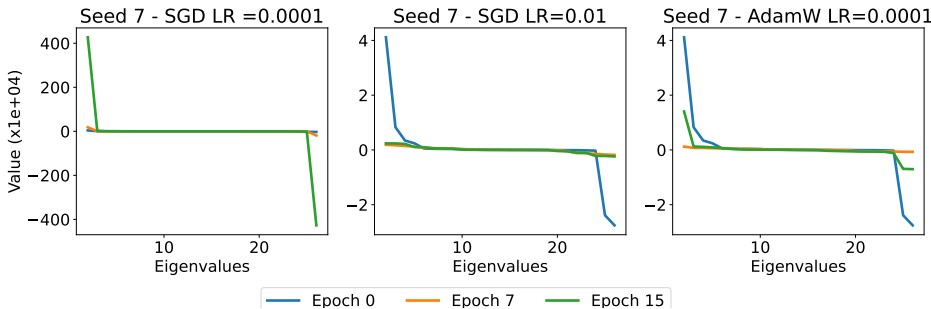

Figure 28: The eigen value evolution for seed 7 on finetuning ViT-Large on Flowers.

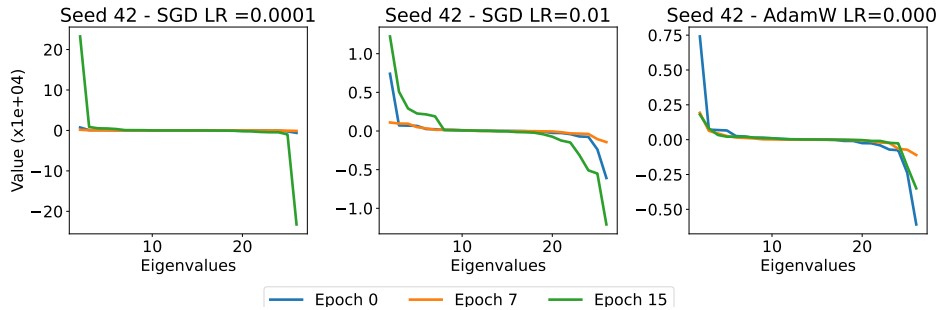

Figure 29: The eigen value evolution for seed 42 on finetuning ViT-Large on Flowers.

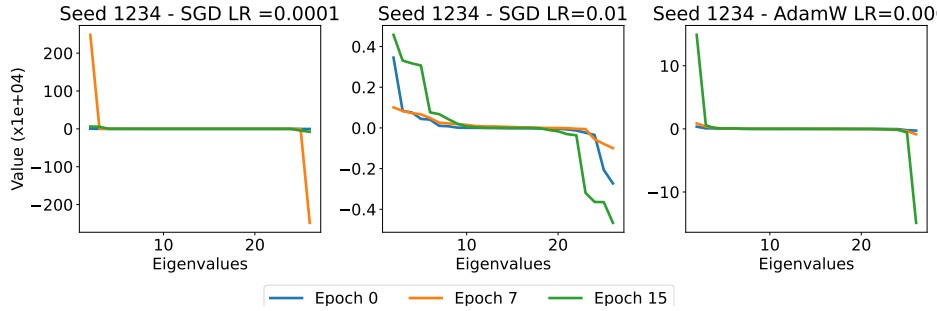

Figure 30: The eigen value evolution for seed 1234 on finetuning ViT-Large on Flowers.

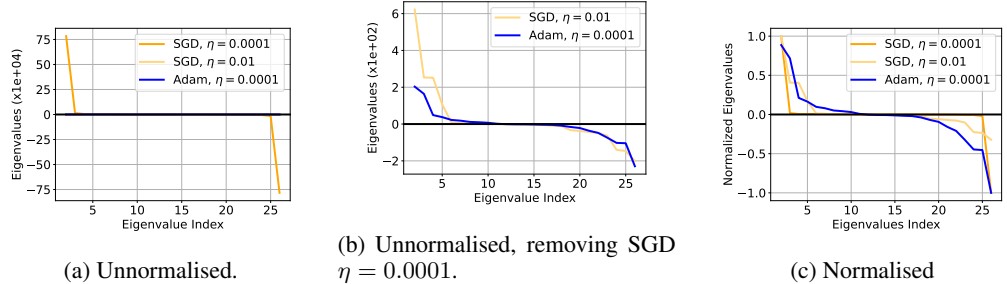

Figure 31: Top 25 eigenvalues of Hessian at solution obtained by SGD and Adam after finetuning ViT-Large on CIFAR10.

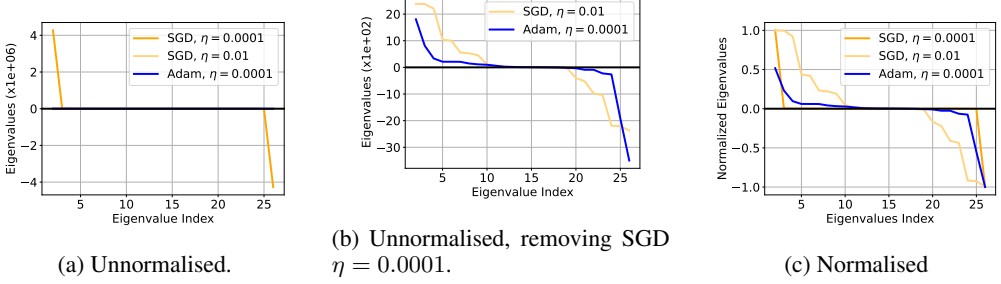

Figure 32: Top 25 eigenvalues of Hessian at solution obtained by SGD and Adam after finetuning ViT-Large on Flowers.

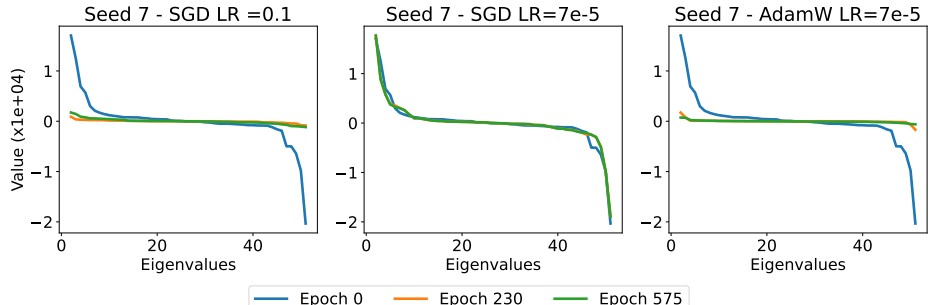

Figure 33: The eigen value evolution for seed 7 on finetuning Bert-base on MRPC.

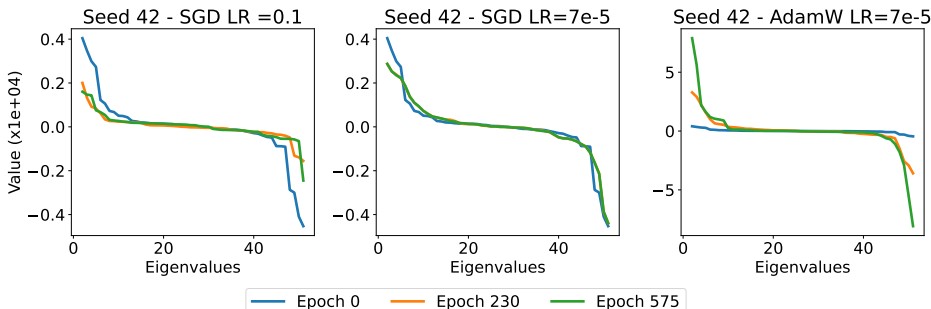

Figure 34: The eigen value evolution for seed 42 on finetuning Bert-base on MRPC.

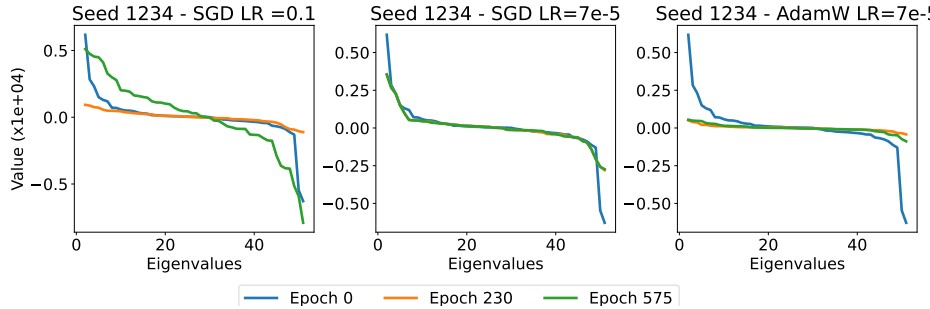

Figure 35: The eigen value evolution for seed 1234 on finetuning Bert-base on MRPC.

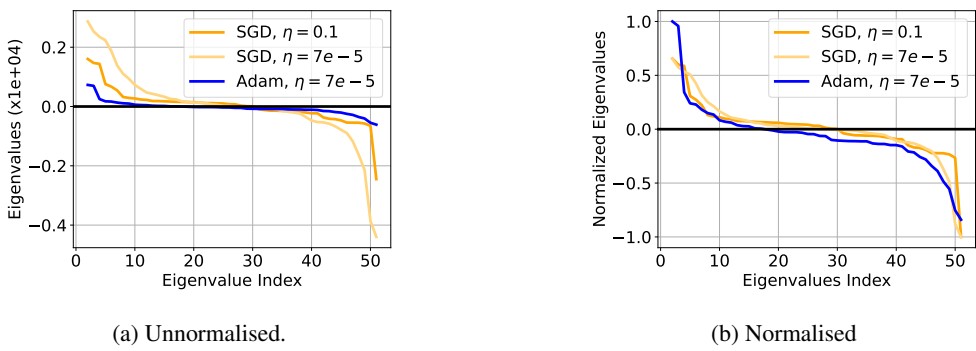

(a) Unnormalised.      (b) Normalised

Figure 36: Top 50 eigenvalues of Hessian at solution obtained by SGD and Adam after finetuning Bert-base on MRPC.

Table 12: Validation accuracy and parameter distance traveled in terms of $L_1$ and $L_2$ norm for finetuning Bert-base on MRPC.

| Metric | SGD ($\eta = 7e-5$) | SGD ($\eta = 0.1$) | Adam ($\eta = 7e-5$) |
|--------|---------------------|--------------------|-----------------------|
| Val Acc | $43.87 \pm 24.02$ | $84.80 \pm 1.00$ | $85.95 \pm 0.64$ |
| $L_1$ | $5002.44 \pm 0.0$ | $6066.93 \pm 34.33$ | $31079.54 \pm 754.26$ |
| $L_2$ | $0.73 \pm 0.00$ | $1.26 \pm 0.01$ | $5.57 \pm 0.24$ |

