# OpenReview forum: "Never Saddle for Reparameterized Steepest Descent as Mirror Flow"
_ICLR.cc/2026/Conference — ICLR 2026 Poster_

### Official Review · Reviewer_NKAm · 2025-10-18

**Soundness:** 2
**Presentation:** 2
**Contribution:** 2
**Rating:** 4
**Confidence:** 4

**Summary:**

This paper studied and compared the dynamical behaviors of different types of steepest descent algorithms. The authors focused on deep linear diagonal models and showed that a steepest flow in the weight matrices with respect to $L_p$ norm induces a "steepest mirror flow" in the end-to-end matrix. They analyzed how the choice of $p$ influences the dynamics, convergence rate and the effects of weight decay. Experiments on both linear and practical models are provided for the comparison between different algorithms.

**Strengths:**

The paper extends the study of reparametrization in gradient flow to steepest flow. This provides a useful platform for studying how reparametrization and optimization geometry interact to shape the training dynamics. I find this extension meaningful and valuable. Several established results are quite interesting:

* Lemma 4.4: The balance equation for steepest flow in training deep diagonal linear network is new and interesting.

* Theorem 4.6: The equivalence between steepest flow with reparametrization and "steepest mirror flow" is quite interesting. To my knowledge, previous work only showed the equivalence between gradient flow with reparametrization and mirror flow.

* Corollary 4.8: It sheds light on how different choices of optimization geometry and reparametrization (network depth) could affect the convergence rate.

**Weaknesses:**

The main weakness is that the main claims are not well supported by the presented theory. The title begins with "Never saddle" and the authors emphasized both in the abstract and in Contributions that "we prove that steeper descent (lower $q$) simultaneously escapes saddles faster and supports feature learning", and that "decoupled weight decay ... yielding more stable feature learning". While these statements are strong and intriguing, I do not find them adequately supported by the theoretical results.

**Regarding saddle escaping**

After Lemma 4.4 and in Figure 3, the authors claimed that "the (curved) path away from zero is shorter for smaller $q$, indicating faster saddle escape". This reasoning is not logically sound: First, the notion of a "shorter path" is ambiguous (e.g., between which endpoints?). Second, the geometry of the invariant manifold alone does not determine how fast the algorithm moves along the manifold.

Besides the above, the only theoretical evidence for the claim that "lower $q$ escapes saddles faster" is based on two results: (i) Corollary 4.8: under initialization "$w_1 = 0$ and $w_i = \lambda > 0$", a smaller $q$ yields a larger coercivity constant $\mu$; and (ii) Theorem 4.2: when $R$ is a separable Bregman function, a larger $\mu$ yields a faster linear convergence rate. However, I do not find these two results sufficient for the claim, for the following reasons:

* It is not established that the $R$ used in Corollary 4.8 (or Theorem 4.6) is a separable Bregman function. Thus it is unclear whether Theorem 4.2 applies;

* The considered initialization "$w_1 = 0$ and $w_i = \lambda > 0$" lies in a measure-zero set;

* Only a subset of saddles (those near the considered initializations) are analyzed, whereas, as noted by the authors, other saddles exist elsewhere in the parameter space.

Therefore, even if Theorem 4.2 is applicable, the results only indicate that: for certain specific saddles and for a measure-zero set of initializations in their neighborhoods, a smaller $q$ leads to a faster escape. I thus think it is overstated to use the title "never saddle", or to claim that smaller $q$ results in faster saddle escaping, which sounds like a general statement, just based on these results.



**Regarding feature learning**

The authors claimed that "smaller $q$ supports feature learning". However, in the theory section (Section 4), I could not find a clear discussion on how the choice of $q$ affects feature learning. The following two aspects might be relevant, but it is unclear how they support the claim:

* In Corollary 4.11 and 4.12, the authors discussed whether the function $R_{L_p, L}$ is a Legendre function under different $q$, network depth $L$, and metric exponent $m$. Then in the left panel of Figure 2, the authors indicated that a metric exponent slightly smaller than $1$ correspond to a feature learning regime (the green band in the figure). However, it is not explained why this is true and how does metric exponent relate to feature learning. For example, why does $m=0.95$ lead to feature learning, while $m=0.5$ or $m=1.5$ does not?

* In Theorem 4.14, the authors discussed the on manifold regularization induced by weight decay under different $q$. However, as shown in Table 1, when $q=2, L=2$, the decoupled weight decay induces an $L_1$ bias; whereas when $q=1, L=2$, it induces an $L_{3/2}$ bias, which is less sparse than $L_1$. As sparsity biases have been linked to feature learning in some settings, this observation actually suggests that $q=2$ may facilitate feature learning instead of a smaller $q$.

The authors also claimed that "decoupled weight decay, as in AdamW, stabilizes feature learning by enforcing novel balance equation". The balance equation in Lemma 4.4 indicates that the weight decay encourages the weights to become more balanced during training. But it is not clear to me how this balancing translates into a more "stable" feature learning.

I suggest that the authors clarify in the paper what they mean by "feature learning" and "stable feature learning", and then compare the algorithms with different values of $q$ as well as with and without weight decay.


**Initial recommendation**

Overall, while the study of how optimization geometry and reparametrization affect the dynamics and the proposed framework are very interesting, I find the main claims of this paper, in particular those on saddle escaping and feature learning, insufficiently supported. Therefore, my initial recommendation is rejection.

**Questions:**

I find some parts of the presentation unclear. Please see the questions below.

**Notations**

* In Example 3.2, is the expression "a deep diagonal linear network $g(m,w)=m \odot w$" actually referring to a shallow network, as there are only two weight matrices $\operatorname{Diag}(w), \operatorname{Diag}(m)$?

* In Definition 3.3 and 4.5, does $\mathbb{I}_n$ denote the all-one vector? This symbol is commonly used for the identity matrix, so clarification would be helpful.

* In Corollary 4.8, the notation $w_i=\lambda>0$ is ambiguous. Does it mean the vector $w_i$ has all entries equal to $\lambda$?

**Regarding Theorem 4.6**

* $\lambda−L_p-$balancedness in Definition 4.5 is defined for shallow network with two weight matrices. Then what does "$\lambda−L_p-$balanced with respect to the first parameter $w_1$" mean in Theorem 4.6, which is stated for deep networks?

* In Lemma 4.4 and Theorem 4.6, in what sense does "almost everywhere" mean (e.g.,"steepest descent satisfies a separable $L_p$-mirror flow almost everywhere")? Does it mean the result may fail only for a measure-zero set of initializations?

* Corollary 4.12 discussed when the function $R_{L_p,\ L}$ is a Legendre function. However, in Theorem 4.6, there seems no restriction on $p$ or $L$. Does Theorem 4.6 indicate that such an $R_{L_p,\ L}$ always exists, even in cases where it is not a Legendre function?

---

> ### Author Response · Authors · 2025-11-21
> **Part 1**
>
> We would like to express our gratitude for your time and efforts in providing valuable comments on our manuscript. Below, we elaborate on your concerns in a detailed point-by-point response. In case of any open questions, we would be happy to discuss them.
>
> **Weaknesses:**
>
> **Saddle escaping:**
> We have made the interpretation more concrete by stating: Observe that the path away from initialization to a point on the curve $m w = x$ for $x \in \mathbb{R}$. Furthermore, we include in the plot two exemplary curves for $m w = \pm 0.1$. The figure can be interpreted in the following way: Converging to the same parameters $x$, a smaller $q$ reaches there faster, which is a visual indication of a more effective saddle escape (assuming that we train for a finite time horizon).
>
> Moreover, we have now explicitly stated that $R_{L_p,L}$ is a Bregman function for $(L-1)/L q \leq 1$. This follows from the properties of the derived implicit function for $\nabla^2  R_{L_p,L}$. More concretely, we use that it is strictly positive (bounded from below), it is an even function, and utilize the asymptotic behavior for large $|x|$, which was also used in Lemma 4.13.
>
> While we consider a specific initialization, the result can be extended, as now detailed in Remark 4.7. The result would not change when we consider different $\lambda$ for different coordinates, as the proof techniques apply pointwise. Moreover, in the case of $L = 2$, we are always able to apply our analysis even to random initialization. This follows from the fact that $\lambda$ would be generated by the initialization. Note that our initialization assumption is a common one and also used in [1,2,3].
>
> The subset of saddles we study are inherent in the model structure (Line 293). Additional saddle points could be induced by the data set. We have clarified this in the introduction. We would like to note that the title is a word play. We propose to change the title to: “Never Saddle Down for Reparameterized Steepest Descent as Mirror Flow”. This connects the saddles directly to the reparameterizations.
>
> **Feature learning:**
> We specify in the following and the revised manuscript what we mean with stability and feature learning.
> 1) Stability: For stability of the dynamics, we need $2- (L-1)/L q \geq 1$. If this is not fulfilled, the mirror map becomes non-convex, violating the Legendre and Bregman function properties, which can lead to finite time blow up of the dynamics as highlighted in Figure 2.
>
> 2) Feature learning: In our context, we define feature learning as having an implicit sparsity bias. For this purpose, we want $2 - q(L-1)/L$ to be as small as possible in Theorem 4.13. However, stability gets violated when $2- (L-1)/L q \leq 1$.  Moreover, in order to have feature learning in the first place, we need $\lambda \rightarrow 0$ as it controls the balance equation. This decrease of $\lambda$ is accomplished by the decoupled weight decay, as shown in Lemma 4.5. Therefore, decoupled weight decay induces stable feature learning by enforcing balance of the parameters. We have clarified this in the revised manuscript.
>
> Moreover, Appendix I illustrates feature learning with an experiment on varying $q$ and depth $L$.
> Our experiment on coupled vs decoupled weight decay experiment for sparsity furthermore shows that higher weight decay leads to more sparsity as $\lambda \rightarrow 0$.
>
> **Questions:**
>
> **Notation:**
> 1) It would also be possible to write the network in this diagonal form. We have added this point to the revised manuscript.
> 2) This refers to the vector whose components are all 1. We changed it to the boldface $1$ with subscript $n$ and explained the notation.
> 3) We have added a boldface one to make this more clear.

---

> ### Author Response · Authors · 2025-11-21
> **Part 2**
>
> **Theorem 4.6:**
> 1) $\lambda$-$L_p$-balancedness with respect to the first parameter means that all other parameter vectors / diagonal matrices $w_i$ for $i > 1$ satisfy the $\lambda$-$L_p$-balancedness with respect to $w_1$. We have added this explanation to the statement in the revised manuscript.
> 2) Almost everywhere refers to the type of solution for these flows. It is a measure theoretic concept. It means that it applies to all points (potentially except points of Lebesgue measure 0). In other words, a solution has to satisfy the given relationship (flow or equality) almost everywhere.
> 3) The function $R_{L_p, L}$ can be always constructed, although the resulting flow may have no global solution, i.e., has finite blow up as shown in Figure 2 (right). The description still holds, however, the trajectories may be unstable due to not having the Legendre properties. Therefore, locally, so in a ball around the initialization, the description makes sense. However, we have adapted the statement by adding that for $q (L-1)/L \leq 1$ the function is a Bregman function. This is also in line with the stability characterization as detailed in the feature learning paragraph.
>
> [1] Pesme, Scott et al. “Implicit Bias of SGD for Diagonal Linear Networks: a Provable Benefit of Stochasticity.” ArXiv abs/2106.09524 (2021): n. pag.
>
> [2] Wang, Shuyang and Diego Klabjan. “A Mirror Descent Perspective of Smoothed Sign Descent.” Conference on Uncertainty in Artificial Intelligence (2024).
>
> [3] Jacobs, Tom and Rebekka Burkholz. “Mask in the Mirror: Implicit Sparsification.” ArXiv abs/2408.09966 (2024): n. pag.

---

> > ### Comment · Reviewer_NKAm · 2025-11-27
> >
> > Thank you for the detailed responses! However, some of my concerns remain:
> >
> > **Figure 3**: As noted in my initial review, the time that the algorithm spends on a path depends on both the path length and the velocity $\\|\dot{\theta}(t)\\|$; a shorter path does not necessarily imply faster escape (the flow may have a much smaller velocity on the shorter path). I believe additional analysis is needed for the claim "Converging to the same parameter $x$, a smaller $q$ reaches there faster".
> >
> > Additionally, it is not obvious in Figure 3 that the $q=1$ line from $(0.1, 0)$ to $mw=0.1$ is shorter than the $q=2$ curve. In fact, in the limit $x\to 0$, the manifold $mw=x$ ($w=f(m)=x/m$) becomes nearly horizontal in the neighborhood of $(0.1,0)$ (since $f'(0.1)=-x/0.01 \to 0$). Now, the $q=2$ curve is clearly shorter than the $q=1$ line, since at $(0.1,0)$ the tangent of the $q=2$ curve is vertical (perpendicular to $mw=x$), whereas the tangent of the $q=1$ line lies in the direction $(1,1)$.
> >
> > **Theorem 4.2**: If I understand the authors correctly, Theorem 4.2 applies only when $q\frac{L-1}{L}\leq 1$, i.e., $q\leq \frac{L}{L-1}$. For large depth $L$, this restricts $q$ to values only slightly above $1$, and when $L > 2$, it does not apply to gradient flow ($q=2$). Since Theorem 4.2 is the core result in discussing saddle escape, this requirement limits the main scope of the paper: the comparison between GF and steepest flow.
> >
> > **Stability**: Thanks for the clarification on stability and sparsity. I understand them better. However, I think that violating the condition $q\frac{L-1}{L} \leq 1$ only means that the function $R$ is not a Bregman divergence; it does not necessarily imply that the dynamics is unstable, or the loss would blow up. Could the authors provide a theoretical justification for the claim that "If this is not fulfilled, ..., which can lead to finite time blow up of the dynamics"? In general, I expect that GF on deep diagonal networks does not always blow up, even in this case $q\frac{L-1}{L} >1$.
> >
> > **Table 1**: My initial concern appears unresolved. According to the table, for $L=2$, in both coupled and decoupled settings, $q=2$ induces a bias that is as sparse as, or even sparser than, the bias induced by $q=1$. This seems to contradict the main claim that "smaller $q$ supports feature learning".

---

> > > ### Author Response · Authors · 2025-12-03
> > >
> > > Thank you for your response. We are grateful for your suggestions, which have substantially improved the manuscript. We are also happy to provide additional clarifications.
> > >
> > > **Figure 3**
> > > We agree that the path is not the only determining factor controlling the velocity of the parameter evolution. Figure 3 is there to provide intuition on why saddle escape happens for smaller $q$ in the feature learning regime. We have clarified that the purpose of this figure is solely to provide an intuition in the revised manuscript. To show the insight formally, we can calculate the arclength for both $q = 1$ and $q = 2$. Because of symmetry, without loss of generality, we can assume that $m, w \geq 0 $. We need to write the differential change in movement over the curve as a change in x. In other words, we need to find dx over the curve. To do this, we calculate the differential arclength, which is given by $\sqrt(dm^2 + dw^2)$ and use the relationship $x = mw$ and the corresponding invariant for a given $q$. For a fair comparison, we use $|m| - |w| = \sqrt{\lambda}$ and $m^2 -w^2 = \lambda$, which correspond to the same starting point. We have to distinguish two cases, as follows.
> > >
> > > 1) Case $q= 1$: $\sqrt(dm^2 + dw^2) = \sqrt{2} dw$ and $x = w^2 + \sqrt{\lambda}w$ gives $dx = (2w + \sqrt{\lambda}) dw$, integrating gives: $\int_0^x f_1(x) dx \simeq \sqrt{2x} - \frac{\sqrt{2}}{2}\sqrt{\lambda} $ where $f_1(x) =  \frac{\sqrt{2}}{\sqrt{\lambda + 4x}} $. The integral can be solved explicitly.
> > > 2) Case $q=2$: $\sqrt(dm^2 + dw^2) = \sqrt(\frac{2w^2 +\lambda}{w^2 + \lambda}) dw$ and $x = w \sqrt{w^2 +\lambda^2}$, integrating gives:   $\int_0^x f_2(x) dx =  \sqrt{2x} - 0.599\sqrt{\lambda} $ where $f_2(x) = \frac{1}{\sqrt{\lambda} (1+4x^2/\lambda^2)^{1/4}} $. The integral can be approximated with the help of a parameter substitution of $y = 2x/\lambda$. The constant $0.599$ comes from approximating the definite integral.
> > >
> > > The fact that the correction for $q=1$ is larger ($0.707$) than that of $q=2$ ($0.599$) shows that the arc length for signGF is shorter than that for GF. Note that the integral of $q=2$ has no closed form expression. Therefore, we study the Taylor approximation at $\lambda = 0$ close to the feature learning regime, which we are mainly interested in.
> > >
> > >
> > > **Theorem 4.2 and finite blow up**
> > > Note that we are mainly interested in the stability of the dynamics for a given $q$. Smaller values of $q$ increase the stability for higher depths, not the other way around.
> > > Furthermore, when $q < \frac{L}{L-1}$, $R$ is not “essentially smooth” due to having a too large metric exponent. Essentially smooth is defined as: When $|x| \rightarrow \infty$, then $|\nabla R(x)| \rightarrow \infty$. Intuitively, this means that the cost of movement for the parameters grows when the parameter becomes larger in magnitude. If this is violated, the parameters can move with less and less cost at larger magnitudes. This can lead to a finite time blow up globally, as it becomes easier to escape. This does not mean that this always happens locally. The dynamics can be stable.
> > > Another cause of the large metric exponent is exponential slowdown as illustrated in Figure 2 (right) of the dynamics, as is known from the saddle-to-saddle dynamics [1].
> > > Both phenomena result from the same cause, i.e. a small metric exponent, and can prevent or hamper feature learning. We have clarified this point in the revised manuscript.
> > >
> > > **Sparsity bias**
> > > Feature learning requires two elements: saddle point escape and sparsity bias. For large $q$, the implicit sparsity bias is stronger but saddle point escape is harder, which prevents or hampers feature learning (Figure 2). Smaller $q$ demands larger depth to induce feature learning. This is a prerequisite for saddle escape. For clarity, we have included the discussion on depth in the contribution statement.
> > >
> > > [1] Scott Pesme and Nicolas Flammarion. Saddle-to-saddle dynamics in diagonal linear networks. In
> > > Thirty-seventh Conference on Neural Information Processing Systems, 2023. URL https://
> > > openreview.net/forum?id=iuqCXg1Gng.

---

### Official Review · Reviewer_e8j7 · 2025-10-31

**Soundness:** 3
**Presentation:** 3
**Contribution:** 3
**Rating:** 6
**Confidence:** 4

**Summary:**

This work introduces steepest mirror flows as a unifying geometric framework to study how optimization algorithms influence reparametrization. By analyzing diagonal linear networks and deep diagonal reparameterizations, the authors show that steeper descent methods, such as sign-based variants, escape saddle points more efficiently than gradient descent. Empirical results from linear regression, classification, and fine-tuning experiments confirm the theoretical predictions of faster saddle escape, stable learning, and distinct sparsity dynamics between coupled and decoupled weight decay.

**Strengths:**

This is an interesting study with insightful findings. Combining mirror descent with reparameterization is a great idea.

**Weaknesses:**

The statement of the first contribution is misleading. This is not the first work studying GF and reparametrization. It is probably meant with respect to the family.

For this type of work, assumptions are always problematic. They are very simplistic (diagonal networks) but it is very hard to potentially show more general results.

In experiments, rather than studying the networks considered in the analysis, they should see if the results hold when the assumptions are not fulfilled (for example, other networks).

**Questions:**

What is meant by: iterates x_t converge? I assume it is meant lim_t->\infinity x_t

---

> ### Author Response · Authors · 2025-11-21
> **Part 1**
>
> We would like to express our gratitude for your time and efforts in providing valuable comments on our manuscript. Below, we elaborate on your concerns in a detailed point-by-point response. In case of any open questions, we would be happy to discuss them.
>
> Weaknesses:
>
> **W1: Contribution specification.** We have specified the statement and explicitly reference the family of steepest flows in the contribution: Steepest mirror flows for a family of reparameterized steepest flow dynamics.
>
> **W2: Limited scope.**
> One of the main goals of theory in deep learning is to describe and explain phenomena that occur in practice. While our model is simple, it does give multiple valuable insights into the mechanics of relevant practical algorithms. Concretely, it allows us to make predictions for real world settings.
>
> **New geometric mechanism**
> Our geometric approach using mirror flow reveals a different principle at work that previous work has shown to escape saddle points. We show that the geometry is sufficient for escaping saddles in our setup, while previous work relies on noise and large learning rate or time rescaling [6]. Therefore, we believe revealing other mechanisms is of great value. In the revised manuscript, we include this discussion on noise and how it aids in saddle point escape in the related work. Our mechanism is corroborated by experiments, see for example Figure 5a where SGD with small learning rate cannot escape the saddle point indicating that noise is not sufficient. Moreover we have included additional experiments on saddle point escape in the appendix.
> Finally, revealing a structural mechanism helps us to identify where to look for developing new theoretical tools to study training dynamics. While we agree that a straightforward generalization using mirror flows may be hard, it can be used as a stepping stone for guiding other theoretical approaches and motivating identifying geometric properties. This is also in line with recent position papers calling for studying algebraic properties of models [4] and studying linear models [5].
>
> **Main use case: finetuning and saddle escape**
> Since Adam and its variants remain the most widely used optimizers for finetuning, and sign decent (steepest descent wit respect $L_{\infty}$ norm) serves as a proxy for Adam, the steepest mirror flow analysis naturally extends to the finetuning setting. In particular, finetuning typically employs a small learning rate to avoid large deviations from the pretrained parameters and to mitigate catastrophic forgetting. This aligns well with our flow analysis. In contrast, SGD has to use a significantly larger learning rate for saddle point escape, which is not desired in finetuning as it would induce catastrophic forgetting. This observation is substantiated by our real world experiments with transformers. We highlighted this in Figure 5a and in additional ablations in Appendix K with new additions on ViTs and Transformers in K.1 and K.2..
>
> **A theoretical use case: feature learning**
> Recent work has characterized the implicit bias for steepest descent flows for homogeneous neural networks on binary classifications for separable data [3]. We have applied their result in Theorem F.1 and highlighted that the $L_{\infty}$ max margin optimization problem cannot distinguish between solutions that correspond to overfitting or feature learning (sparsity). In contrast our theory predicts that depth changes the geometry and with that the bias towards sparsity and feature learning, which is also illustrated in the experiment depicted in Figure 4. This is a clear sign that we need additional tools and theory to describe the implicit bias further.
>
> **A practical use case: sparsity**
> As discussed, pointwise reparameterizations have been proposed together with weight decay to induce sparsity [1,2]. An integration with modern optimizers is valuable, as it leads to concrete design principles. Our theory predicts that for smaller $q$, so for SignGF, we need to have higher depth to induce sparsity (and thus feature learning). Moreover, Figure 5b shows that using AdamW requires higher depth to induce sparsity. This may lead to better sparse training algorithm design in the future.

---

> ### Author Response · Authors · 2025-11-21
> **Part 2**
>
> **W3: Real world examples.** We would like to point out our finetuning experiments, which use ResNet architectures (not reparameterized). Here we show that Adam can escape saddles faster for a small learning rate. This claim is substantiated by a spectral analysis. Furthermore, Adam achieves better generalization than SGD (even compared to SGD with large lr). With the available additional space, we highlight the generalization performance in Table2 as part of the main text for both CIFAR-10 and Flowers. We also include additional experiments on finetuning with a vision transformer and an LLM.
>
> **Additional experiment: Finetuning**
> We also provide experiments in both the vision and language domain with transformer architectures.
> Specifically, we finetune a ViT-Large pretrained on ImageNet on CIFAR-10 for 30 epochs and on Flowers for 15 epochs. As is standard in finetuning, the original classifier head is replaced with a newly initialized one. We evaluate two optimizers—SGD and Adam—with learning rates selected via a sweep: $\eta \in \{9e-5, 1e-4, 1e-4, 5e-4\}$ for Adam and $\eta \in \{1e-3, 5e-3, 1e-2, 5e-2, 1e-1\}$ for SGD. Additionally, we run SGD with the best Adam learning rate to further illustrate our observations on saddle escape. All experiments use batch size $128$, weight decay $0$, cosine annealing learning rate scheduling, and label smoothing of 0.1. The table below reports validation accuracy along with the parameter shift measured in $L_1$ and $L_2$ norms. Adam consistently outperforms SGD and induces a more uniform parameter update pattern, reflected in its substantially larger $L_1$ norm.
>
> | Dataset   | Metric | SGD ($\eta=0.0001$) | SGD ($\eta=0.01$) | Adam ($\eta=0.0001$) |
> |-----------|--------|----------------------|--------------------|------------------------|
> | **CIFAR-10** | Val Acc | $73.27 \pm 3.68$ | $99.07 \pm 0.35$ | $99.28 \pm 0.07$ |
> |           | $L_1$   | $460.47 \pm 219.86$ | $24617.83 \pm 14406.4$ | $453934.91 \pm 22278.43$ |
> |           | $L_2$   | $0.48 \pm 0.059$ | $6.25 \pm 4.29$ | $39.47 \pm 1.01$ |
> | **Flowers** | Val Acc | $1.03 \pm 0.82$ | $98.94 \pm 0.05$ | $99.37 \pm 0.08$ |
> |           | $L_1$   | $25.71 \pm 30.48$ | $4655.83 \pm 576.49$ | $108583.62 \pm 2078.48$ |
> |           | $L_2$   | $0.04 \pm 0.02$ | $1.50 \pm 0.12$ | $8.35 \pm 0.16$ |
>
>
>
> In addition to vision experiments, we conduct a parallel study on language models. Specifically, we finetune a pretrained BERT-base model on the MRPC task from the GLUE benchmark. The model is finetuned for 5 epochs using both SGD and Adam. Learning rates are selected via a sweep:
> $\eta \in \{5\times10^{-5},\, 7\times10^{-5},\, 9\times10^{-5}\}$ for Adam, and
> $\eta \in \{10^{-2},\, 5\times10^{-2},\, 10^{-1},\, 5\times10^{-1}\}$ for SGD.
> We additionally evaluate SGD using the best learning rate obtained for Adam. The table below reports the validation accuracy along with the parameter displacement measured in $L_1$ and $L_2$ norms. Similar conclusions can be made in language tasks.
> | Metric | SGD ($\eta=7\times10^{-5}$) | SGD ($\eta=0.1$) | Adam ($\eta=7\times10^{-5}$) |
> |--------|-----------------------------|------------------|-------------------------------|
> | **Val Acc** | $43.87 \pm 24.02$ | $84.80 \pm 1.00$ | $85.95 \pm 0.64$ |
> | **$L_1$** | $5002.44 \pm 0.00$ | $6066.93 \pm 34.33$ | $31079.54 \pm 754.26$ |
> | **$L_2$** | $0.73 \pm 0.00$ | $1.26 \pm 0.01$ | $5.57 \pm 0.24$ |
>
>
> Further visualisations, such as eigenvalue spectra, are provided in Appendix K of the revised manuscript.
>
> **Additional experiment: Sparse training**
> We furthermore conduct a similar sparse training experiment where we use the reparameterization with varying depths and compare coupled and decoupled weight decay for a ResNet-50 on Imagenet. We confirm the same finding as for our previous experiment for a ResNet-20 on CIFAR-10: Coupled weight decay needs higher depth and weight decay to induce sparsity.
> This demonstrates that our insight on coupling also holds in larger scale settings. Results are reported in Figures 13, 14, and 15 and the validation accuracies in Table 6 of the revised manuscript.
>
> | Optimizer   | Weight Decay | depth L | L1 Mean   | L1 95% Std | Val Accuracy ± CI (%) |
> |-------------|--------------|---------|-----------|------------|-------------------|
> | AdamW       | 1e-1         | 2       | 29,417    | 26         | 76.23 ± 0.07      |
> | AdamW       | 1e-1         | 10      | 10,616    | 157        | 62.20 ± 0.25      |
> | AdamW       | 1e-4         | 2       | 3,532,879 | 1,282      | 73.32 ± 0.11      |
> | AdamW       | 1e-4         | 10      | 8,369,848 | 26,023     | 73.19 ± 0.04      |
> | Adam + wd   | 1e-1         | 2       | 0         | 0          | 1.95 ± 0.48       |
> | Adam + wd   | 1e-1         | 10      | 0         | 0          | 0.58 ± 0.06       |
> | Adam + wd   | 1e-4         | 2       | 4,473     | 18         | 73.35 ± 0.05      |
> | Adam + wd   | 1e-4         | 10      | 324       | 32         | 9.78 ± 0.94       |

---

> ### Author Response · Authors · 2025-11-21
> **Part 3**
>
> **Question:**
> Yes. It indeed means $\lim_{t \rightarrow \infty} x_t$ converges to a minimizer $x^*$. We have now explicitly stated this in the revised manuscript.
>
> [1] Jacobs, Tom and Rebekka Burkholz. “Mask in the Mirror: Implicit Sparsification.” ArXiv abs/2408.09966 (2024): n. pag.
>
> [2] Kolb, Chris et al. “Deep Weight Factorization: Sparse Learning Through the Lens of Artificial Symmetries.” ArXiv abs/2502.02496 (2025): n. Pag.
>
> [3] Tsilivis, Nikolaos et al. “Flavors of Margin: Implicit Bias of Steepest Descent in Homogeneous Neural Networks.” ArXiv abs/2410.22069 (2024): n. Pag.
>
> [4] Marchetti, Giovanni Luca et al. “Algebra Unveils Deep Learning -- An Invitation to Neuroalgebraic Geometry.” (2025).
>
> [5] Nam, Yoonsoo et al. “Position: Solve Layerwise Linear Models First to Understand Neural Dynamical Phenomena (Neural Collapse, Emergence, Lazy/Rich Regime, and Grokking).” ArXiv abs/2502.21009 (2025): n. pag.
>
> [6] Pesme, Scott and Nicolas Flammarion. “Saddle-to-saddle dynamics in diagonal linear networks.” Journal of Statistical Mechanics: Theory and Experiment 2024 (2023): n. pag.

---

### Official Review · Reviewer_iSvT · 2025-11-01

**Soundness:** 1
**Presentation:** 3
**Contribution:** 1
**Rating:** 0
**Confidence:** 5

**Summary:**

The paper proposes “steepest mirror flows” to explain why Adam/AdamW beat SGD in fine‑tuning via faster saddle escape and different implicit regularization.
Steeper (sign‑like) geometry helps saddle escape and that decoupled weight decay (AdamW) stabilizes feature learning, with theory for deep diagonal reparameterizations and small fine‑tuning case studies. Most formal results hold only in separable/diagonal settings; the transformer link is indirect; and empirical support for fine‑tuning claims is narrow.

**Strengths:**

The message that adam escapes saddles better than SGD is believable and may be cool. However, it is not well established. The rest of the claims are not substantiated.

**Weaknesses:**

#### How is this about fine-tuning transformers?
There is not even a linear diagonal attention there, no one ever claimed that a diagonal network is a good model for a transformer, because it is not. How do you argue this?

#### Mirror flow study is incremental and does not adequately support the thesis.
While the diagonal‑network analysis is neat, it is very similar to existent ones and does not bring any real novelty to the community.
*I believe, it is extremely incremental.*
It is way too limited to show that your sign-mirror-descent escapes saddles to claim that *adam* escapes saddles *in transformers*.
There is a too big of a jump from the mathematical argument to the goal. Moreover, I think that such a result is a perturbation of existent ones.

#### Order-2 saddles
So what? We know they are present in neural networks, actually, arguably also of higher order. A long line of works that you do not cite addresses this issue much more in general. That same line of work empirically and sometimes theoretically notice that they are not a problem in practice. You should discuss this line of research. On top of it saddles are not an issue in linear networks generally cause the standard initialization is with high probability outside of the area with saddles.

#### Title mismatch

Even the title is an oversell. It missmatch with the paper, this is not a paper about generally reparameterizing steepest descent as mirror flow. This is a paper about adam escaping saddles which lack novelty and does not support their claims.

#### Experiments
For cifar10 actually SGD generalizes better than Adam with CNNs, and this is also a classical result. I really do not understand the whole point of the paper and how these are supporting experiments.

#### Conclusions
Even though experiments are present, I thus believe the central claims are not adequately supported and the research methodology is not sound.

**Questions:**

-

---

> ### Author Response · Authors · 2025-11-21
> **Part 1**
>
> We would like to express our gratitude for your time and efforts in providing valuable comments on our manuscript. Below, we elaborate on your concerns in a detailed point-by-point response. In case of any open questions, we would be happy to discuss them.
>
> **Finetuning transformers**
> We focus on finetuning as in this setting small learning rates are preferred preventing from altering the representation too much as mentioned in Line 51. That is why our analysis based on flow is most representative for this setting. In addition, we discuss other mechanisms for saddle escape in the updated related work.
>
> Moreover, we meant that attention can be studied by tracking the dynamics of the matrix product $KQ$. The reparameterization has a similar product structure than $KQ$ but then in a simplified diagonal form. This was mentioned on Line 203 around Example 2.3. We have clarified this now in the introduction as well in the revised manuscript. Note that in the setting of the gradient flow, [4,5] consider the aligned or commuting dynamics which reduces to an evolution of the diagonal form.
> We further evaluate finetuning behavior on transformer architectures across both vision and language tasks going beyond the diagonal case. In particular, we finetune a pretrained ViT-Large model on the CIFAR-10 and Flowers datasets, and a pretrained BERT-base model on the MRPC task from the GLUE benchmark. The resulting generalization performance and parameter shifts measured in $L_1$ and $L_2$ norms are summarized in the table below. Across all settings, Adam consistently outperforms SGD at various learning rates. Additionally, Adam produces much larger $L_1$ movement—indicating more uniform updates across parameters. Detailed visualizations, including eigenvalue spectra, are provided in Appendix K of the revised manuscript.
>
> | Dataset   | Metric | SGD ($\eta=0.0001$) | SGD ($\eta=0.01$) | Adam ($\eta=0.0001$) |
> |-----------|--------|----------------------|--------------------|------------------------|
> | **CIFAR-10** | Val Acc | $73.27 \pm 3.68$ | $99.07 \pm 0.35$ | $99.28 \pm 0.07$ |
> |           | $L_1$   | $460.47 \pm 219.86$ | $24617.83 \pm 14406.4$ | $453934.91 \pm 22278.43$ |
> |           | $L_2$   | $0.48 \pm 0.059$ | $6.25 \pm 4.29$ | $39.47 \pm 1.01$ |
> | **Flowers** | Val Acc | $1.03 \pm 0.82$ | $98.94 \pm 0.05$ | $99.37 \pm 0.08$ |
> |           | $L_1$   | $25.71 \pm 30.48$ | $4655.83 \pm 576.49$ | $108583.62 \pm 2078.48$ |
> |           | $L_2$   | $0.04 \pm 0.02$ | $1.50 \pm 0.12$ | $8.35 \pm 0.16$ |
>
> | Metric | SGD ($\eta=7\times10^{-5}$) | SGD ($\eta=0.1$) | Adam ($\eta=7\times10^{-5}$) |
> |--------|-----------------------------|------------------|-------------------------------|
> | **Val Acc** | $43.87 \pm 24.02$ | $84.80 \pm 1.00$ | $85.95 \pm 0.64$ |
> | **$L_1$** | $5002.44 \pm 0.00$ | $6066.93 \pm 34.33$ | $31079.54 \pm 754.26$ |
> | **$L_2$** | $0.73 \pm 0.00$ | $1.26 \pm 0.01$ | $5.57 \pm 0.24$ |

---

> ### Author Response · Authors · 2025-11-21
> **Part 2**
>
> **Mirror flow is not incremental**
> We agree that our analysis studies a simplified, analytically tractable case, but this is a feature, not a bug.
> One of the main goals of theory in deep learning is to describe and explain phenomena that occur in practice. While our model is simple, it does give multiple valuable insights into the mechanics of relevant practical algorithms. Concretely, it allows us to make predictions for real world settings.
>
> **New geometric mechanism**
> Our geometric approach using mirror flow reveals a different principle at work that previous work has shown to escape saddle points. We show that the geometry is sufficient for escaping saddles in our setup, while previous work relies on noise and large learning rate or time rescaling [12]. Therefore, we believe revealing other mechanisms is of great value. In the revised manuscript, we include this discussion on noise and how it aids in saddle point escape in the related work. Our mechanism is corroborated by experiments, see for example Figure 5a where SGD with small learning rate cannot escape the saddle point indicating that noise is not sufficient. Moreover we have included additional experiments on saddle point escape in the appendix.
> Finally, revealing a structural mechanism helps us to identify where to look for developing new theoretical tools to study training dynamics. While we agree that a straightforward generalization using mirror flows may be hard, it can be used as a stepping stone for guiding other theoretical approaches and motivating identifying geometric properties. This is also in line with recent position papers calling for studying algebraic properties of models [10] and studying linear models [11].
>
> **Main use case: finetuning and saddle escape**
> Since Adam and its variants remain the most widely used optimizers for finetuning, and sign decent (steepest descent wit respect $L_{\infty}$ norm) serves as a proxy for Adam, the steepest mirror flow analysis naturally extends to the finetuning setting. In particular, finetuning typically employs a small learning rate to avoid large deviations from the pretrained parameters and to mitigate catastrophic forgetting. This aligns well with our flow analysis. In contrast, SGD has to use a significantly larger learning rate for saddle point escape, which is not desired in finetuning as it would induce catastrophic forgetting. This observation is substantiated by our real world experiments with transformers. We highlighted this in Figure 5a and in additional ablations in Appendix K with new additions on ViTs and Transformers in K.1 and K.2..
>
> **A theoretical use case: feature learning**
> Recent work has characterized the implicit bias for steepest descent flows for homogeneous neural networks on binary classifications for separable data [9]. We have applied their result in Theorem F.1 and highlighted that the $L_{\infty}$ max margin optimization problem cannot distinguish between solutions that correspond to overfitting or feature learning (sparsity). In contrast our theory predicts that depth changes the geometry and with that the bias towards sparsity and feature learning, which is also illustrated in the experiment depicted in Figure 4. This is a clear sign that we need additional tools and theory to describe the implicit bias further.
>
> **A practical use case: sparsity**
> As discussed, pointwise reparameterizations have been proposed together with weight decay to induce sparsity [7,8]. An integration with modern optimizers is valuable, as it leads to concrete design principles. Our theory predicts that for smaller $q$, so for SignGF, we need to have higher depth to induce sparsity (and thus feature learning). Moreover, Figure 5b shows that using AdamW requires higher depth to induce sparsity. This may lead to better sparse training algorithm design in the future.

---

> ### Author Response · Authors · 2025-11-21
> **Part 3**
>
> **Saddle point literature**
> We have extended our literature discussion:
>
> In finetuning, a small learning rate is preferred to not alter the representation too much to prevent catastrophic forgetting [6]. This clashes with the fact that saddle point escape needs time rescaling in gradient flow dynamics [12]. Note that different mechanisms that have been shown and studied allowing for saddle point escape are large learning rate and noise perturbation [1,2,3]. In contrast, our analysis reveals a different mechanism which only relies on the geometry of the dynamics. As we show in experiments (Figure 5a), SGD with a small learning rate can not escape saddle points while Adam can.
>
> **Do we need to escape saddles?**
> In a diagonal linear network, we may initialize outside the saddle point region with high probability. This may be true for particular random initializations. However, consider the following scenario: Let the ground truth vector be $x^* := (1,0, ...,0)$. then if the product at the first coordinate is negative, i.e. $\Pi_{i = 1}^L w_{i,0} < 0$, it has to move through $0$, meaning it has to move around the saddle point. This would be the case with probability $½$ assuming a symmetric random initialization. Therefore, it would still be important to be able to move away or around saddle points.
>
> **Title mismatch**
> The theoretical part considers a class of steepest descent methods acting on a reparameterization. We study a class of steepest descent methods as proxies for SGD and AdamW. This connection is made explicit in Appendix A and we have mentioned this connection throughout the paper. Therefore, our analysis is in line with the title. We will adapt the title as per suggestion of Reviewer NKAm to  “Never Saddle Down for Reparameterized Steepest Descent as Mirror Flow”. This refers more directly to the type of saddles we can avoid, namely the ones created by the reparameterization.
>
>
> **Experiments**
> In our experiments for finetuning Adam outperforms SGD even when SGD has a large learning rate, not the other way around. It is worth noting that SGD usually rely on weight decay to achieve good performance while finetuning Resnet on Cifar10. Here, we do not use weight decay to study the property of both optimisers themselves. We will highlight this more clearly in the revised manuscript by reporting the generalization performance in the main text in Table 2.
>
> We believe that these clarifications highlight that our thesis is well supported by the experiments.
>
>
> [1] Jin, Chi et al. “How to Escape Saddle Points Efficiently.” International Conference on Machine Learning (2017).
>
> [2] Fang, Ying, Zhaofei Yu, and Feng Chen. "Noise helps optimization escape from saddle points in the synaptic plasticity." Frontiers in neuroscience 14 (2020): 343.
>
> [3] Roy, A., Balasubramanian, K., Ghadimi, S., & Mohapatra, P. (2020). Escaping saddle-point faster under interpolation-like conditions. Advances in Neural Information Processing Systems, 33, 12414-12425.
>
> [4] Ataee Tarzanagh, Davoud et al. “Transformers as Support Vector Machines.” ArXiv abs/2308.16898 (2023): n. pag.
>
> [5] Jacobs, Tom et al. “Mirror, Mirror of the Flow: How Does Regularization Shape Implicit Bias?” ICML2025
>
> [6] Zhou, Chao, Tom Jacobs, Advait Gadhikar, and Rebekka Burkholz. “Pay Attention to Small Weights.” The Thirty-ninth Annual Conference on Neural Information Processing Systems (2025).
>
> [7] Jacobs, Tom and Rebekka Burkholz. “Mask in the Mirror: Implicit Sparsification.” ArXiv abs/2408.09966 (2024): n. pag.
>
> [8] Kolb, Chris et al. “Deep Weight Factorization: Sparse Learning Through the Lens of Artificial Symmetries.” ArXiv abs/2502.02496 (2025): n. Pag.
>
> [9] Tsilivis, Nikolaos et al. “Flavors of Margin: Implicit Bias of Steepest Descent in Homogeneous Neural Networks.” ArXiv abs/2410.22069 (2024): n. Pag.
>
> [10] Marchetti, Giovanni Luca et al. “Algebra Unveils Deep Learning -- An Invitation to Neuroalgebraic Geometry.” (2025).
>
> [11] Nam, Yoonsoo et al. “Position: Solve Layerwise Linear Models First to Understand Neural Dynamical Phenomena (Neural Collapse, Emergence, Lazy/Rich Regime, and Grokking).” ArXiv abs/2502.21009 (2025): n. pag.
>
> [12] Pesme, Scott and Nicolas Flammarion. “Saddle-to-saddle dynamics in diagonal linear networks.” Journal of Statistical Mechanics: Theory and Experiment 2024 (2023): n. pag.

---

### Official Review · Reviewer_f8Av · 2025-11-03

**Soundness:** 3
**Presentation:** 3
**Contribution:** 2
**Rating:** 4
**Confidence:** 4

**Summary:**

This paper studies continuous-time steepest descent methods, specifically algorithms taking the form:
$$\begin{align} d x_t = - \mathrm{sign} \left( \nabla_x f(x_t) \right) \odot \left| \nabla_x f(x_t) \right|^{q-1}, \end{align}$$
where $q \in [1,2]$.

Different values of $q$ result in different trajectories. For example:
 For $q=2$, the algorithm becomes gradient flow, which is the continuous-time approximation of gradient descent. For $q=1$, it becomes SignGF (Sign Gradient Flow), which serves as a good proxy for studying optimizers like Adam. The architecture on which the paper focuses is deep diagonal reparameterization, defined as $x = g(w) = \prod_{i=1}^{L} w_i$, where $x$ is represented as the product of $L$ scalars.

The authors demonstrate that under $\lambda$-balanced initializations, these steepest flows can be re-parameterized as steepest mirror flows. Using this reparameterization, the paper analyzes how quickly the flow escapes saddle points and investigates the effect of both coupled and decoupled weight decay.

**Strengths:**

The paper provides a seperation result for signGD with coupled and decoupled weight decay and show that they have different regularization properties which is interesting.

**Weaknesses:**

a) The paper focuses on deep diagonal reparameterizations which is a product of one-dimensional variables with a particular initialization shape. The setting is restrictive to generalize the results of the paper.

**Questions:**

1) How is the manifold regularizer derived ? Is it due to the fact that steepest descent with de coupled weight decay can be written as

$$ d( \nabla_x R(x_t) ) = - \mathrm {sign} \left( \nabla_x f(x_t) \right) \odot \left[   \nabla_x f(x_t) \right]^{q-1} dt  - \nabla M_{reg}(x) dt  $$

it would be nice to detail this as the manifold regularizer appears a bit abrupt.

2) The saddle points defined in Theorem 4.3 are not saddle for with coupled or decoupled weight decay so the feature learning results only work without the weight decay. This point needs to be clearly mentioned in the manuscript.

3) From the abstract, how is the deep diagonal reparameterizations a proxy for attention ?

---

> ### Author Response · Authors · 2025-11-21
>
> We would like to express our gratitude for your time and efforts in providing valuable comments on our manuscript. Below, we elaborate on your concerns in a detailed point-by-point response. In case of any open questions, we would be happy to discuss them.
>
> Weaknesses:
>
> **W1: Restrictive scope.**
> One of the main goals of theory in deep learning is to describe and explain phenomena that occur in practice. While our model is simple, it does give multiple valuable insights into the mechanics of relevant practical algorithms. Concretely, it allows us to make predictions for real world settings.
>
> **New geometric mechanism**
> Our geometric approach using mirror flow reveals a different principle at work that previous work has shown to escape saddle points. We show that the geometry is sufficient for escaping saddles in our setup, while previous work relies on noise and large learning rate or time rescaling [6]. Therefore, we believe revealing other mechanisms is of great value. In the revised manuscript, we include this discussion on noise and how it aids in saddle point escape in the related work. Our mechanism is corroborated by experiments, see for example Figure 5a where SGD with small learning rate cannot escape the saddle point indicating that noise is not sufficient. Moreover we have included additional experiments on saddle point escape in the appendix.
> Finally, revealing a structural mechanism helps us to identify where to look for developing new theoretical tools to study training dynamics. While we agree that a straightforward generalization using mirror flows may be hard, it can be used as a stepping stone for guiding other theoretical approaches and motivating identifying geometric properties. This is also in line with recent position papers calling for studying algebraic properties of models [4] and studying linear models [5].
>
> **Main use case: finetuning and saddle escape**
> Since Adam and its variants remain the most widely used optimizers for finetuning, and sign decent (steepest descent wit respect $L_{\infty}$ norm) serves as a proxy for Adam, the steepest mirror flow analysis naturally extends to the finetuning setting. In particular, finetuning typically employs a small learning rate to avoid large deviations from the pretrained parameters and to mitigate catastrophic forgetting. This aligns well with our flow analysis. In contrast, SGD has to use a significantly larger learning rate for saddle point escape, which is not desired in finetuning as it would induce catastrophic forgetting. This observation is substantiated by our real world experiments with transformers. We highlighted this in Figure 5a and in additional ablations in Appendix K with new additions on ViTs and Transformers in K.1 and K.2..
>
> **A theoretical use case: feature learning**
> Recent work has characterized the implicit bias for steepest descent flows for homogeneous neural networks on binary classifications for separable data [3]. We have applied their result in Theorem F.1 and highlighted that the L_{\infty} max margin optimization problem cannot distinguish between solutions that correspond to overfitting or feature learning (sparsity). In contrast our theory predicts that depth changes the geometry and with that the bias towards sparsity and feature learning, which is also illustrated in the experiment depicted in Figure 4. This is a clear sign that we need additional tools and theory to describe the implicit bias further.
>
> **A practical use case: sparsity**
> As discussed, pointwise reparameterizations have been proposed together with weight decay to induce sparsity [1,2]. An integration with modern optimizers is valuable, as it leads to concrete design principles. Our theory predicts that for smaller $q$, so for SignGF, we need to have higher depth to induce sparsity (and thus feature learning). Moreover, Figure 5b shows that using AdamW requires higher depth to induce sparsity. This may lead to better sparse training algorithm design in the future.

---

> ### Author Response · Authors · 2025-11-21
> **Part 2**
>
> Questions:
>
> **Q1: Manifold regularizer.** Yes this is indeed the correct representation of the dynamics with the on manifold regularization. We have extended Definition 4.15 to include this formula to make the introduction less abrupt.
>
> **Q2: Weight decay and saddles.** We mention this explicitly in the revised manuscript by including Remark 4.4 after Theorem 4.3. Note that the metric would still be smaller for larger $q$ indicating that escaping from a point near the set S would be harder for GF than SignGF.
>
> **Q3: Transformer Q and K.** We meant that attention can be studied by tracking the dynamics of the matrix product $KQ$. The reparameterization has a similar product structure as $KQ$, yet has a simpler diagonal form. Therefore, similar mechanisms are at play that guide the learning dynamics. This was mentioned in Line 203 close to Example 2.3. We have clarified this now in the introduction as well in the revised manuscript. Note that in the gradient flow setting, [7,1] draw the same analogy, as they consider the aligned or commuting dynamics of $KQ$, which reduces to an evolution of the diagonal form.
>
>
>
> [1] Jacobs, Tom and Rebekka Burkholz. “Mask in the Mirror: Implicit Sparsification.” ArXiv abs/2408.09966 (2024): n. pag.
>
> [2] Kolb, Chris et al. “Deep Weight Factorization: Sparse Learning Through the Lens of Artificial Symmetries.” ArXiv abs/2502.02496 (2025): n. Pag.
>
> [3] Tsilivis, Nikolaos et al. “Flavors of Margin: Implicit Bias of Steepest Descent in Homogeneous Neural Networks.” ArXiv abs/2410.22069 (2024): n. Pag.
>
> [4] Marchetti, Giovanni Luca et al. “Algebra Unveils Deep Learning -- An Invitation to Neuroalgebraic Geometry.” (2025).
>
> [5] Nam, Yoonsoo et al. “Position: Solve Layerwise Linear Models First to Understand Neural Dynamical Phenomena (Neural Collapse, Emergence, Lazy/Rich Regime, and Grokking).” ArXiv abs/2502.21009 (2025): n. pag.
>
> [6] Pesme, Scott and Nicolas Flammarion. “Saddle-to-saddle dynamics in diagonal linear networks.” Journal of Statistical Mechanics: Theory and Experiment 2024 (2023): n. Pag.
>
> [7] Ataee Tarzanagh, Davoud et al. “Transformers as Support Vector Machines.” ArXiv abs/2308.16898 (2023): n. pag.

---

### Author Response · Authors · 2025-11-21
**Part1**

**General response**
We would like to thank all the reviewers for their efforts in reviewing our work and providing valuable comments. We have incorporated the comments of all reviewers and updated our draft accordingly, which we believe makes our work stronger. To make it easy for the reviewers to go over these changes, the updated text is in blue in the draft.

We address here one of the main concerns regarding our work: the scope of the analysis. Furthermore, we provide additional experiments: finetuning for  vision transformers and LLMs and a larger scale sparsity experiment for a ResNet50 on Imagenet.

**Scope**
One of the main goals of theory in deep learning is to describe and explain phenomena that occur in practice. While our model is simple, it does give multiple valuable insights into the mechanics of relevant practical algorithms. Concretely, it allows us to make predictions for real world settings.

**New geometric mechanism**
Our geometric approach using mirror flow reveals a different principle at work that previous work has shown to escape saddle points. We show that the geometry is sufficient for escaping saddles in our setup, while previous work relies on noise and large learning rate or time rescaling [6]. Therefore, we believe revealing other mechanisms is of great value. In the revised manuscript, we include this discussion on noise and how it aids in saddle point escape in the related work. Our mechanism is corroborated by experiments, see for example Figure 5a where SGD with small learning rate cannot escape the saddle point indicating that noise is not sufficient. Moreover we have included additional experiments on saddle point escape in the appendix.
Finally, revealing a structural mechanism helps us to identify where to look for developing new theoretical tools to study training dynamics. While we agree that a straightforward generalization using mirror flows may be hard, it can be used as a stepping stone for guiding other theoretical approaches and motivating identifying geometric properties. This is also in line with recent position papers calling for studying algebraic properties of models [4] and studying linear models [5].

**Main use case: finetuning and saddle escape**
Since Adam and its variants remain the most widely used optimizers for finetuning, and sign decent (steepest descent wit respect $L_{\infty}$ norm) serves as a proxy for Adam, the steepest mirror flow analysis naturally extends to the finetuning setting. In particular, finetuning typically employs a small learning rate to avoid large deviations from the pretrained parameters and to mitigate catastrophic forgetting. This aligns well with our flow analysis. In contrast, SGD has to use a significantly larger learning rate for saddle point escape, which is not desired in finetuning as it would induce catastrophic forgetting. This observation is substantiated by our real world experiments with transformers. We highlighted this in Figure 5a and in additional ablations in Appendix K with new additions on ViTs and Transformers in K.1 and K.2..

**A theoretical use case: feature learning**
Recent work has characterized the implicit bias for steepest descent flows for homogeneous neural networks on binary classifications for separable data [3]. We have applied their result in Theorem F.1 and highlighted that the $L_{\infty}$ max margin optimization problem cannot distinguish between solutions that correspond to overfitting or feature learning (sparsity). In contrast our theory predicts that depth changes the geometry and with that the bias towards sparsity and feature learning, which is also illustrated in the experiment depicted in Figure 4. This is a clear sign that we need additional tools and theory to describe the implicit bias further.

**A practical use case: sparsity**
As discussed, pointwise reparameterizations have been proposed together with weight decay to induce sparsity [1,2]. An integration with modern optimizers is valuable, as it leads to concrete design principles. Our theory predicts that for smaller $q$, so for SignGF, we need to have higher depth to induce sparsity (and thus feature learning). Moreover, Figure 5b shows that using AdamW requires higher depth to induce sparsity. This may lead to better sparse training algorithm design in the future.

---

### Author Response · Authors · 2025-11-21
**Part 2**

**Additional experiment: Finetuning**
We also provide experiments in both the vision and language domain with transformer architectures.
Specifically, we finetune a ViT-Large pretrained on ImageNet on CIFAR-10 for 30 epochs and on Flowers for 15 epochs. As is standard in finetuning, the original classifier head is replaced with a newly initialized one. We evaluate two optimizers—SGD and Adam—with learning rates selected via a sweep: $\eta \in \{9e-5, 1e-4, 1e-4, 5e-4\}$ for Adam and $\eta \in \{1e-3, 5e-3, 1e-2, 5e-2, 1e-1\}$ for SGD. Additionally, we run SGD with the best Adam learning rate to further illustrate our observations on saddle escape. All experiments use batch size $128$, weight decay $0$, cosine annealing learning rate scheduling, and label smoothing of 0.1. The table below reports validation accuracy along with the parameter shift measured in $L_1$ and $L_2$ norms. Adam consistently outperforms SGD and induces a more uniform parameter update pattern, reflected in its substantially larger $L_1$ norm.

| Dataset   | Metric | SGD ($\eta=0.0001$) | SGD ($\eta=0.01$) | Adam ($\eta=0.0001$) |
|-----------|--------|----------------------|--------------------|------------------------|
| **CIFAR-10** | Val Acc | $73.27 \pm 3.68$ | $99.07 \pm 0.35$ | $99.28 \pm 0.07$ |
|           | $L_1$   | $460.47 \pm 219.86$ | $24617.83 \pm 14406.4$ | $453934.91 \pm 22278.43$ |
|           | $L_2$   | $0.48 \pm 0.059$ | $6.25 \pm 4.29$ | $39.47 \pm 1.01$ |
| **Flowers** | Val Acc | $1.03 \pm 0.82$ | $98.94 \pm 0.05$ | $99.37 \pm 0.08$ |
|           | $L_1$   | $25.71 \pm 30.48$ | $4655.83 \pm 576.49$ | $108583.62 \pm 2078.48$ |
|           | $L_2$   | $0.04 \pm 0.02$ | $1.50 \pm 0.12$ | $8.35 \pm 0.16$ |



In addition to vision experiments, we conduct a parallel study on language models. Specifically, we finetune a pretrained BERT-base model on the MRPC task from the GLUE benchmark. The model is finetuned for 5 epochs using both SGD and Adam. Learning rates are selected via a sweep:
$\eta \in \{5\times10^{-5},\, 7\times10^{-5},\, 9\times10^{-5}\}$ for Adam, and
$\eta \in \{10^{-2},\, 5\times10^{-2},\, 10^{-1},\, 5\times10^{-1}\}$ for SGD.
We additionally evaluate SGD using the best learning rate obtained for Adam. The table below reports the validation accuracy along with the parameter displacement measured in $L_1$ and $L_2$ norms. Similar conclusions can be made in language tasks.
| Metric | SGD ($\eta=7\times10^{-5}$) | SGD ($\eta=0.1$) | Adam ($\eta=7\times10^{-5}$) |
|--------|-----------------------------|------------------|-------------------------------|
| **Val Acc** | $43.87 \pm 24.02$ | $84.80 \pm 1.00$ | $85.95 \pm 0.64$ |
| **$L_1$** | $5002.44 \pm 0.00$ | $6066.93 \pm 34.33$ | $31079.54 \pm 754.26$ |
| **$L_2$** | $0.73 \pm 0.00$ | $1.26 \pm 0.01$ | $5.57 \pm 0.24$ |


Further visualisations, such as eigenvalue spectra, are provided in Appendix K of the revised manuscript.


**Additional experiment: Sparse training**
We furthermore conduct a similar sparse training experiment where we use the reparameterization with varying depths and compare coupled and decoupled weight decay for a ResNet-50 on Imagenet. We confirm the same finding as for our previous experiment for a ResNet-20 on CIFAR-10: Coupled weight decay needs higher depth and weight decay to induce sparsity.
This demonstrates that our insight on coupling also holds in larger scale settings. Results are reported in Figures 13, 14, and 15 and the validation accuracies in Table 6 of the revised manuscript.

| Optimizer   | Weight Decay | depth L | L1 Mean   | L1 95% Std | Val Accuracy ± CI (%) |
|-------------|--------------|---------|-----------|------------|-------------------|
| AdamW       | 1e-1         | 2       | 29,417    | 26         | 76.23 ± 0.07      |
| AdamW       | 1e-1         | 10      | 10,616    | 157        | 62.20 ± 0.25      |
| AdamW       | 1e-4         | 2       | 3,532,879 | 1,282      | 73.32 ± 0.11      |
| AdamW       | 1e-4         | 10      | 8,369,848 | 26,023     | 73.19 ± 0.04      |
| Adam + wd   | 1e-1         | 2       | 0         | 0          | 1.95 ± 0.48       |
| Adam + wd   | 1e-1         | 10      | 0         | 0          | 0.58 ± 0.06       |
| Adam + wd   | 1e-4         | 2       | 4,473     | 18         | 73.35 ± 0.05      |
| Adam + wd   | 1e-4         | 10      | 324       | 32         | 9.78 ± 0.94       |

---

### Author Response · Authors · 2025-11-21
**Part 3**

[1] Jacobs, Tom and Rebekka Burkholz. “Mask in the Mirror: Implicit Sparsification.” ArXiv abs/2408.09966 (2024): n. pag.

[2] Kolb, Chris et al. “Deep Weight Factorization: Sparse Learning Through the Lens of Artificial Symmetries.” ArXiv abs/2502.02496 (2025): n. Pag.

[3] Tsilivis, Nikolaos et al. “Flavors of Margin: Implicit Bias of Steepest Descent in Homogeneous Neural Networks.” ArXiv abs/2410.22069 (2024): n. Pag.

[4] Marchetti, Giovanni Luca et al. “Algebra Unveils Deep Learning -- An Invitation to Neuroalgebraic Geometry.” (2025).

[5] Nam, Yoonsoo et al. “Position: Solve Layerwise Linear Models First to Understand Neural Dynamical Phenomena (Neural Collapse, Emergence, Lazy/Rich Regime, and Grokking).” ArXiv abs/2502.21009 (2025): n. pag.

[6] Pesme, Scott and Nicolas Flammarion. “Saddle-to-saddle dynamics in diagonal linear networks.” Journal of Statistical Mechanics: Theory and Experiment 2024 (2023): n. pag.

---

### Author Response · Authors · 2025-12-03
**Summary part 1**

Dear Area Chair, Senior Area Chair, and Program Chairs,

Thank you for your time and effort in reviewing our paper and the rebuttal. We would like to summarize our work's main contribution and the discussion of the review process. Our revision directly addresses all major concerns with clearer framing, additional explanations, and substantially larger scale experiments. Below is a concise summary.
1) **Clearer Scope and Contribution**
 We clarified that the paper studies *reparameterized steepest-descent dynamics* and their mirror-flow geometry as a principled model for Adam/Sign-style optimizers. The model is a simplified theoretical framework but reveals geometric effects, for which we provide empirical evidence in the context of finetuning and sparse training. Concretely we study a family of steepest descent flows depending on a hyperparameter $q$ where $q = 2$ corresponds to gradient flow (GF) and $q =1$ to sign gradient flow (signGF).

2) **New Geometric Mechanism for Saddle Escape**
Our main conceptual contribution is the identification of a novel mechanism: the geometry of the dynamics provides an explanation for why optimizers like Sign Descent and Adam can escape from saddles, even with small learning rates—which are typical in the finetuning setting—and little noise. Experiments corroborate this finding: SGD struggles to escape saddles in finetuning, while SignGF/Adam can. Furthermore, we have extended the literature discussion on other mechanisms such as noise and large learning rates that also facilitate saddle point escape but are less helpful in the context of finetuning.

3) **Stronger Practical Evidence via Finetuning**
 We added finetuning experiments with a ViT-Large and BERT-base. Across all settings, Adam escapes saddles and reaches better solutions, matching our theoretical predictions.

4) **Clarified Feature Learning and Sparsity**
 We have clarified our definitions of stability and feature learning, which are the critical elements to understand how depth, optimizer parameter $q$, and weight decay (de)coupling interact. We find that a smaller $q$ allows for saddle escape for higher depth, and find that higher depth in this case is also needed for sparsity—and therefore feature learning. Additional ResNet-50 training on ImageNet experiments support our theory, complementary to our ResNet-20 training on CIFAR-10 experiments: 1) Decoupled weight decay needs larger depth/regularization to induce sparsity. 2) Coupled weight decay corresponds to solving the known equivalent regularized optimization problem, inducing higher sparsity with higher depth. Moreover, the high sparsity regime for coupled weight decay induces bad generalization, as it hampers the movement away from the critical point at zero.


5) **Technical Clarifications**
 We have resolved all notation issues, added a formula for on-manifold regularization to smooth the transition between paragraphs, clarified the relationship between reparameterization and transformers in the introduction, and expanded explanations of convergence and Bregman function conditions.


6) **Updated Title**
 To emphasize the specific type of studied saddle points, following the suggestion of Reviewer NKAm, we have changed our title to: “Never Saddle Down for Reparameterized Steepest Descent as Mirror Flow.”

**Criticism of Reviewer iSVT**

While the feedback by Reviewer iSVT was very critical, we believe we have addressed all their points of criticisms and refuted their strong claims regarding our contributions.
For example, “This paper is about Adam escaping saddles which lacks novelty“: the theory is concerned with steepest descent as mirror flow, which is novel as noted by the other reviewers, and we argue explicitly over the relationship between signGF ($q=1$) and Adam. Furthermore, the reviewer argues that SGD outperforms Adam on vision tasks. While this is true for pretraining, however, this is not the case for finetuning, as we show in various experiments in the image classification setting. Another critique is the model’s simplicity, which we argue is a strength and not a weakness. The reparameterization captures a part of the inherent structure of the training dynamics in deep learning: the depth. Simplifications are usually necessary for theory to provide intuitive insights and inspire hypotheses, which we have furthermore empirically confirmed.

**Discussion with Reviewer NKAm**

In discussion with Reviewer NKAm, we were able to clarify most of their concerns. Furthermore, in our second response we provided additional explanations to the 3 remaining points. (1) We state that Figure 3 is mainly an illustration, and provide an explicit calculation of the arclength, (2) we clarify that when $q < \frac{L}{L-1}$ we have instability, and (3) in order to have feature learning, we need to escape saddle points.

---

### Author Response · Authors · 2025-12-03
**Summary part 2**

**Conclusion**
Overall, the reviewers find the theoretical analysis interesting and novel, providing new insights into the training dynamics of modern optimization. Our theory provides new conjectures about saddle escape capabilities for Adam and AdamW that are particularly relevant in finetuning settings. We validate these conjectures in extensive experiments with both vision and language language models. Accordingly, the revised paper clearly presents a novel geometric mechanism for saddle escape.

---

### Meta-Review · Area_Chair_9z6D · 2026-01-05

**Summary:**

This work shows that when the parameters are reparametrized, the dynamics is equivalent to a family of steepest mirror flows. I personally think this work is reasonably interesting and novel. Showing the attractivity of these reparametrized saddles is useful for understanding the optimization of neural networks. A main weakness is the restriction of the setting to diagonal networks, but this could be a nice first step towards understanding these problems. I lean towards accepting this paper, although I do not mind rejecting it.

Reviewer iSvT gives a score of zero with very high confidence, which, in my opinion, is too adversarial, and I decided to discount it. Removing this score, this paper has a score of 4.67, which is essentially borderline. While I agree that the authors did not make a very good argument about transformers, the criticism about the relevance to transformers is also not strong, and I personally think the theory presented can be regarded as reasonably related to transformers (because matrix rescaling symmetry is really the most general form of rescaling symmetry, but the authors fail to point this out).

A missing reference is https://arxiv.org/abs/2107.11774, which shows that the problem with escaping these saddles (even with Adam) is really the noisy sampling of data -- and this is a limitation of the work that the authors should have acknowledged.

**Reviewer Concerns:**

I think the questions are reasonably well answered.

As explained above, I think it is better to discount the review of iSvT due to its overly strong adversariality and aggressive tone (such as 'so what?'). I think the criticisms are not well justified.

**Reviewer Scores:**

I find it difficult to predict this.

---

### Decision · Program_Chairs · 2026-01-26

Accept (Poster)